

**Biogeochemical controls on wintertime ammonium accumulation in the surface layer of**
**the Southern Ocean**
Shantelle Smith[1*], Katye E. Altieri[1], Mhlangabezi Mdutyana[1,2], David R. Walker[3], Ruan G.
Parrott[1], Kurt A.M. Spence[1], Jessica M. Burger[1], Sarah E. Fawcett[1,4]
[1] Department of Oceanography, University of Cape Town, Private Bag X3, Rondebosch,
Cape Town, South Africa
[2] Southern Ocean Carbon and Climate Observatory (SOCCO), CSIR, Rosebank, Cape
Town, South Africa
[3] Department of Conservation and Marine Sciences, Cape Peninsula University of
Technology, Cape Town, South Africa
[4] Marine and Antarctic Research centre for Innovation and Sustainability (MARIS),
University of Cape Town, Cape Town, South Africa
[*] Corresponding author: smtsha023@myuct.ac.za
**1.  Abstract**
The production and consumption of ammonium ($NH_4^+$) are essential upper-ocean nitrogen
cycle pathways, yet in the Southern Ocean where $NH_4^+$ has been observed to accumulate in
surface waters, its mixed-layer cycling remains poorly understood. For surface samples
collected between Cape Town and the marginal ice zone (MIZ) in winter 2017, we found that
$NH_4^+$ concentrations were five-fold higher than is typical for summer, and lower north than
south of the Subantarctic Front (SAF; 0.01–0.26 µM versus 0.19–0.70 µM). Our observations
confirm that $NH_4^+$ accumulates in the Southern Ocean's winter mixed layer, particularly in
polar waters. $NH_4^+$ uptake rates were highest near the Polar Front (PF; $12.9 \pm 0.4$ nM day$^{-1}$) and
in the Subantarctic Zone ($10.0 \pm 1.5$ nM day$^{-1}$), decreasing towards the MIZ ($3.0 \pm 0.8$ nM day$^{-1}$) despite high ambient $NH_4^+$ concentrations, likely due to low sea surface temperatures and
light availability. By contrast, rates of $NH_4^+$ oxidation were higher south than north of the PF
($16.0 \pm 0.8$ versus $11.1 \pm 0.5$ nM day$^{-1}$), perhaps due to the lower light and higher iron
conditions characteristic of polar waters. Augmenting our dataset with $NH_4^+$ concentration
measurements spanning the 2018/2019 annual cycle reveals that mixed-layer $NH_4^+$
accumulation south of the SAF likely derives from sustained heterotrophic $NH_4^+$ production in
late summer through winter that outpaces $NH_4^+$ consumption by temperature-, light, and iron-
limited microorganisms. Our observations thus imply that the Southern Ocean becomes a
biological source of $CO_2$ to the atmosphere for half the year not only because nitrate drawdown
is weak, but also because the ambient conditions favour net heterotrophy and $NH_4^+$
accumulation.
**2.  Introduction**
The Southern Ocean impacts the Earth system through its role in global thermohaline
circulation, which drives the exchange of heat and nutrients among ocean basins (Frolicher et
al., 2015; Popp et al., 1999; Sarmiento et al., 2004). The Southern Ocean also plays an integral



role in mediating climate, by transferring carbon to the deep ocean via its biological and
solubility pumps (Sarmiento & Orr, 1991; Volk & Hoffert, 1985) and through the release of
deep-ocean $CO_2$ to the atmosphere during deep-water ventilation (i.e., $CO_2$ leak; Broecker &
Peng, 1992; Lauderdale et al., 2013; Sarmiento & Toggweiler, 1984). Upper Southern Ocean
circulation is dominated by the eastward-flowing Antarctic Circumpolar Current (ACC) that
consists of a series of broad circumpolar bands ("zones") separated by oceanic fronts. Southern
Ocean fronts can drive water mass formation (Ito et al., 2010) and nutrient upwelling that
supports elevated biological activity (Longhurst, 1998; Sokolov & Rintoul, 2007).
Concentrations of the essential macronutrients, nitrate ($NO_3^-$) and phosphate ($PO_4^{3-}$), are
perennially high in Southern Ocean surface waters, in contrast to most of the global ocean.
Consumption of these nutrients, and thus primary productivity in the Southern Ocean, is limited
by numerous (often overlapping) factors, including temperature, light, micronutrient
concentrations, and grazing pressure (e.g., Boyd et al., 2001; Martin et al., 1990; Reay et al.,
2001; Smith Jr & Lancelot, 2004). These limitations vary with Southern Ocean sector (i.e.,
longitude), zone (i.e., latitude), and season, resulting in spatial and seasonal variations in
chlorophyll-a concentrations, primary production, plankton community composition, and
nutrient uptake regime (Shadwick et al., 2015; Thomalla et al., 2011; Mengesha et al., 1998;
Mdutyana et al., 2020). For example, the Antarctic Zone (AZ; see Text S1 for definitions of
zones and fronts), which includes the Open and Polar Antarctic Zones (OAZ and PAZ,
respectively), is characterized by sparser phytoplankton populations than the Polar Frontal
Zone (PFZ; Mengesha et al., 1998) even though AZ spring blooms generally host higher
diatom abundances than the blooms of the Subantarctic Zone (SAZ) and PFZ (Kopczyńska et
al., 2007). In addition to the seasonal cycles of temperature and light, Southern Ocean
ecosystems are influenced by seasonal changes in nutrient availability. In winter, deep mixing
replenishes the nutrients required for phytoplankton growth but the low temperatures and light
levels impede biological activity (Rintoul & Trull, 2001). Once the mixed layer shoals in spring
and summer, phytoplankton begin to consume the available nutrients until some form of
limitation (usually iron; Mtshali et al., 2019; Nelson et al., 2001) sets in. This balance between
wintertime nutrient recharge and summertime nutrient drawdown is central to the role of the
Southern Ocean in setting atmospheric $CO_2$ (Sarmiento & Toggweiler, 1984).
Iron limitation, which sets in following the spring/early summer bloom, causes phytoplankton
to increase their dependence on recycled ammonium ($NH_4^+$; Timmermans et al., 1998), which
has a far lower iron requirement than $NO_3^-$ assimilation (Price et al., 1994). The extent to which
phytoplankton rely on $NO_3^-$ versus $NH_4^+$ as their primary N source has implications for
Southern Ocean $CO_2$ removal since phytoplankton growth fuelled by upwelled $NO_3^-$ ("new
production") must be balanced on an annual basis by the export of sinking organic matter
("export production"; Dugdale & Goering, 1967), which drives $CO_2$ sequestration (i.e., the
biological pump; Volk & Hoffert, 1985). By contrast, phytoplankton growth on $NH_4^+$ or other
recycled N forms ("regenerated production") yields no net removal of $CO_2$ to the deep ocean
(Dugdale & Goering, 1967; Eppley & Peterson, 1979). To-date, considerable research has
focused on $NO_3^-$ cycling in the Southern Ocean mixed layer because of the importance of this
nutrient for the biological pump (e.g., DiFiore et al., 2006; Francois et al., 1992; Johnson et al.,
2017; Mdutyana et al., 2020; Primeau et al., 2013; Sarmiento & Toggweiler, 1984; Sigman &
Boyle, 2000) and global ocean fertility (Sarmiento et al., 2004). By contrast, the active cycling



of regenerated N within the seasonally-varying mixed layer – including the production of $NH_4^+$
and its consumption by phytoplankton uptake and nitrification (the microbial oxidation of $NH_4^+$
to nitrite ($NO_2^-$) and then $NO_3^-$) – remains poorly understood.
$NH_4^+$ is produced in the euphotic zone as a by-product of heterotrophic metabolism (i.e.,
ammonification; Herbert, 1999) and as a consequence of grazing by zooplankton (Lehette et al.,
2012; Steinberg & Saba, 2008), and is removed by phytoplankton uptake (in euphotic waters)
and nitrification (mainly in aphotic waters). Heterotrophic bacteria can also directly consume
$NH_4^+$ (Kirchman, 1994) and have been hypothesized to do so at significant rates in the Southern
Ocean mixed layer in winter (Cochlan, 2008; Mdutyana et al., 2020). $NH_4^+$ assimilation by
phytoplankton, in contrast to $NO_3^-$ consumption, requires relatively little energy (Dortch, 1990)
such that $NH_4^+$ is usually consumed in the surface ocean as rapidly as it is produced (Glibert,
1982; La Roche, 1983), resulting in very low open-ocean $NH_4^+$ concentrations (<0.2 μM;
Paulot et al., 2015). Additionally, $NH_4^+$ is often the preferred N source to phytoplankton
communities dominated by smaller species, while larger phytoplankton such as diatoms that
invest more energy in nutrient consumption specialize in the assimilation of $NO_3^-$ (e.g.,
Chisholm, 1992; Fawcett & Ward, 2011). Phytoplankton communities typically shift towards
smaller species when iron and/or light are limiting (Pearce et al., 2010; Tagliabue et al., 2014;
Deppeler & Davidson, 2017), since a higher cellular surface area-to-volume ratio renders small
phytoplankton less vulnerable to diffusion limitation (Hudson & Morel, 1993; Mei et al., 2009)
and a larger cell volume limits light absorption efficiency (Finkel et al., 2004; Fujuki &
Taguchi, 2002).
In addition to the consequences for small versus large phytoplankton abundance, which has
implications for the organic matter sinking flux (i.e., the strength of the biological pump;
Alldredge & Gotschalk, 1988; Richardson & Jackson, 2007) and higher trophic levels (e.g.,
Venkataramana et al., 2019), determining the dominant N source to phytoplankton provides a
means of estimating their potential for $CO_2$ removal, as per the new production paradigm
(Dugdale & Goering, 1967). The N isotopic composition ($\delta^{15}N$, in ‰ vs. $N_2$ in air, =
($^{15}N/^{14}N_{sample}/^{15}N/^{14}N_{air} - 1$) x 1000) of particulate organic N (PON) can be used to infer the
dominant N source to phytoplankton (Altabet, 1988; Lourey et al., 2003; Fawcett et al., 2011;
Van Oostende et al. 2017) since the assimilation of subsurface $NO_3^-$ yields PON that is higher
in $\delta^{15}N$ than that fuelled by recycled $NH_4^+$ (the $\delta^{15}N$ of which is inferred from isotopic
fractionation associated with its production to be low) (Macko et al. 1986; Silfer et al. 1992;
Checkley & Miller, 1989; Sigman et al, 1999). The $\delta^{15}N$ of bulk PON yields an integrated view
of the autotrophic N uptake regime (Fawcett et al., 2011; 2014; Lourey et al., 2003), which can
be complicated by overlapping processes such as bacterial degradation of organic matter
(Möbius, 2013; Smart et al., 2020). By contrast, $^{15}N$ tracer-derived N uptake rates provide an
instantaneous measure of the extent of phytoplankton reliance on new versus regenerated N
(Lipschultz, 2008), although these rates can be poorly-suited to extrapolation.
Nitrification was historically considered unimportant in euphotic zone waters due to the
evidence for light inhibition of nitrifiers (Hooper & Terry, 1974; Horrigan & Springer, 1990;
Olson, 1981; Schön & Engel, 1962) and competition with phytoplankton for $NH_4^+$ (Smith et al.,
2014; Ward, 1985; 2005; Zakem et al., 2018). However, this view has been challenged in
numerous oceanic regions (Yool et al., 2007) including the Southern Ocean (Smart et al., 2015;





Cavagna et al., 2015; Fripiat et al., 2015), with elevated rates of $NH_4^+$ oxidation recently
observed throughout the winter mixed layer in all major Southern Ocean zones (Mdutyana et
al., 2020). Wintertime upper-ocean $NH_4^+$ dynamics thus have implications for annual estimates
of carbon export potential, insofar as $NO_3^-$ produced by nitrification in the winter mixed layer
that is subsequently supplied to spring/summer phytoplankton communities constitutes a
regenerated rather than a new source of N on an annual basis (Yool et al., 2007; Mdutyana et
al., 2020).
Surface concentrations of $NH_4^+$ and other reduced N forms are often near-zero in spring and
early/mid-summer in the open Southern Ocean (Daly et al., 2001; Sambrotto & Mace, 2000;
Savoye et al., 2004; Henley et al., 2020) as $NH_4^+$ is readily consumed by phytoplankton. In late
summer, a peak in $NH_4^+$ concentration has been observed and attributed to enhanced bacterial
and zooplankton activity following elevated phytoplankton growth (Mengesha et al., 1998;
Becquevort et al., 2000; Dennett et al., 2001; Sambrotto & Mace, 2000; El-Sayed, 1984). One
might expect this high-concentration $NH_4^+$ pool to be quickly consumed given the capacity of
phytoplankton for rapid $NH_4^+$ uptake, leaving the winter mixed layer $NH_4^+$-deplete. However,
the limited available observations suggest that wintertime surface $NH_4^+$ concentrations are high
(often >1 µM), particularly south of the Subantarctic Front (SAF) (Bianchi et al., 1997;
Philibert et al., 2015; Mdutyana et al., 2020; Henley et al., 2020). If ambient $NH_4^+$ is not
depleted following the late summer peak in its concentration despite the high rates of $NH_4^+$
uptake and oxidation measured in autumn and winter (Bianchi et al., 1997; Thomalla et al.,
2011; Philibert et al., 2015; Mdutyana et al., 2020), then $NH_4^+$ regeneration must be occurring
at an elevated rate, either coincident with $NH_4^+$ consumption in winter and/or prior to this in
late summer and/or autumn. Under these conditions, the Southern Ocean mixed layer may
become net heterotrophic and thus a biological source of $CO_2$ to the atmosphere.
Here, we focus on $NH_4^+$ cycling in the Southern Ocean mixed layer in winter, a season assumed
to be largely biologically dormant (Arrigo et al., 2008; Schaafsma et al., 2018) and for which
$NH_4^+$ cycle data are scarce. We confirm that $NH_4^+$ accumulates throughout the winter mixed
layer, particularly south of the SAF, and examine a number of potential causes thereof,
including a contribution from the residual late-summer $NH_4^+$ pool, sustained $NH_4^+$ production
in the autumn/winter, and limited $NH_4^+$ uptake and/or oxidation in winter. We further consider
the possible drivers and implications of each of these scenarios. Finally, using $NH_4^+$
concentration data collected over a full annual cycle, we propose a seasonal cycle for the
mixed-layer $NH_4^+$ pool south of the SAF.
**3. Methods**
3.1 Cruise track and sample collection
Samples were collected on the southward (S) and northward (N) legs of a winter cruise
between Cape Town, South Africa, and the marginal ice zone (MIZ) of the Southern Ocean
onboard the R/V *SA Agulhas II* (VOY25; 28 June to 13 July 2017) (Fig. 1). Leg S, involving
only surface underway collections, crossed the Atlantic sector of the Southern Ocean, while leg
N bordered the Atlantic and Indian sectors (30°E; WOCE IO6 line) and involved eight
conductivity-temperature-depth (CTD) hydrocast stations. Frontal positions were determined





using the ship's hull-mounted thermosalinograph and supported by temperature, salinity, and
oxygen concentration data from CTD measurements made during leg N. The criteria for
determining frontal positions included identifying sharp gradients in potential temperature,
salinity, potential density, and oxygen concentrations (Belkin & Gordon, 1996; Lutjeharms &
Valentine, 1984; Orsi et al., 1995). For leg N, the mixed layer depth (MLD) was determined
for each Niskin (up)cast as the depth between 10 m and 400 m at which the Brunt Väisälä
Frequency squared, $N^2$, reached a maximum (Carvalho et al., 2017).
During leg S, samples were collected every four hours from the ship's underway system (~7 m
intake; "underway stations") while samples on leg N were collected from surface (~10 m)
Niskin bottles mounted on the CTD rosette ("CTD stations"). $NH_4^+$ samples were also taken at
13 depths over the upper 500 m at all CTD stations. At all stations (underway + CTD), ~40 mL
of unfiltered seawater was collected for the analysis of $NH_4^+$ concentrations in duplicate 50 mL
high density polyethylene (HDPE) bottles that had been stored ("aged") with
orthophthaldialdehyde (OPA) working reagent. Unfiltered seawater was collected in 50 mL
polypropylene centrifuge tubes for the analysis of macronutrients including urea. Immediately
following collection, $NH_4^+$ and nutrient samples were stored at -20°C.
Duplicate size-fractionated chlorophyll-a samples were collected by filtering seawater (500
mL) through 25 mm-diameter glass fibre filters with pore sizes of 0.3 µm and 2.7 µm
(Sterlitech, GF-75 and Grade D, respectively). Acetone was added to foil-wrapped borosilicate
test tubes containing the filters that were then incubated at -20 °C for 24 hours. Additionally,
duplicate seawater samples (4 L) were gently vacuum-filtered through combusted 47 mm-
diameter, 0.3 µm-pore size GF-75 filters for POC and PON concentrations and $\delta^{15}$N-PON.
Filters were stored in combusted foil envelopes at -80°C.
For microscopy, unfiltered seawater samples (250 mL) were collected along leg S in darkened
glass bottles and immediately fixed by the addition of 2.5 mL of Lugol's iodine solution (2%
final concentration), then stored at low room temperature away from direct sunlight until
analysis. Surface seawater samples (~2 mL) were collected in triplicate microcentrifuge tubes
for flow cytometry. These samples were fixed with glutaraldehyde (1% final concentration) and
stored at -80°C until analysis (Marie et al., 2005; Vaulot et al., 1989).
Ten incubation experiments were conducted during leg S to measure the rate of net primary
production (NPP). $NH_4^+$ and chlorophyll-a samples were collected at the beginning of each
experiment as described above. In addition, four NPP experiments were conducted during leg
N using seawater collected from Niskin bottles fired at 10 m. In all cases, pre-screened (using
200-µm mesh to remove large grazers) seawater was collected in three 2-L polycarbonate
bottles to which $NaH^{13}CO_3$ was added at ~5% of the ambient DIC concentration. Bottles were
incubated on the deck for 5 to 6.5 hours in custom-built incubators shaded with neutral-density
screens to mimic the 55% light level (typically encountered between 5 and 10 m) and supplied
with running surface seawater. Following incubation, each sample was divided (1 L per size
fraction) and gently vacuum filtered through 0.3 µm, and 2.7 µm glass fibre filters that were
stored in combusted foil at -80°C until analysis.





N uptake (as $NO_3^-$, $NH_4^+$ and urea) and $NH_4^+$ oxidation experiments were conducted at five
stations during leg S, with $NH_4^+$ oxidation measured at two additional stations the ice edge
(Fig. 1). On leg N, experiments were also conducted using seawater collected from 10 m at the
same four CTD stations as the NPP experiments. In all cases, duplicate polycarbonate bottles
were amended with $^{15}N$-labeled $NO_3^-$, $NH_4^+$ or urea at ~10% of the ambient N concentration,
estimated based on past wintertime measurements (Mdutyana et al., 2020) and, in the case of
$NH_4^+$, coincident shipboard analyses. Incubations were carried out as described above for NPP.
For $NH_4^+$ oxidation, duplicate black 250 mL HDPE bottles were amended with 0.1 µM $^{15}NH_4^+$
and 0.1 µM $^{14}NO_2^-$ (the latter as a "trap" for the $^{15}NO_2^-$ produced by $NH_4^+$ oxidation given the
expected low ambient $NO_2^-$ concentrations (<0.2 µM; Zakem et al., 2018; Fripiat et al., 2019;
Mdutyana et al., 2020). $NH_4^+$ oxidation bottles were incubated for 24 hours under the same
temperature conditions as the N uptake and NPP experiments. Subsamples (50 mL) were
collected from each bottle immediately following the addition of $^{15}NH_4^+ + ^{14}NO_2^-$ ($T_0$) and at the
end of the experiments ($T_f$), and frozen at -20°C until analysis.
3.2. Sample processing
3.2.1. Ammonium concentrations
$NH_x$ ($NH_4^+ + NH_3$) concentrations were measured shipboard following the fluorometric method
of Holmes et al. (1999) and using a Turner Designs Trilogy fluorometer 7500-000 equipped
with a UV module. The detection limit, calculated as twice the pooled standard deviation of all
standards, was 0.06 µM. $NH_x$ is hereafter referred to as $NH_4^+$ given convention in the
oceanographic literature and the dominance of $NH_4^+$ over $NH_3$ at seawater pH. To prevent
possible in/efflux of contaminant ammonia ($NH_3$) due to the temperature difference between
winter surface waters and the shipboard laboratory, samples were frozen immediately upon
collection and OPA working reagent was subsequently added to the frozen samples prior to
defrosting them for analysis. Samples were slowly warmed to room temperature in a water bath
after OPA addition, incubated in the dark for four hours once defrosted, then analysed in
triplicate. Standards and blanks were made daily using Type-1 ultrapure Milli-Q water.
Precision was ± 0.03 µM for replicate samples and standards.
3.2.2. Macronutrient concentrations
Following the cruise, duplicate seawater samples were analysed manually for $NO_2^-$ and $PO_4^{3-}$
(Bendschneider & Robinson, 1952; Murphy & Riley, 1962) using a Thermo Scientific Genesys
30 Visible spectrophotometer. Standards and blanks were prepared in Type-1 ultrapure Milli-Q
water. Precision was ± 0.05 µM for $NO_2^-$ and ± 0.06 µM for $PO_4^{3-}$, and the detection limit for
$NO_2^-$ and $PO_4^{3-}$ was 0.05 µM. $NO_3^- + NO_2^-$ and $Si(OH)_4$ concentrations were measured in
duplicate using a Lachat QuickChem 8500 Series 2 flow injection autoanalyzer. Aliquots of a
certified reference material (JAMSTEC) were measured during each run to ensure
measurement accuracy (SD ≤ 2%). The precision of the $NO_3^- + NO_2^-$ and $Si(OH)_4$
measurements was ± 0.4 µM and ± 0.2 µM, respectively, and the detection limit was 0.1 µM
and 0.2 µM. The $NO_3^-$ concentration was calculated by subtraction (i.e., $[NO_3^- + NO_2^-] - [NO_2^-$
]), with error propagated according to standard statistical practices. Urea-N (hereafter, urea)
concentrations were determined according to the room-temperature, single-reagent colorimetric



method (Revilla et al., 2005) using a Thermo Scientific Genesys 30 Visible spectrophotometer;
precision was ± 0.04 µM and the detection limit was 0.04 µM.

### 3.2.3. Chlorophyll-a concentrations

Chlorophyll-a concentrations ([chl-a]) were determined shipboard using the nonacidified
fluorometric method (Welschmeyer, 1994). The fluorometer was calibrated with an analytical
standard (*Anacystis nidulans*, Sigma-Aldrich®) prior to and following the cruise. The [chl-a] of
the 0.3-2.7 µm size class (picophytoplankton) was calculated by subtracting the measured [chl-
a] of the >2.7 µm size class (nanophytoplankton) from the >0.3 µm size class (bulk). We
assumed based on previous work (e.g., Hewes et al., 1985, 1990; Weber & El-Sayed, 1987) that
the wintertime phytoplankton community would be composed primarily of small cells (i.e.,
typically <10 µm), such that we did not separate micro- from nanophytoplankton.

### 3.2.4. Bulk POC, PON and $\delta^{15}$N-PON

The NPP and N uptake filters were fumed with hydrochloric acid in a desiccator for 24 hours to
remove inorganic C, then dried for 24 hours at 40°C and packaged in tin cups. Filters to be
measured for $\delta^{15}$N were dried in the same way as the NPP/N uptake filters, but not acidified.
Samples were analysed using a Delta V Plus isotope ratio mass spectrometer (IRMS) coupled
to a Flash 260 elemental analyser, with a detection limit of 0.17 µmol C and 0.07 µmol N and
precision of ±0.005 At% for C and N. Eight unused pre-combusted filters (blanks) were
prepared with each batch run of ~88 samples. POC and PON content was determined from
daily standard curves of IRMS area versus known C and N masses. For isotope ratios, sample
measurements were standardised to Merck Gel ($\delta^{15}$N = 7.5‰, $\delta^{13}$C = -20.1‰; Merck), Valine
($\delta^{15}$N = 12.1‰, $\delta^{13}$C = -26.8‰; Sigma), Choc ($\delta^{15}$N = 4.3‰, $\delta^{13}$C = -17.8‰), and NH₄Cl
($\delta^{15}$N = -0.6‰), internal laboratory standards calibrated against IAEA reference materials.

### 3.2.5. Size-fractionated rates of NPP and N uptake

Carbon and N uptake rates (NPP, $\rho NH_4^+$, $\rho NO_3^-$, $\rho Urea$) were calculated according to the
equations outlined in Dugdale & Wilkerson (1986) as:
$$\rho M = \frac{[PM] \, x \, (At\%_{meas} - At\%_{amb})}{T \, x \, (At\%_{init} - At\%_{amb})}$$
(Eqn 1)

$$\text{where, } At\%_{init} = \frac{([M] \, x \, At\%_{amb}) + ([M_{tracer}] \, x \, At\%_{tracer})}{[M] + [M_{tracer}]}$$
(Eqn 2)

Here, M is the species of interest (C, NH₄⁺, NO₃⁻, or urea); $\rho M$ is the uptake rate of that species
(nM day⁻¹); [PM] is the concentration of POC or PON (µM) on the filters; [M] is the ambient
concentration of DIC, NH₄⁺, NO₃⁻, or urea at the time of sample collection; [M_{tracer}] is the
concentration of NaH¹³CO₃, ¹⁵NH₄⁺, ¹⁵NO₃⁻, or ¹⁵N-urea added to the incubation bottles; and T
is the incubation period (days). The PM and $\rho M$ of the picoplankton size class was calculated
by subtracting the >2.7 µm-filter measurements (i.e., nanoplankton) from the >0.3 µm-filter
(i.e., bulk) measurements.





The specific carbon fixation rate ($V_C$) was calculated as $\rho C/POC$ and the specific uptake rate of
total N ($V_{Ntot}$) was calculated as $\rho N_{tot}/PON$ (where $\rho N_{tot} = \rho NH_4^+ + \rho NO_3^- + \rho Urea$). The f-ratio
(Eppley & Peterson, 1979), used to estimate the fraction of NPP potentially available for
export, was then calculated as:
$$\qquad f - ratio = \frac{V_{NO_3}}{V_{NO_3} + V_{NH_4} + V_{urea}} \qquad\qquad (Eqn\ 3)$$
No urea uptake experiments were conducted at the underway stations at 50.7ºS and 55.5ºS
(both AZ); here, the f-ratio was calculated omitting $V_{urea}$. For the other AZ stations at which
urea uptake was measured, including $V_{urea}$ decreased the fraction of new-to-total production by
only 4-8% compared to f-ratio calculations based on $V_{NO3}$ and $V_{NH4}$.
3.2.6. Ammonia oxidation rates
The azide method of McIlvin and Altabet (2005) was used to convert $NO_2^-$ deriving from $NH_4^+$
oxidation to $N_2O$ gas that was measured using a Delta V Plus IRMS with a custom-built purge-
and-trap front end (McIlvin & Casciotti, 2011). This configuration yields a detection limit of
0.2 nmol N with a $\delta^{15}N$ precision of ± 0.1‰. The $\delta^{15}N$ of $NO_2^-$ was derived from $^{45}N_2O/^{44}N_2O$
and the rate of $NH_4^+$ oxidation ($NH_4^+{}_{ox}$; nM day$^{-1}$) was calculated following Peng et al. (2015)
as:
$$\qquad NH_4^+{}_{ox} = \frac{\Delta(^{15}NO_2^-)}{f_{NH_4}^{15} \times T} \qquad\qquad (Eqn\ 4)$$
Here, $\Delta(^{15}NO_2^-)$ is the change in the concentration of $^{15}NO_2^-$ (nM) between the start and end of
the incubation, calculated as the difference in the measured $\delta^{15}N$ of $NO_2^-$ between the $T_f$ and $T_0$
samples, $f_{NH_4}^{15}$ is the fraction of the $NH_4^+$ substrate labelled with $^{15}N$ at the start of the
incubation, and T is the incubation length (days). All $^{15}NO_2^-$ produced during the incubations
was assumed to derive from $^{15}NH_4^+$ oxidation. The detection limit ranged from 0.02 to 0.11 nM
day$^{-1}$, calculated according to Santoro et al. (2013) and Mdutyana et al. (2020).
3.2.7 Plankton community composition
Microphytoplankton and microzooplankton groups (>5-10 μm) were identified and counted in
a subsample (20 mL) from each 250 mL amber bottle using the Utermöhl technique (Utermöhl,
1958) and following the recommendations of Hasle (1978). Plankton groups and individual
species were counted and identified using an inverted light microscope (Olympus CKX41) at
200x magnification.
Cells were also enumerated using an LSR II flow cytometer (BD Biosciences) equipped with
blue, red, violet, and green lasers. Here, our focus was on enumerating pico- and nanoplankton.
Prior to flow cytometric analysis, 1 mL of each sample was incubated with 10 μL of 1% (v/v)
SYBR Green-I, which stains DNA, at room temperature in the dark for 10 minutes (Marie et
al., 1997). Autofluorescence was detected in the following bandpass filter sets, named for
commonly-used fluorochromes: allophycocyanin (APC, 660/20), R-phycoerythrin (PE)
(575/25), fluorescein isothiocyanate (FITC) (525/20), PE-cyanine 7 (PE-Cy7) (780/40), PE-
Texas Red (610/20), and Pacific Blue (450/50). Background 'noise' was gated out based on the
forward and side light scatter values (FSC = 800 and SSC = 200). DNA-containing cells were
isolated in each sample based on their detected autofluorescence on the FITC bandpass filter
(above a minimal fluorescence threshold of x10$^3$ RFU). Subsequently, based on their detected
autofluorescence on the APC bandpass filter relative to the PE bandpass filter, the isolated
DNA-containing cells were grouped into the following populations: Nano- and picoeukaryotes,
and *Synechococcus*. Additionally, small heterotrophic cells were identified as containing DNA
but with the lowest detected autofluorescence across all bandpass filters, except the FITC
(Marie et al., 1997; Gasol & Del Giorgio, 2000). All particles lacking DNA were considered
detritus. For each sample, data acquisition was terminated when a minimum of 5000 and
maximum of 10000 events were recorded. The populations of interest were gated using FlowJo
10.3 software (TreeStar, Inc.; www.flowjo.com). Relative cell sizes were determined using 60
μL of SPHERO™ Blank Calibration Particles, 1.8 – 2.2 μm in diameter, added to 1 mL of
selected samples to yield a final concentration of ~6x10$^5$ particles mL$^{-1}$. Relative to the 1.8 –
2.2 μm calibration beads, nanoeukaryotes were larger than 2.2 μm, picoeukaryotes and
heterotrophic cells were smaller than 1.8 μm, and *Synechococcus* exhibited a range of sizes
around 2 μm, with two distinct subgroups; one of ~2 μm in size and another slightly larger than
2.2 μm (see Fig. S1). *Synechococcus* was isolated from the nanoeukaryotes by its pigment
characteristics – both subgroups of *Synechococcus* had high PE relative to APC content
(Barlow et al., 1985; Marie et al., 1997), whereas nanoeukaryotes had high APC and PE.
Since no direct measurements of NH$_4^+$ regeneration (i.e., heterotrophy) were made in this study,
potential heterotrophic activity is evaluated from the abundance of heterotrophic cells
determined via flow cytometry and the ratio of bulk POC to PON concentrations (POC:PON).
The availability of organic matter to heterotrophs is estimated from the abundance of detritus
and the ratio of POC-to-chl-a concentrations (POC:chl-a; Holm-Hansen et al., 1989).
The correlations among latitude, N concentrations, inorganic carbon and N uptake rates, and
NH$_4^+$ oxidation rates were investigated at the 5% significance level using the Pearson
correlation coefficient and the R packages, stats (R Core Team, 2020) and corrplot (Wei &
Simko, 2017).
**4. Results**
4.1 Hydrography
Sea surface temperature (SST) decreased from Cape Town (~34°S) to the edge of the MIZ
(61.7°S) by ~17 °C (Fig. 1). During leg N, fairly deep MLDs were observed (124-212 m),
similar to June and July climatological MLDs compiled from Argo float data for this region
(Dong et al., 2008). While the focus of this study is the surface (i.e., upper ~10 m), we describe
the hydrography of the mixed layer here to demonstrate that sampling took place under
conditions typical of winter, with the deep MLDs evincing ongoing wintertime mixing and
associated nutrient recharge. Where not specified, the trends discussed below refer to the
surface data only. For each parameter, the average ± 1 standard deviation (SD) calculated for
each Southern Ocean zone is reported in Table 1.
4.2 Macronutrient concentrations





The surface and mixed-layer concentrations of $NH_4^+$ ranged from below detection to 0.70 µM
along legs S and N (Fig. 2a and b). The concentrations were higher in the PFZ, OAZ, and PAZ
(0.42 ± 0.01 µM, 0.52 ± 0.01 µM, and 0.58 ± 0.01 µM, respectively) than in the Subtropical
Zone (STZ) and SAZ (0.08 ± 0.03 µM and 0.06 ± 0.01 µM, respectively), with a sharp gradient
observed in the PFZ, just south of the SAF. South of the SAF, high $NH_4^+$ concentrations
persisted near-homogeneously throughout the mixed layer, ranging from 0.65 ± 0.01 µM at
station 58.5°S to 0.27 ± 0.01 µM at station 48.0°S, with concentrations that were below
detection north of the SAF (Fig. 2b). Beneath the mixed layer, the $NH_4^+$ concentration
decreased rapidly at all stations to values below detection by 200 m.
The concentrations of $PO_4^{3-}$ and $NO_3^-$ increased southwards from <1 µM and <10 µM in the
STZ to >1.5 µM and >20 µM in the PFZ, OAZ, and PAZ (Fig. S2a and 2c), with the sharpest
gradients occurring near the SAF. The concentrations of $Si(OH)_4$ increased rapidly across the
PF, from an average of 3.2 ± 1.1 µM between 35.0°S and 48.0°S to 45.6 ± 0.6 µM between
52.1°S and 58.9°S (Fig. S2c). The $NO_2^-$ concentrations were consistently low across the
transect (0.16 ± 0.02 µM; Fig. S2b), as were the concentrations of urea (0.20 ± 0.04 µM),
although slightly lower urea concentrations were observed in the SAZ than in the other zones.
### 4.3 Chlorophyll-a, POC and PON
The highest bulk (i.e., >0.3 µm) [chl-a] was observed near the South African continental shelf,
decreasing across the STF and remaining low thereafter (Fig. 3a), consistent with previous
autumn and winter studies (Froneman et al., 1999; Philibert et al., 2015; Scharek et al., 1994).
The proportion of chl-a in the >2.7 µm size class (hereafter, "nano+" size class) varied across
the region but was >50% at all stations, with higher (>80%) contributions near the fronts and at
many OAZ and PAZ stations (Fig. 3b). The nano+ contribution was ≤60% at only five stations
(three in the SAZ, two in the OAZ).
The concentrations of bulk POC and PON were highest north of the STF and slightly higher in
the OAZ than in the SAZ and PFZ (Fig. S3a and b). The contribution of the nano+ size fraction
to POC and PON across the transect was 80.6 ± 31.8% and 69.8 ± 50.3%, respectively (Fig.
S3c and d). The ratio of bulk POC:chl-a (weight:weight) was on average low in the STZ, SAZ,
and PFZ, and reached a maximum in the OAZ (Fig. 4a). Contrastingly, the ratio of POC:PON
(mol:mol) appeared to decrease southwards, although there was no significant difference
among zones (p-value > 0.05) (Fig. 4b). The $\delta^{15}$N-PON also decreased southwards from the
STZ and SAZ to the PFZ and OAZ (Fig. 4c). Despite considerable differences among zones,
the $\delta^{15}$N-PON was relatively homogenous within each zone.
### 4.4 Rates of net primary production, nitrogen uptake, and ammonium oxidation
The surface rates of bulk NPP were high in the STZ, and two- to six-fold higher in the SAZ and
PFZ than has been reported previously for the Atlantic sector in winter (Mdutyana et al., 2020;
Froneman et al., 1999) (Fig. 5a). By contrast, NPP was low in the OAZ, consistent with
previous measurements (Kottmeier & Sullivan, 1987; Mdutyana et al., 2020). The relative
contribution of the small size class (0.3-2.7 µm) generally increased southwards, from 14.6% at
37.0°S to 75.6% at 53.5°S, before decreasing to <20.0% at ~55.5°S near the SACCF.



The bulk $NH_4^+$ uptake rates ($\rho NH_4^+$) generally increased southwards from the STZ to the SAZ
and PFZ, and then decreased across the OAZ to reach a minimum at the southernmost station
(58.5°S; 3.0 ± 0.8 nM day$^{-1}$) (Fig. 5b). In the nano+ size fraction, $\rho NH_4^+$ changed little
latitudinally, although it was slightly lower in the PFZ than in the other zones. The contribution
of nanoplankton to $\rho NH_4^+$ ranged from 32.8% in the PFZ to 71.9% in the STZ. The bulk $NO_3^-$
uptake rates ($\rho NO_3^-$) were also low in the STZ, while the highest $\rho NO_3^-$ was measured in the
SAZ before decreasing southwards. $\rho NO_3^-$ in the nano+ size class followed the same trend as
total community $\rho NO_3^-$, with the nanoplankton accounting for 71.5 ± 0.3% of bulk $\rho NO_3^-$ on
average. The rates of bulk urea uptake ($\rho Urea$) were highest in the STZ, with the SAZ and the
PFZ hosting similar rates, and the lowest rates were measured in the OAZ. $\rho Urea$ for the nano+
size class followed a similar trend to bulk $\rho Urea$, and nanoplankton accounted for 51.8% of
$\rho Urea$ in the SAZ to 100% in the PAZ. The uptake rates of the different N forms were not
significantly correlated with one another or with the ambient N concentrations (Fig. S4).
Surface ammonium oxidation rates ($NH_4^+_{ox}$) increased southwards, with higher $NH_4^+_{ox}$ in the
OAZ and PAZ than in the STZ, SAZ, and PFZ (Fig. 5c). Generally, $NH_4^+_{ox}$ was comparable to
previous wintertime measurements from the surface of the open Southern Ocean (Bianchi et al.,
1997; Mdutyana et al., 2020), and also similar to summertime rates measured deeper in the
mixed layer in the Ross and Scotia Seas (Tolar et al., 2016). $NH_4^+_{ox}$ was not correlated with the
ambient $NH_4^+$ concentration (Fig. S4).
4.5 Plankton community composition
The abundance of microplankton, analysed at 16 stations on leg S, was generally low, with the
highest cell counts at stations 37.2°S and 41.3°S in the STZ and no cells counted at 38.1°S
(STZ) and 55.5°S (OAZ) (Fig. 6a). Total microplankton abundance was on average higher in
the STZ than in the SAZ, PFZ, and OAZ. The greatest diversity of microplankton groups was
observed at 41.3°S near the AF and at 50.0°S near the PF. The observation of enhanced
plankton diversity and abundance near the fronts, particularly the PF, is consistent with
previous studies showing higher biomass and variability in phytoplankton communities
associated with these features (Hense et al., 2000; Kopczynska et al., 2007; Moore & Abbott,
431  2000).

Centric diatoms (including *Planktoniella*, *Coscinodiscus*, and *Thalassiosira* species) were
detected only at 58.9°S (3 cells mL$^{-1}$), the southernmost station. Pennate diatoms (including
*Pseudo-nitzschia*, *Pleurosigma*, and *Navicula* species) were more abundant in the STZ, PFZ,
and OAZ, with negligible abundances observed in the SAZ. Higher pennate diatom abundances
occurred near the PF (7 cells mL$^{-1}$), as has been observed in summer (e.g., Bracher et al.,
1999). Dinoflagellates were identified at every station except 38.1°S and were most abundant
in the STZ and PFZ. At all but three stations, small (<15 μm) dinoflagellates were the most
abundant group, although the larger *Protoperidinium* dinoflagellate species (mainly
heterotrophic; Jeong & Latz, 1994) were almost as abundant in the PFZ and at 54.0°S. The
abundance of microzooplankton (ciliates only, 20-200 μm) was highest across the STZ, and
microzooplankton were also identified in the PFZ at 46.1°S (3 cells mL$^{-1}$) and 48.9°S (3 cells
mL$^{-1}$) and in the OAZ at 50.0°S (1 cells mL$^{-1}$) and 54.0°S (4 cells mL$^{-1}$). All other stations
were characterized by negligible (<1 cells mL$^{-1}$) microzooplankton abundances.



Nano- and picoeukaryotes, *Synechococcus*, and small heterotrophs (collectively, "small cells")
sampled at 13 stations along leg S were roughly $10^3$-times more abundant than the
microplankton (Fig. 6b). Notwithstanding a lack of data from the STZ, the highest small cell
abundances occurred in the SAZ near the SAF. Across the transect, picoeukaryotes were
generally more abundant than all other phytoplankton groups (average picoeukaryote
contribution to total small cells of 12-54%; nanoeukaryotes of 7-39%; *Synechococcus* of 15-
42%). A similar trend was observed previously for the Southern Ocean in spring (Detmer &
Bathmann, 1997) and late summer (Fiala et al., 1998), in contrast to mid-summer observations
showing nanoplankton dominance (e.g., Ishikawa et al., 2002; Weber & El-Sayed, 1987).
Additionally, picoeukaryotes were two- to three orders of magnitude more abundant in the SAZ
and PFZ than in the OAZ. Nanoeukaryotes dominated small cell abundances near the PF at
50.0°S (39%) and in the southern OAZ at 55.5°S (36%), while *Synechococcus* dominated at
42.7°S and 54.0°S (42% and 33%, respectively). Nanoeukaryote abundance was higher in the
SAZ than in the PFZ and OAZ, as was the abundance of *Synechococcus*.
The relative contribution of heterotrophs to total small cells varied considerably (10-62%),
reaching a maximum south of the PF at 53.0°S and 57.8°S (62% and 50%; Fig. 7a).
Heterotroph abundance followed a similar pattern to that of the nanoeukaryotes, with higher
abundances in the SAZ than in the PFZ and OAZ. The food source available to heterotrophs,
represented by the small detrital particles, was highest near the southern edge of the SAF. More
generally, detrital particles were more abundant in the PFZ than in the SAZ and OAZ. The
relative contributions of detrital, photosynthetic, and heterotrophic particles are shown in Fig.
S5.
**5. Discussion**
5.1 Drivers of $NH_4^+$ cycling in the surface layer of the Southern Ocean
Previous work has suggested that $NH_4^+$ accumulates in the Southern Ocean mixed layer
following the late summer increase in zooplankton abundance and heterotrophic activity, then
decreases into autumn as heterotrophic activity subsides, to be depleted by winter due to
advective processes and consumption (Koike et al., 1986; Serebrennikova & Fanning, 2004).
However, our data show that $NH_4^+$ concentrations are elevated in the Southern Ocean mixed
layer in winter, particularly south of the SAF (Fig. 2). Similarly elevated winter surface-layer
$NH_4^+$ has been observed previously in both the Atlantic and Indian sectors, with concentrations
typically increasing towards the south (Philibert et al., 2015; Mdutyana et al., 2020; Bianchi et
al., 1997). Numerous overlapping processes are likely involved in setting the ambient $NH_4^+$
concentrations, as summarized in Fig. 8. In this study, we directly measured the rates of $NH_4^+$
uptake by different size fractions of the winter plankton community, as well as the rates of
$NH_4^+$ oxidation. We infer the contribution of heterotrophic bacteria and microzooplankton to
$NH_4^+$ production from cell count data and the abundance of small heterotrophs relative to
phytoplankton and detritus. For the $NH_4^+$ cycle processes in Fig. 8 that are not quantified or
inferred here – microzooplankton grazing, atmospheric $NH_4^+$ deposition, $NH_3$ air-sea exchange,
sea-ice melt, and dissolved organic nitrogen (DON) conversion to $NH_4^+$ –, we consider their
potential role in Southern Ocean $NH_4^+$ cycling based on findings reported in the literature.



The high $NH_4^+$ concentrations observed in the winter PFZ and AZ (OAZ + PAZ) may result
from net $NH_4^+$ accumulation during late summer, autumn and/or winter. The persistence of high
$NH_4^+$ concentrations that are near-homogeneously distributed throughout the mixed layer
suggests a residence time for the winter $NH_4^+$ reservoir in excess of the time-scale for upper-
ocean mixing. One implication of this suggestion is that the wintertime $NH_4^+$ pool likely
reflects processes that occurred earlier in the season, as well as those that are ongoing. We posit
that the elevated $NH_4^+$ concentrations in the PFZ and AZ may result from higher wintertime
rates of $NH_4^+$ production than consumption and/or from the gradual but incomplete depletion in
winter of $NH_4^+$ produced mainly in late summer and autumn. We evaluate both possibilities
throughout the discussion below.
5.1.1 Ammonium consumption
*Ammonium uptake* – Microbial growth is limited in the winter Southern Ocean (Arrigo et al.,
2008; Smith Jr et al., 2000, Takao et al., 2012), resulting in low cell abundances and nutrient
uptake rates (Church et al., 2003; Iida & Odate, 2014; Mdutyana et al., 2020). While the
concentrations of chl-a and rates of NPP were low across our transect, they were not negligible
(Fig. 3a and 5a), consistent with previous reports for this season (Mordy et al., 1995; Pomeroy
& Wiebe, 2001). Southern Ocean phytoplankton are adapted to survive suboptimal conditions;
for example, numerous species achieve their maximum growth rates at temperatures that are
considerably lower than the optimal growth temperatures of temperate and tropical species (2-9
°C versus 10-30 °C and 15-35 °C, respectively), with sharp declines in growth rates observed
for temperatures outside this range (Boyd et al., 2013; Coello-Camba & Agusti, 2017; Fiala &
Oriol, 1990). In addition, ice-free Southern Ocean waters typically extend to <60°S in the east
Atlantic and west Indian sectors in winter, so that although irradiance levels may not be optimal
for phytoplankton growth, there is always some light available for photosynthesis. The hostile
conditions of the open winter Southern Ocean do not, therefore, prevent ecosystem functioning
(Pomeroy & Wiebe, 2001), although the microbial dynamics and associated biogeochemical
processes differ from those occurring in summer (Smart et al., 2015; Mdutyana et al., 2020).
We measured fairly low $NH_4^+$ uptake rates in surface waters (3.0-13.2 nM day$^{-1}$; Fig. 5b)
compared to previous wintertime observations (ranging from 32-66 nM day$^{-1}$; Cota et al., 1992;
Mdutyana et al., 2020; Philibert et al., 2015). Such low rates, if generally representative of
winter, would limit mixed-layer $NH_4^+$ drawdown, especially south of the PF where $\rho NH_4^+$ was
particularly low. Recycled N ($NH_4^+$ + urea) nonetheless accounted for most of the N consumed,
including in the AZ (Fig. 5b).
The $\delta^{15}$N-PON data (Fig. 4c) suggest that this elevated reliance on recycled N persisted from
the late summer. In theory, PON generated in early- through mid-summer from the
consumption of upwelled $NO_3^-$ ($\delta^{15}$N-$NO_3^-$ of 5.2‰ in the AZ and 6.2‰ in the SAZ; Smart et
al., 2015; Fripiat et al., 2019) will have a $\delta^{15}$N of ~0‰ in the AZ and 1-2‰ in the SAZ given
the isotope effect of $NO_3^-$ assimilation and the degree of seasonal $NO_3^-$ drawdown (Sigman et
al., 1999; Granger et al., 2004; 2010). Such $\delta^{15}$N-PON values have indeed been observed in
early- and mid-summer (Lourey et al. 2003; Smart et al. 2020; Soares et al., 2015). By late
summer, $\delta^{15}$N-PON declines to -5 to -1‰, with the lowest values occurring in the AZ (Lourey
et al. 2003; Smart et al. 2020; Trull et al., 2008). Since the $\delta^{15}$N of recycled N is expected to be





low (<0‰; Checkley & Miller, 1989, Macko et al., 1986), the early-to-late summer decline in
$\delta^{15}$N-PON implies a switch from dominantly $NO_3^-$- to dominantly recycled N-supported
phytoplankton growth (Lourey et al., 2003). For the SAZ, the subsequent late summer-to-
winter rise in $\delta^{15}$N-PON (i.e., from ~ -1‰ to 1-2.5‰; Fig. 4c) has previously been attributed to
PON decomposition by heterotrophic bacteria (Smart et al., 2020), during which $^{14}$N-$NH_4^+$ is
preferentially remineralized, leaving the remaining PON enriched in $^{15}$N (Möbius, 2013). That
$NH_4^+$ concentrations are not elevated in the SAZ mixed layer in winter (Fig 2b.) indicates that
the remineralized $NH_4^+$ is rapidly re-assimilated by phytoplankton and/or oxidized to $NO_2^-$ in
this zone. In the AZ, the $\delta^{15}$N-PON of -3 to -1‰ that we observe in winter surface waters
requires the sustained consumption of low-$\delta^{15}$N N (i.e., recycled $NH_4^+$ and urea) to offset a
remineralization-driven $\delta^{15}$N rise similar to that of the SAZ. We conclude that Southern Ocean
phytoplankton dominantly consume regenerated N from late summer until at least July (albeit
at low rates in winter), particularly south of the PF.
The fact that the $NH_4^+$ concentration was high in the winter mixed layer despite $NH_4^+$ being the
preferred phytoplankton N source in late summer through winter implies that low rates of $NH_4^+$
uptake contributed to the accumulation of this N form. Multiple factors may cause low rates of
photoautotrophic $NH_4^+$ uptake, including deplete $NH_4^+$ and micronutrient concentrations, light
limitation, and low temperatures. North of the SAF, $NH_4^+$ concentrations below detection likely
limited $\rho NH_4^+$, as evidenced by the fact that in a series of experiments conducted on the same
cruise, $\rho NH_4^+$ increased with the addition of $NH_4^+$ at these stations (Mdutyana, 2021). By
contrast, south of the SAF, $NH_4^+$ concentrations were similar to or higher than the half-
saturation constant ($K_m$) derived for $NH_4^+$ uptake in the winter Southern Ocean (0.2 to 0.4 μM;
Mdutyana, 2021), suggesting that something other than $NH_4^+$ availability was limiting to
phytoplankton at these latitudes.
Iron is not directly involved in $NH_4^+$ assimilation but is required for electron transport during
photosynthesis and respiration (Raven, 1988). While iron limitation is widespread across the
Southern Ocean (Janssen et al., 2020; Pausch et al., 2019; Viljoen et al., 2019), iron availability
appears to be higher in winter than during other seasons (Mtshali et al., 2019; Tagliabue et al.,
2014) due to enhanced mixing, storms, and increased aeolian deposition (Coale et al., 2005;
Honjo et al., 2000; Sedwick et al., 2008). The fact that $\rho NO_3^-$ and $\rho NH_4^+$ were generally similar
across the transect (Fig. 5b) argues against a dominant role for iron in controlling $\rho NH_4^+$ since
$NO_3^-$ assimilation has a far higher iron requirement than $NH_4^+$ consumption (Morel et al.,
561   1991).

In contrast to $NH_4^+$ and iron availability, light limitation is exacerbated in winter due to low
insolation, increased cloud-cover, and mixed layers that can be hundreds of meters deeper than
the euphotic zone (Brightman & Smith Jr., 1989; Buongiorno Nardelli et al., 2017; Sallée et al.,
2010). Light is thus often considered the dominant constraint on Southern Ocean primary
productivity in this season (Thomalla et al., 2011; Llort et al., 2019; Wadley et al., 2014).
However, since $NH_4^+$ consumption by phytoplankton is fairly energetically inexpensive
(Dortch, 1990), it should occur even under low light (recognizing that light remains critical for
coincident $CO_2$ fixation). Heterotrophic bacteria can also consume $NH_4^+$ (Kirchman, 1994),
including in the dark since they derive energy from organic carbon oxidation rather than light.



At an ecosystem level, therefore, $NH_4^+$ consumption may not be primarily limited by light,
although this parameter clearly strongly controls the rate of NPP (Fig. 5a).
Previous observations suggest that temperature influences $NH_4^+$ uptake, especially in winter
(Glibert, 1982; Reay et al., 2001). The negative effect of temperature appears to be enhanced
under high-nutrient and low-light conditions, at least in the case of phytoplankton growth rates
(Baird et al., 2001). Additionally, Southern Ocean phytoplankton may be psychrotolerant and
not psychrophilic, which means that while they can function at *in situ* wintertime temperatures,
their optimal temperatures for growth and photosynthesis are higher (Reay et al., 2001; Smith
Jr & Harrison, 1991; Tilzer et al., 1986). Experiments conducted coincident with our sampling
showed that the maximum rate of $NH_4^+$ uptake ($V_{max}$) achievable by the *in situ* community was
strongly negatively correlated with temperature and latitude (Mdutyana, 2021), with the latter
parameter indicative of the combined role of light, temperature, and possibly iron, the
concentration of which appears to increase from the SAZ to the AZ (Tagliabue et al., 2012).
We conclude that these three drivers, along with $NH_4^+$ availability north of the SAF, all play a
role in controlling photoautotrophic $NH_4^+$ uptake in the winter Southern Ocean, with complex
interactions among them that are difficult to disentangle.
In addition to physical and chemical limitations, microbial preference for other N species may
impact the depletion of the $NH_4^+$ pool. For example, the preferential uptake of urea and other
DON species by some organisms (e.g., cyano- or heterotrophic bacteria) could dampen total
$NH_4^+$ uptake rates. While large contributions of urea to total N uptake have previously been
observed in the Southern Ocean in summer and autumn (predominantly in the SAZ; Joubert et
al., 2011; Thomalla et al., 2011), we measured fairly low ρUrea (Fig. 5b), which is perhaps
unsurprising given the low ambient urea concentrations (Table 1). The exceptions were stations
37°S and 43.0°S where ρUrea was higher than $\rho NH_4^+$, coincident with very low ambient $NH_4^+$
(0.10 μM and below detection) and relatively high urea concentrations (0.36 μM and 0.15 μM).
Community composition can also alter the N uptake regime. Smaller phytoplankton, such as
the numerically-dominant nano- and picoeukaryotes, are more likely to consume $NH_4^+$ and urea
than $NO_3^-$ (Koike et al., 1986; Lee et al., 2012, 2013), especially in the Southern Ocean where
$NO_3^-$ assimilation is severely limited by iron and light availability (Sunda & Huntsman, 1997).
Across our transect, the sum of $NH_4^+$ and urea uptake (i.e., reduced N uptake) exceeded $NO_3^-$
uptake for both the total phytoplankton community (transect average of $12.0 \pm 0.9$ nM day$^{-1}$ for
reduced N versus $5.8 \pm 1.0$ nM day$^{-1}$ for $NO_3^-$; f-ratio of 0.36) and the 0.3-2.7 μm size fraction
($5.0 \pm 1.2$ nM day$^{-1}$ versus $1.9 \pm 1.2$ nM day$^{-1}$; f-ratio of 0.27 (Fig. 5b). That said, the $NO_3^-$
uptake rates were not negligible, including in the 0.3-2.7 μm size fraction. In the PFZ and AZ,
$NO_3^-$ uptake by the 0.3-2.7 μm size fraction was more strongly correlated with the abundance
of picoeukaryotes than *Synechococcus* (r = 0.75 and 0.03, respectively), consistent with
observations of dominant reliance on $NO_3^-$ by picoeukaryotes and $NH_4^+$ by *Synechococcus* in
other ocean regions (Casey et al., 2009; Fawcett et al., 2011, 2014; Treibergs et al., 2014;
Painter et al., 2014). Nonetheless, *Synechococcus* can consume all N forms (Capone et al., 2008
and references therein) and has evolved strategies to conserve iron by using other trace metals
in some enzymes (Palenik et al., 2003). Thus, *Synechococcus* may be adapted to consume $NO_3^-$
in the Southern Ocean when reduced N concentrations are near depletion (e.g., north of the
SAF in winter), but are likely to consume $NH_4^+$ as long as it is available, as implied by their



strong correlation with $NH_4^+$ concentration south of the SAF (r = 0.65). In the nano+ size class,
$NO_3^-$ uptake was likely driven in the SAZ by dinoflagellates and some nanoeukaryotes, and in
the PFZ and AZ by diatoms, which remain active in these zones in winter (Weir et al., 2020).
By contrast, nanoeukaryotes, which have a higher per-cell nutrient requirement than the
equally-abundant picoeukaryotes, may have dominated $NH_4^+$ uptake in the PFZ and AZ given
that higher nanoeukaryote abundances corresponded with lower $NH_4^+$ concentrations at a
number of stations (e.g., stations 50.0°S, 51.1°S, and 55.5°S; Fig. 6b).
The low abundances of diatoms and dinoflagellates and absence of coccolithophores (Fig. 6a)
across our transect is expected given the limitations imposed on nutrient uptake and $CO_2$
fixation by winter Southern Ocean conditions. The lower surface area-to-volume ratio of larger
cells means that they rapidly experience diffusion-limitation of $NH_4^+$ and micronutrient uptake
and are more susceptible to light limitation (Finkel et al., 2004), resulting in their being
outcompeted by smaller species for essential resources (Franck et al., 2005; Cavender-Bares et
al., 1999). The near-absence of centric diatoms is also best explained thus, particularly given
their low surface area-to-volume ratio compared to pennate species (Kobayashi & Takahashi,
2002) that are more likely to consume $NH_4^+$ (Semeneh et al., 1998) and were more abundant.
That said, we did not observe a clear relationship between pennate diatom abundance and $NH_4^+$
concentration, except proximate to the PF (stations 47.9°S, 48.9°S, and 50.0°S) where higher
pennate abundance was associated with lower $NH_4^+$. Diatom success in winter may also be
limited by enhanced mixing, as this group is generally adapted for stratified waters
(Kopczynska et al., 2007).
In sum, $NH_4^+$ uptake rates were low across our transect but not negligible, indicating that
phytoplankton activity in winter, which is dominated by smaller species, represents a sink for
$NH_4^+$. Hostile Southern Ocean conditions imposed limitations on $NH_4^+$ uptake that varied with
latitude, with $NH_4^+$ concentrations controlling $\rho NH_4^+$ north of the SAF, while light and
temperature were important south of the SAF, with a possible supporting role for iron.
Additionally, *Synechococcus,* nanoeukaryotes, and pennate diatoms likely dominated $NH_4^+$
consumption, consistent with previous observations from the Southern Ocean and elsewhere
(Klawonn et al., 2019; Semeneh et al., 1998).
*Ammonium oxidation* – Nitrification removes more mixed-layer $NH_4^+$ than phytoplankton
consumption south of the PF, with $NH_4^+$ oxidation rates that were two- to five-times the co-
occurring $NH_4^+$ uptake rates (Fig. 5c). The comparative success of $NH_4^+$ oxidisers may be due
to decreased competition with phytoplankton for $NH_4^+$ in winter, augmented by decreased
photoinhibition (Wan et al., 2018; Lu et al., 2020) and elevated $NH_4^+$ availability (Baer et al.,
2014; Mdutyana et al., 2020; Mdutyana, 2021). One implication of the dominance of $NH_4^+$
oxidation is that in addition to the limitations on phytoplankton $NH_4^+$ uptake discussed above,
low phytoplankton success in the AZ may also result from nitrifiers outcompeting
phytoplankton under conditions of low incident light and enhanced mixing for scarce resources
(e.g., trace elements required for enzyme functioning, such as iron and copper; Shafiee et al.,
2019; Maldonado et al., 2006; Amin et al., 2013).
Although $NH_4^+$ oxidisers appear to be truly psychrophilic given the southward increase in $NH_4^+$
oxidation rates, the effect of temperature is difficult to disentangle in an environment with





multiple overlapping drivers. While several studies have reported a minimal effect of
temperature on $NH_4^+$ oxidation rates (Bianchi et al., 1997; Baer et al., 2014; Horak et al., 2013;
Mdutyana et al., in review), nitrifiers in the winter Southern Ocean may yet be living at
suboptimal temperatures (Jones et al., 1988). Indeed, a relative inefficiency of $NH_4^+$ oxidation
at low temperatures could be inferred from the general southward increase in the ratio of $NH_4^+$
to $NO_2^-$ concentration ($NH_4^+$:$NO_2^-$; Fig. S6). This trend is unexpected given the lower affinity
of nitrite oxidizing bacteria for $NO_2^-$ compared to that of ammonia oxidisers for $NH_4^+$, which
should result in an accumulation of $NO_2^-$ relative to $NH_4^+$ (Pachiadaki et al., 2017; Zakem et al.,
2018; Zhang et al., 2020). However, other factors such as mixing and increased predation and
viral lysis can also affect $NH_4^+$:$NO_2^-$, and the dynamics of $NH_4^+$ are less predictable in space
and time than those of $NO_2^-$ because of their different residence times (Zakem et al., 2018).
The $K_m$ derived for $NH_4^+$ oxidation in the winter Southern Ocean has recently been reported to
be low (0.03 to 0.14 µM), with ammonia oxidizers observed to become saturated at ambient
$NH_4^+$ concentrations of ~0.1-0.2 µM (Mdutyana, 2021). This means that south of the SAF in
winter 2017, ammonia oxidizers were not substrate limited (further implied by the lack of
correlation between $NH_4^+_{ox}$ and $NH_4^+$ concentration; Fig. S4), which raises the question of why
$NH_4^+$ oxidation did not occur at higher rates. The answer may involve temperature, in that
psychrophilic organisms can be less responsive to high substrate concentrations at low
temperatures (Baer et al., 2014). Another possibility is that $NH_4^+$ oxidation was iron-limited
(Shiozaki et al., 2016; Mdutyana, 2021), with a recent culture study revealing the surprisingly
low affinity for iron of the globally-abundant ammonia oxidiser, *Nitrosopumilus maritimus*
(Shafiee et al., 2019). In any case, $NH_4^+$ oxidisers were moderately successful across the surface
Southern Ocean in winter, with low light, reduced competition with phytoplankton, and
substrate repletion likely explaining the elevated $NH_4^+$ oxidation rates south of the PF
compared to the stations to the north.
5.1.2 Ammonium production and other inputs
$NH_4^+$ production, although not measured directly in this study, must be sustained during the
winter to retain an $NH_4^+$ pool that is high in concentration relative to the early summer. With
low or no $NH_4^+$ production in the autumn and winter, the $NH_4^+$ pool south of the SAF would be
depleted in 10 to 38 days (median of 21 days) given the consumption rate ($\rho NH_4^+ + NH_4^+_{ox}$)
and $NH_4^+$ concentration measured at each station (Text S2). Heterotrophic $NH_4^+$ production
must, therefore, be ongoing in winter despite the limited production of PON substrate.
*Heterotrophic activity by bacteria* – Heterotrophic bacteria may contribute significantly to
$NH_4^+$ accumulation via ammonification of organic N (Hewes et al., 1985; Koike et al., 1986;
Treguer & Jacques, 1992), including in winter (Rembauville et al., 2017). However, since these
bacteria are likely more active in late summer and autumn when both temperature and the
supply of fresh PON are high (Becquevort et al., 2000; Dennet et al., 2001), we expect that the
winter $NH_4^+$ pool includes residual $NH_4^+$ produced towards the end of the growing season. At
the time of our sampling, heterotrophic abundances were ten-fold lower to two-fold higher than
total pico- and nanophytoplankton abundances (Fig. 7a). Higher ratios of heterotrophic-to-
photosynthetic cells occurred at stations with higher $NH_4^+$ concentrations (e.g., stations 48.9°S,
53.0°S, 54.0°S and 57.8°S), suggesting a role for the short-term balance between $NH_4^+$





production and consumption in controlling the ambient $NH_4^+$ concentration in winter. The
heterotrophic bacteria were likely consuming detritus (as opposed to living cells), with the
relative availability of detrital substrate evident from the high detrital particle counts (Fig. 7b)
and the persistently high POC:chl-a ratios, particularly south of the PF (Fig. 4a; Holm-Hansen
et al., 1989). Additionally, a southward increase in heterotrophic biomass (which has a C:N
ratio typically ≤5:1) can be inferred from the southward decline in POC:PON (Fig. 4b; Frigstad
et al., 2011; del Giorgio & Cole, 1998), although this could also be due to iron and light
limitation of $CO_2$ fixation (Mongin et al., 2006; Talmy et al., 2016). Active remineralization of
detritus south of the SAF is further implicated by lower ratios of detrital-to-heterotrophic
particles coincident with higher $NH_4^+$ concentrations (Fig. 7b). Finally, the specific uptake rate
of $NO_3^-$ + $NH_4^+$ + urea (i.e., $V_{Ntot}$) exceeded that of $CO_2$ fixation ($V_C$) at some AZ stations (Fig.
S7). Similar observations in the winter Southern Ocean have been interpreted as indicating the
consumption of reduced N by heterotrophic bacteria (thus evincing their activity), which occurs
in the absence of $CO_2$ fixation, thereby decoupling $V_C$ and $V_{Ntot}$ (Text S2; Mdutyana et al.,
712   2020).

Despite the indirect evidence for an active heterotrophic bacterial population at the time of
sampling, it is possible that heterotrophic activity was also limited in the wintertime Southern
Ocean, in part because PON concentrations are generally low in this season (Pomeroy &
Wiebe, 2001; Smart et al., 2020). That said, bacteria may be more efficient at lower
temperatures than phyto- and zooplankton given their similar metabolic rates in temperate and
polar waters (Pomeroy & Wiebe, 2001 and references therein). Additionally, bacteria may be
less vulnerable to resource limitation because of their small size. Only slight differences in $Q_{10}$
values (i.e., the proportional increase in growth rate with a 10 °C rise in temperature) between
phytoplankton and heterotrophs are required for heterotrophic $NH_4^+$ production to exceed
phytoplankton $NH_4^+$ uptake (Koike et al., 1986). Nonetheless, it is highly unlikely that the
surface $NH_4^+$ pool measured in winter derived solely from wintertime bacterial production
given that yet higher $NH_4^+$ concentrations have been observed in late summer/autumn
(Becquevort et al., 2000; Dennett et al., 2001); this is discussed further in section 5.2 below.
*Heterotrophic activity by zooplankton* – The microzooplankton enumerated in this study may
also contribute to $NH_4^+$ accumulation, although they are probably less important in winter than
heterotrophic bacteria given their low and variable abundances (Fig. 6a). At the PFZ and AZ
stations characterized by high ratios of heterotrophic-to-photosynthetic cells but relatively low
absolute heterotrophic bacterial abundances, the coincident elevated $NH_4^+$ concentrations could
be due to the higher microzooplankton abundances compared to other stations (e.g., station
54.0°S). In other words, elevated microzooplankton abundances may help to explain the high
$NH_4^+$ concentrations at stations where the abundance of small heterotrophs was relatively low.
Above, we have assumed that the pathways leading to $NH_4^+$ production are associated with
heterotrophy. However, there are other possible mechanisms of $NH_4^+$ generation that should be
considered.
*DON cycling* – $NH_4^+$ can be released by heterotrophic bacteria that directly consume DON
(e.g., urea) (Billen, 1983; Tupas & Koike, 1990), and possibly also by ammonia oxidisers that
convert DON to $NH_4^+$ intracellularly, through the equilibration between intra- and extracellular





$NH_4^+$ pools (Kitzinger et al., 2019). DON can also be converted to $NH_4^+$ through
photodegradation by UV radiation (e.g., Aarnos et al., 2012). However, bacterial
decomposition of DON (rather than PON) to $NH_4^+$ is implicit in most estimates, qualitative and
quantitative, of heterotrophic bacterial remineralization. Additionally, the magnitude of cellular
$NH_4^+$ efflux by ammonia oxidisers is likely be extremely low given that they also require $NH_4^+$
to fix $CO_2$. Finally, the low light levels of the wintertime Southern Ocean mean that
photodegradation is unlikely to yield a significant $NH_4^+$ flux. We thus conclude that DON
conversion to $NH_4^+$, through any mechanism, is probably negligible.
*External inputs of ammonium* – High surface ocean $NH_4^+$ concentrations may theoretically
derive from external inputs of $NH_4^+$, such as from nitrogen fixation, $NH_4^+$ aerosol deposition, or
sea-ice melt. Nitrogen fixation should be negligible in the winter Southern Ocean due to the
extremely cold temperatures, low light and iron availability, and high $NO_3^-$ concentrations
(Jiang et al., 2018; Knapp et al., 2012; Kustka et al., 2003). Similarly, $NH_4^+$ aerosols are
unlikely to be abundant over regions of the Southern Ocean remote from islands and coastal
Antarctica. Those that are present mainly originate from surface ocean $NH_3$ efflux; once re-
deposited, this $NH_4^+$ does not constitute a new input term to surface waters (Altieri et al., 2021).
Additionally, $NH_4^+$ aerosol concentrations are at a minimum in winter (Legrand et al., 1998; Xu
et al., 2019). $NH_4^+$ deposition to the surface Southern Ocean is thus likely minimal. Finally,
since our sampling took place before the sea-ice reached its northernmost extent (Cavalieri &
Parkinson, 2008), the dominant process would have been sea-ice formation rather than sea-ice
melt, the latter a source of $NH_4^+$ at times (Kattner et al., 2004; Zhou et al., 2014), although
probably not during our study. Additionally, we observed elevated $NH_4^+$ as far north as 46ºS,
which is ~1700 km beyond the reach of sea-ice melt.
5.2 Seasonal cycling of $NH_4^+$ in the Southern Ocean mixed layer south of the SAF
To contextualize our wintertime observations, we need to explore the seasonality of the $NH_4^+$
pool in the surface Southern Ocean, especially given that $NH_4^+$ production in late summer and
autumn almost certainly contributes to wintertime $NH_4^+$ accumulation. Surface $NH_4^+$
concentrations were measured during three additional cruises in the Atlantic sector (December
2018-March 2019, early- and late summer; July-August 2019, winter; October-November
2019, spring; Fig. 9a-e). During these cruises, underway samples were collected for analysis of
$NH_4^+$ concentrations every two hours between Cape Town and Antarctica (early- and late
summer) or the MIZ (winter and spring), and analysed as described in section 3.2.1 for winter
772  2017.

In early summer, the surface $NH_4^+$ concentrations were uniformly low across the transect
(average of 0.11 ± 0.09 µM; Fig. 9a) due to rapid consumption by phytoplankton, as has been
observed previously (Mdutyana et al., 2020; Savoye et al., 2004; Daly et al., 2001). South of
the SAF, $NH_4^+$ concentrations increased significantly as the growing season progressed,
reaching an average concentration of 0.81 ± 0.92 µM by late summer (Fig. 9b). This $NH_4^+$
increase can be explained by elevated heterotrophic activity following the spring/summer
phytoplankton bloom (Mengesha et al., 1998; Le Moigne et al., 2013), coupled with iron-
and/or silicate-limitation of phytoplankton (Hiscock et al., 2003; Sosik & Olson, 2002) and
enhanced grazing pressure (Becquevort et al., 2000). The $NH_4^+$ concentrations measured south





of the SAF during the 2019 winter cruise (Fig. 9c) were similar to those observed in winter
2017 (0.48 ± 0.30 µM and 0.52 ± 0.11 µM, respectively), confirming that our 2017
observations are generally representative of the wintertime Southern Ocean. Additionally, the
winter measurements indicate that mixed-layer $NH_4^+$ concentrations remain high between late
summer and winter, consistent with sustained heterotrophic $NH_4^+$ production.
Our hypothesis for sustained late summer-to-winter heterotrophic activity is supported by
calculations of the residence time of $NH_4^+$ south of the SAF (Text S3). Using the $NH_4^+$
concentrations and $\rho NH_4^+$ measured in late summer 2019 (Deary, 2020), we calculate that the
$NH_4^+$ pool would be depleted in 2 to 27 days (median of 5 days) without coincident $NH_4^+$
production. Indeed, given the average $\rho NH_4^+$ south of the SAF in late summer (50.6 ± 24.0
nM/day), the net decline in $NH_4^+$ concentration of 0.33 ± 0.97 µM between late summer and
winter (a roughly four-month period) requires an average $NH_4^+$ production rate of 52.9 ± 25.0
nM/day. This estimate is comparable to $NH_4^+$ remineralisation rates measured in the AZ near
the Antarctic Peninsula in summer (the only measurements of $NH_4^+$ regeneration available for
the Southern Ocean; average of 55 nM day$^{-1}$; Goeyens et al., 1991).
By the early spring, the $NH_4^+$ concentrations south of the SAF had declined to near or below
the methodological detection limit (0.09 ± 0.08 µM; Fig. 9d), implicating increased
photosynthetic activity following the alleviation of light-limitation that results in the
consumption of nutrients introduced into surface waters in winter. We postulate that the
residual $NH_4^+$ would have been consumed prior to significant $NO_3^-$ drawdown because far less
energy (i.e., light) is required for its assimilation (Dortch, 1990). $NH_4^+$ concentrations south of
the SAF rose again by the late spring to an average value only slightly lower than that
measured in winter (0.37 ± 0.69 µM; Fig. 9e). However, late-spring $NH_4^+$ concentrations were
only elevated in the PFZ (range of 0.11 ± 0.01 to 4.39 ± 0.03 µM, average of 0.71 ± 1.04 µM),
as has been observed previously (Bathmann et al., 1997), which we attribute to increased
heterotrophic activity in response to elevated regional springtime phytoplankton growth driven
by frontal upwelling (Becquevort et al., 2000; Mayzaud et al., 2002). Excluding the PFZ data
yields a far lower late-spring average $NH_4^+$ concentration of 0.18 ± 0.14 µM, which we take as
broadly representative of this season.
Using our high-resolution $NH_4^+$ concentration measurements, we propose a seasonal cycle for
mixed-layer $NH_4^+$ south of the SAF (Fig. 9f). Our proposal is consistent with previous
characterizations of the early summer-to-autumn evolution of Southern Ocean $NH_4^+$
concentrations (i.e., from below detection due to phytoplankton uptake to elevated due to net
heterotrophic activity), but contradicts the hypothesis that $NH_4^+$ will subsequently decline due
to persistent but low rates of photosynthesis that yield insufficient biomass to support late-
summer heterotrophy, thus resulting in a coincident decrease in photosynthetic and
heterotrophic activity (Koike et al., 1986; Serebrennikova & Fanning, 2004). Instead, our data
evince a gradual decline in mixed-layer $NH_4^+$ concentrations from late summer through winter
that we attribute to heterotrophic $NH_4^+$ production outpacing $NH_4^+$ consumption in late
summer/autumn, with $NH_4^+$ regeneration then decreasing during winter to lower rates than the
combination of phytoplankton $NH_4^+$ consumption and $NH_4^+$ oxidation. By late spring, $NH_4^+$
reaches concentrations similar to those observed in early summer as the improved growing
conditions (i.e., elevated light and iron availability; Ellwood et al., 2008; Mtshali et al., 2019)



allow phytoplankton to rapidly consume any $NH_4^+$ remaining at the end of winter and
subsequently produced in spring. An exception to this scenario is elevated (and localized) $NH_4^+$
production near fronts, such as in late spring 2019, which likely resulted from biological
activity supported by frontal upwelling of silicate- and iron-bearing Upper Circumpolar Deep
Water (Prézelin et al., 2000).
5.3 Implications
*Potential for ammonium inhibition of nitrate uptake* – The low rates of $NO_3^-$ uptake
characteristic of winter Southern Ocean surface waters have been attributed to light,
temperature, and micronutrient (especially iron) limitation of phytoplankton growth (Martin et
al., 1990; Reay et al., 2001; Strzepek et al., 2019; Sunda & Huntsman, 1997). Wintertime $NO_3^-$
uptake may be further inhibited by the high $NH_4^+$ concentrations (Goeyens et al., 1995;
Philibert et al., 2015; Reay et al., 2001), as has been observed in other regions (Dortch, 1990;
Flynn et al., 2018). Previous Southern Ocean studies have identified an inhibitory effect of
$NH_4^+$ on $NO_3^-$ uptake at $NH_4^+$ concentrations >1 µM (and occasionally between 0.5 µM and 1
µM; Cochlan, 1986; Cochlan et al., 2002; Kristiansen & Farbrot, 1991; Reay et al., 2001). Such
concentrations were measured at a number of stations along our 2019 transects (Fig. 9b,c,e; and
in 2017 if inhibition occurs at $NH_4^+$ concentrations of 0.5 µM; Fig. 1). If the seasonal
accumulation of $NH_4^+$ inhibits $NO_3^-$ drawdown, this amounts to an inefficiency in the biological
pump. However, some culture studies report only a slight inhibition of $NO_3^-$ uptake, even at
high $NH_4^+$ concentrations (>>1 µM; Bagwell, 2009; Dortch, 1990 and references therein),
while others have detected no influence of $NH_4^+$ on $NO_3^-$ consumption (Rees et al., 1999),
suggesting that this effect is not straightforward. In winter 2017, we observed little evidence of
$NH_4^+$ inhibition of $NO_3^-$ uptake – for example, the southward decrease in $\rho NO_3^-$ was not
sharper than that of $\rho NH_4^+$ despite the increase in $NH_4^+$ concentration, and we observed no
relationship between $NH_4^+$ concentration and the proportion of $NO_3^-$-to-total N uptake (i.e., the
f-ratio, r = 0.28 including urea; n=7). We conclude that $NH_4^+$ inhibition of $NO_3^-$ uptake is
unlikely in open Southern Ocean surface waters, but may occur near fronts and/or the coasts of
islands and Antarctica where $NH_4^+$ can accumulate to concentrations >>1 µM (Henley et al.,
2017; Koike et al., 1986; Krell et al., 2005; Goeyens et al., 1995). In the case of coastal waters,
the damping effect of $NH_4^+$ inhibition on the biological pump is only relevant if the $NH_4^+$ being
consumed in lieu of $NO_3^-$ derives from *in situ* regeneration rather than being supplied from
land.
*Palaeoceanographic proxies* – $NH_4^+$ cycling in the Southern Ocean mixed layer may be
important for palaeoceanographic proxies (Smart et al., 2020; Robinson et al., 2020), such as
those that use the $\delta^{15}N$ of organic matter preserved in fossil foraminifer or diatom shells to
infer the extent of upper ocean $NO_3^-$ consumption in the past (and by extension, the role of
Southern Ocean biology in determining atmospheric $CO_2$; e.g., Martínez-García et al., 2014;
Studer et al., 2015). A recent ground-truthing study from the Southern Ocean showed that the
$\delta^{15}N$ of foraminifer-bound organic N tracks the $\delta^{15}N$ of PON rather than $NO_3^-$ (Smart et al.,
2020), in contrast to results from the low-latitude ocean (Ren et al., 2012; Smart et al., 2015).
Between summer and winter, the $\delta^{15}N$ of mixed-layer PON declines in the Southern Ocean
(particularly in the AZ) due to enhanced mixed-layer $NH_4^+$ cycling (Fig. 4c; Lourey et al.,
2003); this decrease will subsequently be reflected in the $\delta^{15}N$ of the foraminifera that feed on





PON (Smart et al., 2020) and the late summer/autumn diatom communities that consume
proportionally more $NH_4^+$ relative to $NO_3^-$ than in spring and early summer (Studer et al., 2015;
Kemeny et al., 2018). Thus, a decrease in the $\delta^{15}N$ of fossil foraminifera or diatoms could
reflect enhanced $NH_4^+$ consumption by the upper ocean ecosystem rather than a change in the
extent of $NO_3^-$ drawdown, although this will depend on the degree to which surface conditions
in the different seasons are communicated to the sediments (Smart et al., 2020). Further
clarifying the seasonal mixed-layer $NH_4^+$ cycle in the Southern Ocean may thus aid
interpretations of palaeoceanographic records.
*Ocean ammonia emissions* – The implications of $NH_4^+$ cycling extend beyond the upper ocean
to the atmosphere. Ammonium aerosols that influence Earth's albedo through scattering and
absorption of solar radiation and cloud formation (Tevlin & Murphy, 2019) are formed in the
marine boundary layer from reactions of $NH_3$ gas with acidic species, usually sulfur derived
from surface ocean dimethylsulfide emissions. The ocean is the largest natural source of $NH_3$
globally, however, the magnitude of the marine $NH_3$ source remains highly uncertain (Paulot et
al., 2015). Surface ocean $NH_4^+$ concentrations play a central role in determining the sign and
magnitude of the air-sea $NH_3$ flux, along with wind speed, surface ocean temperature, and pH.
Therefore, the biogeochemical pathways that drive seasonality in surface ocean $NH_4^+$
concentrations are an important control on the remote Southern Ocean air-sea $NH_3$ flux, with
implications for aerosol composition, cloud formation, and climate (Altieri et al., 2021).
**6.  Summary**
This study, conducted in the Southern Ocean during the infrequently-sampled winter season,
provides new insights into the internal cycling of N in the mixed layer of a globally-important
region. We used measurements of $NO_3^-$, $NH_4^+$, and urea uptake, $NH_4^+$ oxidation rates, $\delta^{15}N$-
PON, and the ratio of heterotrophic-to-photosynthetic cells to investigate $NH_4^+$ consumption,
and the ratios of POC:chl-a and POC:PON, the relationship of $V_{Ntot}$ to $V_C$, and measurements
of plankton community composition to evaluate the potential for heterotrophic $NH_4^+$
production. We attribute the elevated $NH_4^+$ concentrations that persist in the winter mixed layer
south of the SAF to sustained heterotrophic $NH_4^+$ production in excess of phytoplankton- and
nitrifier-mediated $NH_4^+$ consumption, driven by temperature-, light-, and possibly iron-
limitation of the $NH_4^+$ consumers. We further conclude that heterotrophic bacteria are the main
$NH_4^+$ producers in winter and that the contributions of DON degradation, nitrogen fixation,
aerosol deposition, and sea-ice melt to the Southern Ocean's mixed-layer $NH_4^+$ pool are
negligible. Future measurements of heterotrophic $NH_4^+$ production rates are required to validate
our conclusions, and higher spatial resolution sampling of community composition and N
consumption rates may help to explain smaller-scale variability in $NH_4^+$ concentrations,
particularly near the fronts.
From observations of surface $NH_4^+$ concentrations made between December 2018 and
November 2019, we suggest that the high-concentration $NH_4^+$ pool cannot be generated solely
during winter. Instead, we propose that $NH_4^+$ initially accumulates in late summer following the
peak phytoplankton growing season, after which sustained heterotrophy throughout the autumn
and winter prevents this $NH_4^+$ from being depleted until the early spring. The persistence of
elevated $NH_4^+$ concentrations across the polar Southern Ocean between late summer and winter



implies that the mixed layer is a biological source of $CO_2$ to the atmosphere for at least half the
year, not only because $NO_3^-$ drawdown is weak at this time (Arteaga et al., 2019; Johnson et al.,
2017), but also because the ambient conditions allow for $NH_4^+$ accumulation. Additionally,
high surface ocean $NH_4^+$ concentrations may alter components of the ocean-atmosphere $NH_x$
cycle and may have implications for palaeoceanographic reconstructions based on N isotope
measurements.
**Acknowledgements**
We acknowledge Captain Knowledge Bengu and the crew of the R/V *SA Agulhas II*, and Chief
Scientists Hermann Luyt, Marcello Vichi, and Thomas Ryan-Keogh. We thank Tahlia Henry
for CTD operations and CTD and SDS data processing. We are grateful to the students from
the Cape Peninsula University of Technology for help with sample collection and analysis of
chl-a samples. We thank Sedick Gallie for his assistance with sampling and for conducting the
microscope counts, and Raquel Flynn, Mishka Rawatlal, and Raymond Roman for assistance
with nutrient analyses. We acknowledge the Flow Cytometry Core Facility at the University of
Cape Town (UCT) and the efforts of Ian Newton at the Stable Light Isotope Laboratory (UCT).
This work was supported by the South African Departments of Forestry, Fisheries, and
Environment (formerly Environmental Affairs) and Science and Innovation (DSI), and the
National Research Foundation (NRF) through the South African National Antarctic Program
(SANAP; 110732 to K.E.A and 105539, 110735, and 129232 to S.E.F.), Equipment-related
Travel and Training Grant (118615 to K.E.A.), Competitive Support for Rated Researchers
Grant (111716 to K.E.A.), and Incentive Fund (115335 to S.E.F.). S.S., M.M., K.A.M.S., and
J.M.B. acknowledge funding from the NRF through postgraduate scholarships (120105,
112380, 113193, and 108757, respectively). S.S. was partially supported by a UCT Vice-
Chancellor Research Scholarship and M.M. by the UCT Harry Crossley Foundation Research
Fellowship. S.E.F. and K.E.A. acknowledge the support of the UCT Vice-Chancellor Future
Leaders 2030 programme. S.E.F. acknowledges an African Academy of Sciences/Royal
Society FLAIR fellowship and K.E.A. acknowledges support from UCT through a University
Research Council Launching Grant and a University Equipment Committee Grant. We further
acknowledge the support of the DSI Biogeochemistry Research Infrastructure Platform
(BIOGRIP).

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

**Figure and Table Captions**



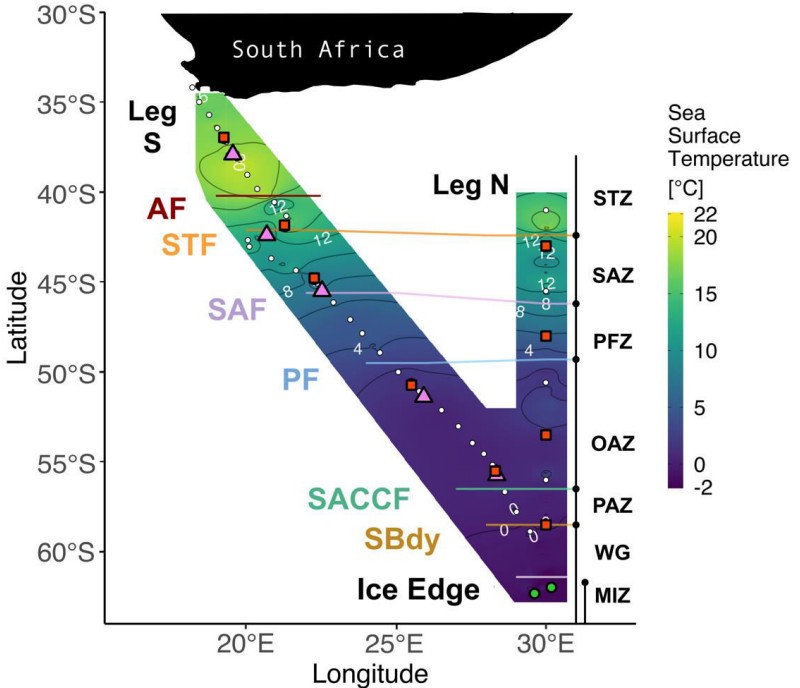


*Figure 1*: Winter 2017 cruise track overlaid on sea surface temperature (SST) measured by the hull-mounted thermosalinograph. The underway (Leg S) and CTD (Leg N) stations are indicated by white circles. Stations at which net primary production (NPP), nitrogen uptake, and ammonium oxidation experiments were conducted are denoted by red squares. The pink triangles indicate stations where only NPP experiments were conducted while the green circles show stations where only ammonium oxidation was measured. Solid lines indicate the positions of the fronts, identified using temperature and salinity, measurements. Abbreviations for fronts: AF – Agulhas Front (~40.2°S); STF – Subtropical Front (~42.1°S); SAF – Subantarctic Front (~45.6°S); PF – Polar Front (~49.5°S); SACCF – Southern Antarctic Circumpolar Current Front (~56.5°S); SBDY – Southern Boundary (~58.5°S). Abbreviations for zones: STZ – Subtropical Zone; SAZ – Subantarctic Zone; PFZ – Polar Frontal Zone; OAZ – Open Antarctic Zone; PAZ – Polar Antarctic Zone; WG – Weddell Gyre; MIZ – Marginal Ice Zone. Figure produced using the package ggplot2 (Wickham, 2016).

*Table 1*: Mean (± 1 SD) of surface ocean POC, PON, chl-a, and nutrient concentrations, cell abundances, and nutrient uptake rates measured in each zone of the Southern Ocean in winter 2017. Where no SD is given, only one sample was measured. ND – no data available. Abbreviations as in Figure 1.

1505





| | STZ | SAZ | PFZ | OAZ | PAZ |
|---|---|---|---|---|---|
| $NH_4^+$ (µM) | 0.08±0.03 | 0.06±0.01 | 0.42±0.01 | 0.52±0.01 | 0.58±0.01 |
| $PO_4^{3-}$ (µM) | 0.44±0.07 | 0.90±0.06 | 1.59±0.1 | 2.00±0.13 | 1.99±0.09 |
| $NO_3^-$ (µM) | 3.6±0.2 | 10.5±0.5 | 21.5±0.2 | 26.7±0.4 | 27.5±0.4 |
| $Si(OH)_4$ (µM) | 2.6±0.1 | 2.5±1.8 | 6.6±0.1 | 40.3±0.5 | 45.0±0.8 |
| $NO_2^-$ (µM) | 0.15±0.02 | 0.13±0.02 | 0.17±0.02 | 0.19±0.01 | 0.21±0.02 |
| Urea (µM) | 0.23±0.04 | 0.11±0.04 | 0.26±0.08 | 0.24 | 0.21±0.03 |
| chl-a (>0.3 µm) (µg $L^{-1}$) | 0.65±0.08 | 0.43±0.05 | 0.35±0.03 | 0.25±0.02 | 0.21±0.00 |
| chl-a (>2.7 µm) (µg $L^{-1}$) | 0.50±0.05 | 0.30±0.04 | 0.24±0.02 | 0.18±0.02 | 0.17±0.02 |
| chl-a (0.3-2.7 µm) (µg $L^{-1}$) | 0.15±0.1 | 0.13±0.07 | 0.11±0.04 | 0.06±0.03 | 0.04±0.02 |
| chl-a (% of total of >2.7 µm) | 77.5±13.9 | 73.1±10.9 | 69.8±8.7 | 76.7±11.3 | 80.1±8.5 |
| POC (>0.3 µm) (µM) | 4.38±6.67 | 3.4±0.43 | 3.23±0.26 | 3.43±0.48 | 3.47+0.22 |
| POC (>2.7 µm) (µM) | 2.63±0.51 | 2.59±0.43 | 1.87±1.22 | 1.92±0.36 | 4.64 |
| PON (>0.3 µm) (µM) | 0.61±0.19 | 0.48±0.08 | 0.44±0.08 | 0.50±0.14 | 0.54±0.09 |
| PON (>2.7 µm) (µM) | 0.26±0.06 | 0.28±0.06 | 0.24±0.28 | 0.22±0.12 | 0.38±0.02 |
| POC (% of total of >2.7µm) | 79.70±33.25 | 79.58±24.17 | 50.89±65.32 | 77.19±28.10 | 143.02 |
| PON (% of total of >2.7µm) | 69.04±43.26 | 67.12±29.47 | 53.8±121.41 | 66.98±49.74 | ND |
| POC:chl-a (g $g^{-1}$) | 103.0±22.1 | 102.5±14.4 | 122.5±11 | 234.1±29.2 | 219.3±1.0 |
| POC:PON (M/M) | 7.81±6.49 | 6.90±1.25 | 7.13±0.71 | 6.72±1.62 | 5.80±3.75 |
| $\delta^{15}$N-PON | 1.4±0.9 | 1.2±1.0 | 0.3±0.5 | -1.3±0.5 | -1.3±0.4 |
| NPP (>0.3 µm) (nM $day^{-1}$) | 497.1±42.4 | 277.5±21.3 | 289.7±19.2 | 85.3±26.1 | 27.7±0.2 |
| NPP (>2.7 µm) (nM $day^{-1}$) | 384.7±29.7 | 178.2±23.4 | 193.5 | 49.6±5.0 | ND |
| $\rho NH_4^+$ (>0.3 µm) (nM $day^{-1}$) | 5.7±0.8 | 8.9±1.1 | 12.9±0.4 | 4.8±0.1 | 3.0±0.8 |
| $\rho NH_4^+$ (>2.7 µm) (nM $day^{-1}$) | 4.0±1.1 | 4.1±1.2 | 4.2±4.7 | 3.1±0.4 | ND |
| $\rho NO_3^-$ (>0.3 µm) (nM $day^{-1}$) | 4.1±0.4 | 11.5±1.4 | 5.9±1 | 3.6±0.4 | 3.7±1.8 |
| $\rho NO_3^-$ (>2.7 µm) (nM $day^{-1}$) | 3.4±0.3 | 6.6±0.4 | 4.3±0.4 | 2.6±0.8 | 2.7±1.2 |
| $\rho$Urea (>0.3 µm) (nM $day^{-1}$) | 7.5±0.6 | 6.9±0.3 | 6.5±1.0 | 2.1±0.3 | 0.6±0.01 |
| $\rho$Urea (>2.7 µm) (nM $day^{-1}$) | 4.9±0.3 | 3.8±0.2 | 4.0±0.6 | 1.3±0.2 | 0.7±0.4 |
| $NH_4^+$ox (nM $day^{-1}$) | 9.3±0.5 | 12.9±0.6 | 11.1 | 17.7±0.6 | 14.3±1.0 |
| Total microplankton (cells $mL^{-1}$) | 13±11 | 5±3 | 9±3 | 6±6 | 4±2 |
| Centric diatoms (cells $mL^{-1}$) | <1 | <1 | <1 | <1 | 1±2 |
| Pennate diatoms (cells $mL^{-1}$) | 2±4 | <1 | 2±1 | 2±3 | <1 |
| Dinoflagellates (cells $mL^{-1}$) | 7±6 | 4±0 | 6±2 | 3±2 | 2±0 |
| Micro-zooplankton (cells $mL^{-1}$) | 4±3 | <1 | 2±2 | 1±2 | <1 |
| Nanoeukaryotes (cells $mL^{-1}$) | ND | 2.2±1.4 E+03 | 1.5±0.7 E+03 | 1.6±0.7 E+03 | 1.4E+03 |
| Picoeukaryotes (cells $mL^{-1}$) | ND | 4.5±2.9 E+03 | 4.9±3.7 E+03 | 1.5±0.5 E+03 | 8E+02 |
| Synechococcus (cells $mL^{-1}$) | ND | 3.8±1.8 E+03 | 2.3±1.1 E+03 | 1.4±0.2 E+03 | 1E+03 |
| Small heterotrophs (cells $mL^{-1}$) | ND | 4.5±3.2 E+03 | 2.3±1.2 E+03 | 2.1±2.3 E+03 | 3.2E+03 |
| Detritus (particles $mL^{-1}$) | ND | 38.2±14.9 E+03 | 63.8±42.9 E+03 | 25.7±18.6 E+03 | 2.57E+04 |
| $NH_4^+$ : $NO_2^-$ | 0.62±0.17 | 0.44±0.3 | 2.53±0.10 | 2.88±0.07 | 2.79±0.07 |

1506

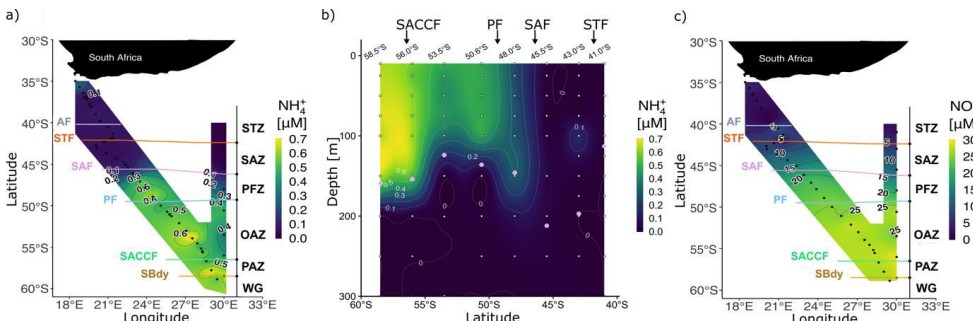

1507

*Figure 2*: Concentrations of dissolved ammonium (NH₄⁺) a) at the surface for Legs S and N and b) with depth for Leg N, and c) concentrations of nitrate (NO₃⁻) at the surface for Legs S and N. Pink circles in panel b show the mixed layer depth at each CTD station. Abbreviations as in Figure 1. Figure produced using the package ggplot2 (Wickham, 2016).

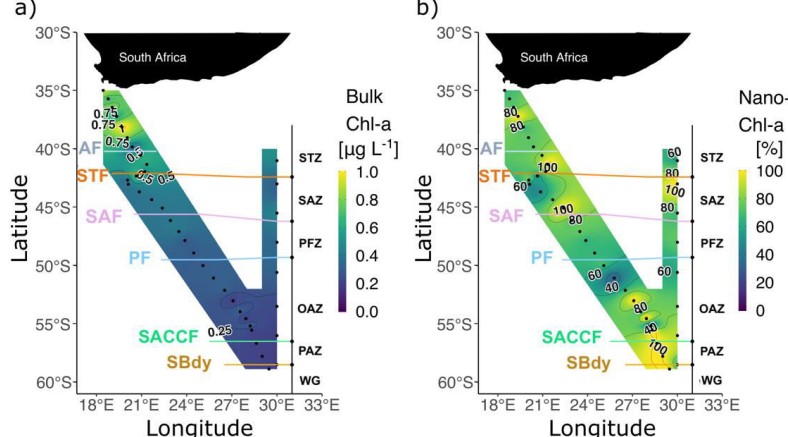


*Figure 3*: a) Bulk (>0.3 µm) chlorophyll-a (chl-a) concentrations and b) proportion of chlorophyll-a in the >2.7 µm size fraction (i.e., nanophytoplankton; % of total bulk chl-a) at the surface for Legs S and N. Abbreviations as in Figure 1. Figure produced using the package ggplot2 (Wickham, 2016).

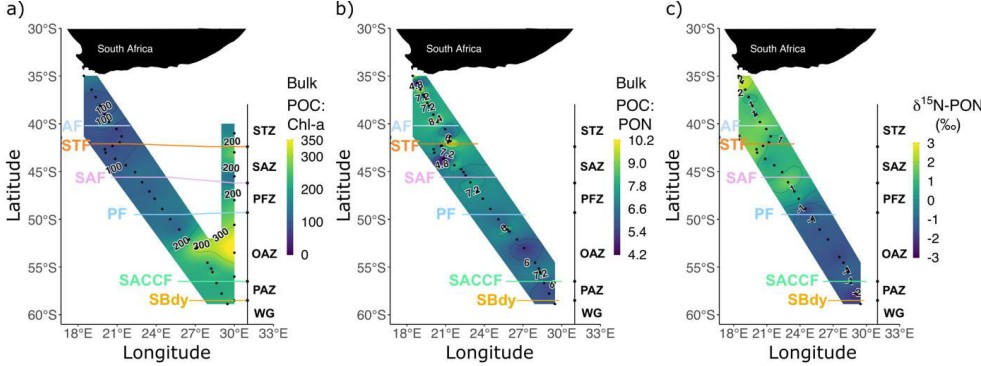


Biogeosciences Open Access
Discussions
EGU

*Figure 4*: a) Bulk (>0.3 μm) POC to chlorophyll-a ratio (weight:weight) at the surface for Legs S and N,
and b) bulk POC to PON (molar) ratio and c) $\delta^{15}$N-PON at the surface for Leg S. The stations nearest
South Africa at which biomass concentrations were extremely high have been excluded from panels b
and c. Abbreviations as in Figure 1. Figure produced using the package ggplot2 (Wickham, 2016).

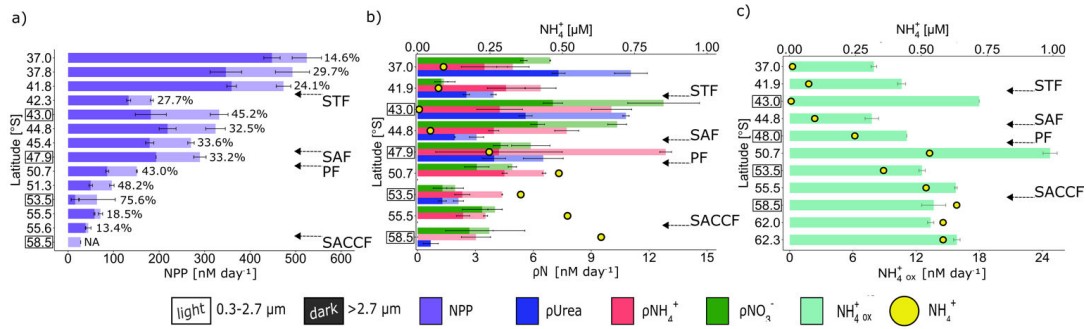


*Figure 5*: Surface rates of a) net primary production (NPP; ρC) for two plankton size fractions (>0.3 and
>2.7 μm); b) urea, ammonium (NH₄⁺), and nitrate (NO₃⁻) uptake for two plankton size fractions (>2.7
μm overlaid on >0.3 μm), and c) NH₄⁺ oxidation. Error bars indicate ±1 standard deviation of duplicate
experiments. The percentage of total NPP attributable to the 0.3-2.7 μm size fraction is written next to
each bar in panel a. NPP and NH₄⁺ uptake were not measured for the >2.7 μm size fraction at 58.5°S,
and urea uptake was not measured at 50.7°S and 55.5°S. On panels b and c, the surface NH₄⁺
concentration at each station is shown by the yellow circles. Leg N stations (i.e., at which samples were
collected from Niskin bottles fired at 10 m) are indicated by the open square around the station latitude.
Abbreviations as in Figure 1. Figure produced using the package ggplot2 (Wickham, 2016).

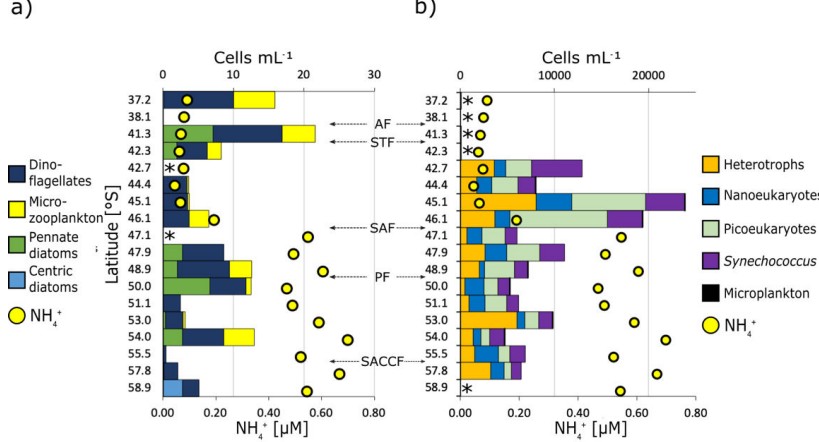


*Figure 6*: Surface community composition for a) plankton >5-10 μm (enumerated by microscopy) and
b) the total community <15 μm (enumerated by flow cytometry). The surface NH₄⁺ concentration at
each station is shown by the yellow circles. * indicates stations at which no measurements were made.
The abundance axis in panel b is $10^3$-times greater than the abundances shown in panel a. The fronts are
indicated on panel a with abbreviations as in Figure 1.

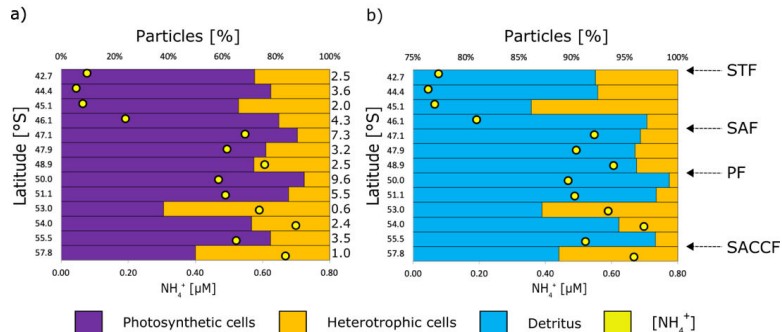

Figure 7: Relative abundances of a) total photosynthetic versus heterotrophic cells and b) detritus (DNA-negative) versus heterotrophic cells at the surface for Leg S. The surface $NH_4^+$ concentration is indicated by the yellow dots. The values shown on the right side of panel a are the heterotrophic-to-photosynthetic cell ratios. The upper x-axis in panel b begins at 75% in order to highlight the (much smaller) heterotrophic contribution to the summed detrital + heterotrophic particles. Abbreviations as in Figure 1.

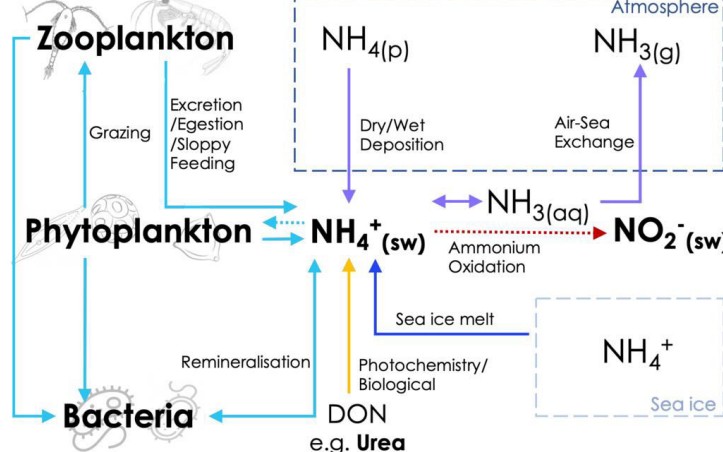

Figure 8: Schematic of the possible mixed-layer $NH_4^+$ consumption and production pathways. Bold text indicates components of the $NH_4^+$ cycle that were directly measured (seawater concentrations of $NH_4^+$, $NO_2^-$, and urea; phytoplankton and microzooplankton cell abundances) or inferred (bacterial $NH_4^+$ remineralization) in this study. Dotted lines indicate processes for which we have rate measurements (phytoplankton uptake of $NH_4^+$; oxidation of $NH_4^+$ to $NO_2^-$). Dashed-line boxes represent the atmosphere and sea-ice, with all other processes occurring in the ocean. DON – dissolved organic nitrogen; $NH_{3(aq)}$ – aqueous (seawater) ammonia; $NH_{4(p)}$ – ammonium aerosols (including ammonium sulphate, ammonium bisulphate, and ammonium nitrate); $NH_{3(g)}$ – ammonia gas.
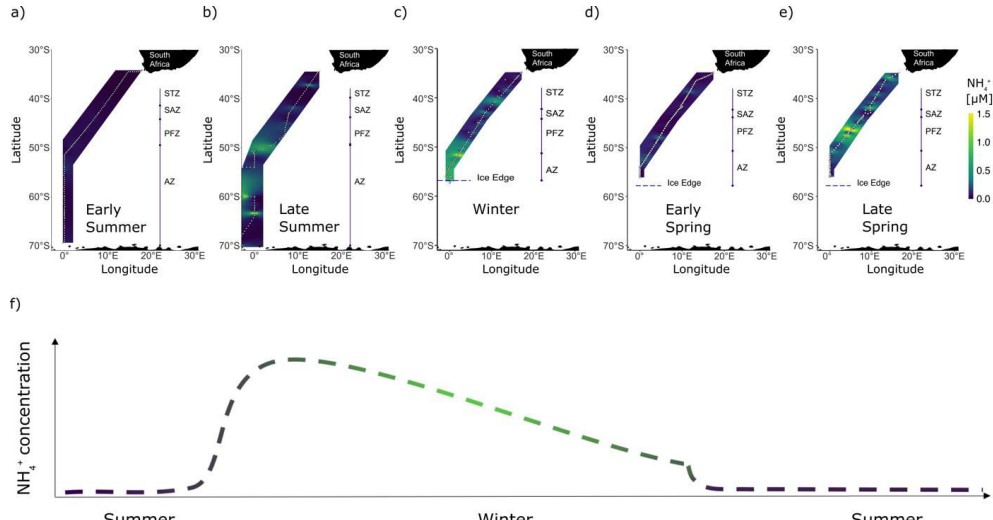

*Figure 9*: Surface concentrations of NH₄⁺ across the Atlantic sector of the Southern Ocean measured between December 2018 and November 2019. Five unique transects additional to the winter 2017 dataset are shown: a) early summer 2018, b) late summer 2019, c) winter 2019, d) early spring 2019, and e) late spring 2019. f) Proposed seasonal cycle of NH₄⁺ concentrations in the mixed layer for the waters south of the Subantarctic Front. The colour gradient in panel f indicates the transition period between winter and summer. Panels a and b cover a latitudinal extent of 30-70°S, while panels c-e cover 30-60°S due to the presence of sea-ice. Early- and late summer data were collected during the SANAE 58 Relief Voyage (6 December 2018 to 15 March 2019; VOY035); winter data were collected during the SCALE 2019 ([www.scale.org.za](www.scale.org.za)) winter cruise to the MIZ (18 July to 12 August 2019; VOY039); and spring data were collected during the SCALE 2019 spring cruise to the MIZ (12 October to 20 November 2019; VOY040). All sampling was conducted onboard the R/V *SA Agulhas II*. Abbreviations as in Figure 1, with AZ referring to the combined OAZ and PAZ. Figure produced using the package ggplot2 (Wickham, 2016).