# Peer review of "Biogeochemical controls on ammonium accumulation in the surface layer of the Southern"

_Biogeosciences, 2021_

## Author Response (AR1)

**Smith et al.** Biogeochemical controls on wintertime ammonium accumulation in the surface layer of the Southern Ocean

**Response to Reviewer's comments**

We thank the Reviewers for their comments, which will considerably improve our manuscript.

All three Reviewers acknowledged the importance of our observations from the understudied wintertime Southern Ocean and are supportive of the dataset and its interpretation being published once we have addressed their concerns. We will begin by responding to their major comments (many common to all three reviewers), and then address their more minor concerns.

**MAJOR COMMENTS**

1. **Lengthy introduction and discussion, including overly speculative discussion points**
   The Reviewers requested that we shorten the Introduction and Discussion sections, with a specific focus on reducing overly speculative content and removing sections that distract from the focus of the paper. In total, we have removed 836 words from the manuscript. Our edits are summarized below.

(i) Introduction: We have shortened the introductory text by 268 words.
Specifically, we have removed all content not directly related to the later discussion, for example, the following phrase at L99-100, since it is superfluous given what was stated previously –

> "…while larger phytoplankton such as diatoms that invest more energy in nutrient assimilation specialize in the assimilation of $NO_3^-$"

As suggested by Reviewer 1, we have shortened L107-152 (by 181 words) as below, removing all content that is not integral to the data we later present.

> We have removed the phrase at L107-110 mentioning implications and the two sentences at L118-123 which may detract from the purpose of the paragraph.
> We have made the sentence at L127-131 more succinct since the preceding phrase already states that several Southern Ocean studies found evidence of nitrification in the mixed layer.
> The sentence at L141-143 has been removed to avoid speculation.
> Similarly, we have removed the phrase at L147-149 since we state elsewhere that winter ammonium uptake rates are expected to be lower than those reported in summer, and the excised text could be read as contradicting that statement.

(ii) Discussion: We have revisited all the Discussion subsections to address the issues of length (in total, we have shortened the Discussion by 1611 words) and speculation.

Specifically, we have made a point of using language that does not imply suggestions as fact, especially those related to processes we did not measure (e.g., using "may", "probably", "likely"). However, since we observe high ammonium concentrations and low ammonium uptake rates in winter, ammonium must have been produced in excess of the rate at which it was consumed prior to our sampling. This notion is now supported by calculations of the residence time and production rates of ammonium, which were previously presented in the Supplement but are now incorporated into the main text (please see section 2 below). While we are confident that heterotrophy (mainly by bacteria) is the dominant ammonium production pathway, it is worth considering other possible ammonium sources (e.g., internal DON cycling or external inputs), particularly given that heterotrophy may be reduced under cold conditions when the supply of labile organic matter is limited. We nonetheless recognize that a lack of direct measurements introduces some uncertainty to this discussion. We thus rely on the literature to inform our expectations, which in some cases means describing observations from other ocean basins, although we have tried to keep the non-Southern Ocean references to a minimum. Invoking previous findings to fill gaps in a dataset is not unusual, and ignoring such information disregards a number of potentially relevant biogeochemical processes that have been identified by others as important to oceanic ammonium cycling. Therefore, we have decided to retain a (significantly more concise) version of the sections entitled "*DON cycling*" and "*External inputs of ammonium*". We have also amended the section titled "*Heterotrophic activity by zooplankton*" to ensure that we do not over-interpret our limited

dataset (see section 2 below). We have also entirely removed section 5.3 (*Implications*) from the Discussion (see (iv) below for details).

(iii) All three Reviewers were sceptical of our use of $NH_4^+:NO_2^-$ as a potential indicator of the inefficiency of $NH_4^+$ oxidizers at low temperatures. We have removed the paragraph at L654-666 (and Figure S6) in response to this feedback and for the following reasons. Firstly, the mere presence of $NH_4^+$ oxidisers in the Southern Ocean implies their adaptation to the polar environment. Secondly, the interpretation of $NH_4^+:NO_2^-$ is not straightforward, particularly given that these nutrients are intermediaries in N cycle processes other than oxidation (as pointed out by Reviewer 2). Lastly, the reason $NH_4^+$ oxidation rates are not higher is now discussed more succinctly (and hopefully, more convincingly) in the subsequent paragraph (L667-680), making the content at L654-666 unnecessary.

(iv) Implications: We have removed the implications section (formerly section 5.3) entirely as we agree with the Reviewers that much of the original text distracted from the paper's focus. Relatedly, we have changed section 6 from *Summary* to "*Summary and implications*" so that we can incorporate a much-shortened version of the implications text into our concluding remarks. Specifically, we feel it necessary to directly address the possibility of $NH_4^+$ inhibition of $NO_3^-$ uptake, as this phenomenon has been observed previously in the Southern Ocean and elsewhere (e.g., Cochlan et al., 2002; Flynn et al., 2018; Kristiansen & Farbrot, 1991; Reay et al., 2001) and is likely to occur to some of our readers. To that end, the following paragraphs were added to the end of section 6 (*Summary and implications*) –

> "The persistence of elevated $NH_4^+$ concentrations across the polar Southern Ocean between late summer and winter implies that the mixed layer is a biological source of $CO_2$ to the atmosphere for half the year, not only because $NO_3^-$ drawdown is weak at this time (e.g., Gibson & Trull, 1999; Gray et al., 2018; Hauck et al., 2015; Mongwe et al., 2018; Shadwick et al., 2015), but also because the ambient conditions allow for $NH_4^+$ accumulation. There are additional implications of our observations. For example, $NH_4^+$ concentrations >1 µM (and at times >0.5 µM) have been reported to inhibit $NO_3^-$ assimilation, including in the Southern Ocean (Cochlan, 1986; Goeyens et al., 1995; Philibert et al., 2015; Reay et al., 2001). Inhibition of $NO_3^-$ assimilation due to the seasonal accumulation of $NH_4^+$ would constitute an inefficiency in the biological pump. However, we observed little evidence of this effect in winter 2017 – the southward decrease in $\rho NO_3^-$ was not stronger than that of $\rho NH_4^+$ despite the latitudinal increase in $NH_4^+$ concentration, and we observed no relationship between $NH_4^+$ concentration and the proportion of $NO_3^-$ to $NO_3^-+NH_4^+$ uptake (i.e., the f-ratio; Table S1).

> The implications of $NH_4^+$ cycling extend beyond the upper ocean to the atmosphere, since ammonium aerosols that influence Earth's albedo (Tevlin & Murphy, 2019) are formed in the marine boundary layer from reactions of $NH_3$ gas with acidic species. In the remote Southern Ocean, marine $NH_3$ emissions, which are the largest natural contributors to $NH_3$ globally, are likely the dominant local source of $NH_3$ to the atmosphere (Paulot et al., 2015). Surface ocean $NH_4^+$ concentrations play a central role in determining the sign and magnitude of the air-sea $NH_3$ flux, along with wind speed, surface ocean temperature, and pH. Therefore, the biogeochemical pathways that underpin seasonal changes in surface ocean $NH_4^+$ concentrations represent an important control on the remote Southern Ocean air-sea $NH_3$ flux, with consequences for aerosol composition, cloud formation, and climate (Altieri et al., 2021)."

**2. Over-reaching conclusions made from the data regarding heterotrophy**

We recognize that our lack of direct measurements of heterotrophy means that any quantitative evaluation of $NH_4^+$ production must be undertaken with caution. In response to the Reviewers' comments, we have significantly reduced our discussion of heterotrophy, retaining only the text that is well-supported by either our data or the literature. We have also taken care in our choice of language so as not to over-interpret our data or over-reach in the conclusions that we draw from

them. Specifically –

We have now fully incorporated Text S3 into the revised manuscript (see our general response to Reviewer 3) – here we present estimates of the rate of heterotrophic $NH_4^+$ production in winter and late summer, calculated using *in situ* measurements of $NH_4^+$ concentrations and uptake rates from the 2017 winter and 2019 summer cruises. We have some confidence in these rates since the late summer estimate is in agreement with the only *in situ* $NH_4^+$ production rates measured to-date in the Southern Ocean ($52.9 \pm 25.0$ nM day$^{-1}$ in our study vs 55 nM day$^{-1}$ near the Antarctic Peninsula in summer; Goeyens et al., 1991). Additionally, our estimates of $NH_4^+$ production at stations south of the SAF in winter 2017 were ~2 nM day$^{-1}$ higher than the corresponding $NH_4^+$ uptake rates ($23.4 \pm 6.6$ nM day$^{-1}$ vs $21.4 \pm 0.6$ nM day$^{-1}$), thus providing evidence for a substantial, but declining, heterotrophic contribution to the $NH_4^+$ pool in autumn and winter.

We have removed the use of POC:PON and POC:chl-a from our analysis of heterotrophy (and the related Fig. 4a and 4b) since both of these metrics can be influenced by other factors, such as adaptations to low temperatures, light levels, and iron concentrations (Eppley, 1972; Greene et al., 1991; Mongin et al., 2006; Talmy et al., 2016). $\delta^{15}$N-PON, which was included as Fig. 4c, is now shown in a single-panelled Figure 4.

We have decided to keep an updated (and shortened) version of the discussion of the heterotrophic bacterial and detrital abundance measurements as a means of qualitatively evaluating the potential for heterotrophy. To ensure that our intention here is clear to the reader, we have added the following caveat to the Methods section:

> "In this study, we did not directly measure $NH_4^+$ regeneration (i.e., heterotrophy). Instead, we use the abundance of heterotrophic bacteria as a qualitative indicator of $NH_4^+$ regeneration potential, recognizing that cell abundance does not imply activity. Additionally, we estimate the rate of $NH_4^+$ production from our concentration and rate data (see section 3.3). The availability of organic matter to heterotrophs is inferred from the abundance of detritus."

Additionally, we have altered our approach to addressing the topic of heterotrophy in general, given the limitations of our dataset and the concerns of the Reviewers. The text regarding the activity of heterotrophic bacteria has been reduced to two short paragraphs, and that referring to zooplankton has been reduced to one. Below is the text included in the amended version of the manuscript. We present it here in full because the issue of heterotrophy was arguably the biggest sticking point for our Reviewers. Please note our use of less assertive language (indicated by the text in **bold**) and our stronger reliance on the data we have in hand (underlined).

> "*Heterotrophic activity by bacteria* – Heterotrophic bacteria contribute significantly to $NH_4^+$ production in the Southern Ocean (Hewes et al., 1985; Koike et al., 1986; Tréguer & Jacques, 1992), including in winter (Rembauville et al., 2017). In our dataset, lower ratios of photosynthetic-to-heterotrophic cells were observed at stations with higher $NH_4^+$ concentrations (e.g., stations 48.9°S, 53.0°S, 54.0°S, and 57.8°S; Fig.7a), consistent with a role for the heterotrophic bacteria present at the time of sampling in generating the ambient $NH_4^+$ pool. The **potential for ongoing heterotrophic activity** can also be inferred from the high detrital particle counts along the transect (Fig. 7b). However, since heterotrophic bacteria are likely more active in late summer and autumn when the temperature and the supply of labile PON are higher (Becquevort et al., 2000; Dennett et al., 2001; Pomeroy & Wiebe, 2001; Smart et al., 2020), we expect that the winter $NH_4^+$ pool includes $NH_4^+$ produced in late summer and autumn. A further consideration is assimilation of $NH_4^+$ by heterotrophic bacteria, reported to occur at elevated rates in the Southern Ocean mixed layer in winter (Mdutyana et al., 2020; Text S3). If this process is a persistent feature of the winter Southern Ocean, it will decrease the net contribution of heterotrophic bacteria to $NH_4^+$ accumulation. We conclude that it is unlikely that the surface $NH_4^+$ pool measured in winter derived solely from wintertime bacterial $NH_4^+$ production given that yet higher $NH_4^+$ concentrations

have been observed in late summer and autumn (Becquevort et al., 2000; Dennett et al., 2001), including in the present study (see section 5.2 below).

*Heterotrophic activity by zooplankton* – While the microzooplankton enumerated in this study occurred at very low abundances, those that were present **likely** contributed to the $NH_4^+$ flux. For example, at stations 48.9°S and 54.0°S in the PFZ and AZ, respectively, both the ratios of photosynthetic-to-heterotrophic cells and the absolute abundances of heterotrophic bacteria were low, while the microzooplankton abundances and $NH_4^+$ concentrations were elevated compared to nearby stations. The implication of these observations is that elevated microzooplankton abundances **may** help to explain high $NH_4^+$ concentrations in waters with low numbers of heterotrophic bacteria, **although we note that this scenario only occurred at two stations**. On balance, we posit that microzooplankton are less important for wintertime $NH_4^+$ production than heterotrophic bacteria given their low abundances in the surface layer (Fig. 6a; Atkinson et al., 2012)."

**3. The inclusion of the seasonal dataset in the Discussion**

We realize that it is unconventional to incorporate a new dataset into the Discussion section of a manuscript. Given how critical the data from the 2018-2019 annual cycle are to our Discussion, and how valuable they will be to the community, we have decided to keep the dataset in the paper but to incorporate it fully into all relevant sections. As such, the sample set is first introduced at the end of the Introduction:

"Here, we focus mainly on $NH_4^+$ cycling in the Southern Ocean mixed layer in winter, a season assumed to be largely biologically dormant (Arrigo et al., 2008; Schaafsma et al., 2018) and for which $NH_4^+$ cycle data are scarce. We confirm that $NH_4^+$ accumulates throughout the winter mixed layer south of the SAF, and examine the potential drivers thereof. Using $NH_4^+$ concentration data collected over a full annual cycle, we propose that these drivers include a contribution from the residual late-summer $NH_4^+$ pool, sustained $NH_4^+$ production in the autumn and winter, and limited wintertime $NH_4^+$ uptake and oxidation that nonetheless exceed the rate of in situ $NH_4^+$ production. Finally, from our temporally-resolved $NH_4^+$ concentration data, we propose – for the first time – a measurement-based seasonal cycle for the mixed-layer $NH_4^+$ pool south of the SAF."

We describe the dataset in section 3.1 of the Methods (*Cruise tracks and sample collection*), with a new supplemental figure (Fig. S1) showing the cruise tracks and sampling locations (as in Figure 1 of the main text for winter 2017) –

"Samples were collected for a series of analyses on the southward (S) and northward (N) legs of a winter cruise between Cape Town, South Africa, and the marginal ice zone (MIZ) onboard the R/V *SA Agulhas II* (VOY25; 28 June to 13 July 2017) (Fig. 1). Samples were also collected for $NH_4^+$ concentration analysis on three cruises onboard the R/V *SA Agulhas II* during 2018/19: early- and late summer samples were collected during the SANAE 58 Relief Voyage (6 December 2018 to 15 March 2019; VOY035); winter samples were collected during the SCALE 2019 (www.scale.org.za) winter cruise to the MIZ (18 July to 12 August 2019; VOY039); and spring samples were collected during the SCALE 2019 spring cruise to the MIZ (12 October to 20 November 2019; VOY040) (Fig. S1)."

In section 3.2.1. of the Methods (*Ammonium concentrations*), we note that "On all cruises, $NH_4^+$ concentrations were measured shipboard using the fluorometric method of Holmes et al. (1999) and a Turner Designs Trilogy fluorometer…"

We have moved the figures showing the seasonal dataset (previously Figure 9) to the Results

section (renumbered as Figure 8) and have added the following Results text –

> " 4.6 2018/19 cruises: ammonium concentrations
> In early summer, surface $NH_4^+$ concentrations were uniformly low across the transect (average of $0.11 \pm 0.09$ µM; Fig. 8a). South of the SAF, $NH_4^+$ increased to an average concentration of $0.81 \pm 0.92$ µM by late summer (Fig. 8b). By winter 2019, the $NH_4^+$ concentrations south of the SAF were ~40% lower than they had been in late summer (Fig. 8c), and were similar to those observed in winter 2017 ($0.50 \pm 0.30$ µM and $0.52 \pm 0.11$ µM, respectively), confirming that our 2017 observations are generally representative of the wintertime Southern Ocean. By early spring, the $NH_4^+$ concentrations south of the SAF had declined to near or below detection ($0.09 \pm 0.08$ µM; Fig. 8d) before rising again by late spring to an average value only slightly lower than that measured in winter ($0.40 \pm 0.74$ µM; Fig. 8e). However, the late-spring $NH_4^+$ concentrations were only elevated in the PFZ (range of $0.11 \pm 0.01$ to $4.39 \pm 0.03$ µM, average of $0.77 \pm 1.11$ µM), as has been observed previously (Bathmann et al., 1997). Excluding the PFZ data yields a far lower late-spring average of $0.17 \pm 0.11$ µM south of the SAF, which we take as more broadly representative of this season."

The Discussion section 5.2 (*Seasonal cycling of $NH_4^+$ in the Southern Ocean mixed layer south of the SAF*) has been updated (with all the 2018/19 data methods- and results-related text removed), although the sense of it remains the same. This section now begins with the following amended sentence to further justify the inclusion of these data –

> "The $NH_4^+$ concentration data collected over the 2018/19 annual cycle provide context for interpreting our winter 2017 dataset, allowing us to address our hypothesis that $NH_4^+$ production in late summer and autumn contributes to the elevated $NH_4^+$ concentrations measured in winter."

**MINOR COMMENTS**
Below, we respond to each of the Reviewers' smaller suggestions and comments in turn and outline the changes we will make to the manuscript to address them. The Reviewer comments are in black text and our responses are in blue.

**Reviewer 1: Anonymous**

Referee comment on "Biogeochemical controls on wintertime ammonium accumulation in the surface layer of the Southern Ocean" by Shantelle Smith et al., Biogeosciences Discuss., https://doi.org/10.5194/bg-2021-149-RC1, 2021

**General comments**

The authors present an interesting and thought-provoking study on the dynamics of ammonium cycling in the Southern Ocean. There are some nice new observations reported here building upon previous work by the UCT group, with the authors presenting a novel scenario for the seasonal accumulation and utilization of ammonium in surface waters. The paper however can be made more succinct, particularly in the discussion, which at times speculates far beyond what can be demonstrated from the presented data or confidently gleaned from the literature.

Response: Please see our responses (1-3) above. In the amended version of the manuscript, we have made a concerted effort to shorten and made more concise the Introduction and Discussion text, in the ways we have indicated above. In particular, we have ensured that the focus of the study remains clear throughout the manuscript and have endeavoured to remove speculation (please see in particular our response (2) above regarding our discussion of heterotrophy).

**Specific comments**

**Smith et al.** Biogeochemical controls on wintertime ammonium accumulation in the surface layer of the Southern Ocean

**Introduction:**

Nicely sets the premise of the study but could be shortened and modified to avoid overly descriptive presentation of key concepts.
Response: Please see our response (1) above. We have revisited the introduction (shortening it by 268 words), ensuring the topics included are relevant to the new data presented.

Are L107-123 really required in the intro?
Response: Since new $\delta^{15}$N-PON data are presented in support of our argument, we feel that some of this text is relevant. Nonetheless, we agree that it can be made more succinct, and have thus amended it as follows –

> "In addition to the implications for size distribution, the dominant N source to phytoplankton is indicative of their potential for $CO_2$ removal, as per the new production paradigm (Dugdale & Goering, 1967). The N isotopic composition ($\delta^{15}$N, in ‰ vs. $N_2$ in air, = ($^{15}$N/$^{14}$N$_{sample}$/$^{15}$N/$^{14}$N$_{air}$ – 1) x 1000) of particulate organic N (PON; a proxy for phytoplankton biomass) can be used to infer the dominant N source to phytoplankton (Altabet, 1988; Fawcett et al., 2011; Lourey et al., 2003; Van Oostende et al., 2017) since the assimilation of subsurface $NO_3^-$ yields PON that is higher in $\delta^{15}$N than that fuelled by recycled $NH_4^+$ uptake (Treibergs et al., 2014). As such, measurements of bulk $\delta^{15}$N-PON can be used to infer the net N uptake regime (Altabet, 1988; Fawcett et al., 2011; 2014; Lourey et al., 2003)."

L107-152 could perhaps be more succinct but do clearly present the working premise behind the study
Response: We have shortened this part of the Introduction by 12 lines and made the remaining text more succinct. Nonetheless, we feel it essential to retain mention of the ideas presented here as they provide the motivation for the study and outline some of the important implications.

**Methods:**

Some care and clarification needed in presentation of methodological details.

L177-178: Can you add approximate irradiance depths here?
Response: We have made the following edit –
> "…while samples on leg N were collected from surface (~10 m, approximately 55% light depth) Niskin bottles…".

L184-185: How many replicate samples were collected?
Response: We have added this information to the methods. Samples were collected in duplicate for all nutrients (including $NH_4^+$) except urea where only one sample was collected due to water budget constraints.

L204: Please quantify 13C addition and (presumably) your working assumption of ambient DIC concentration.
Response: DIC concentrations were measured *in situ* by the Council for Scientific Research (unpublished results) using a VINDTA 3C instrument (range of measured DIC concentrations = 2017 to 2130 µmol/L). The concentration of $^{13}$C-DIC tracer in the 2L bottles was 100 µmol/L. We have clarified both of these points in the amended version of the manuscript.

L213: Were (replicate) 2L bottles used as for NPP incubations? If not please add description
Response: We have added this information to the Methods section. Replicate 1 L bottles were used for the N uptake experiments.

Section 3.2.1 – Please clarify the method, particularly the time period samples were left frozen for prior to analysis.

**Smith et al.** Biogeochemical controls on wintertime ammonium accumulation in the surface layer of the Southern Ocean

Response: We have added the following information to the Methods section –
> "…samples were frozen immediately upon collection, for a maximum of 24 hours.".

L236: Clarify if the same sample was measured 3 times or whether you measured 3 replicates once each.

Response: We have added this information to the Methods section – "…then each replicate was analysed in triplicate". In other words, duplicate samples were measured three times each.

L233: Please clarify procedures. How long were samples frozen?/ How long after sampling were samples analysed? What impact could this have had on NH₄ concentrations? Freezing NH$_4^+$ samples is not ideal as it may lead to problems. See Degobbis (1973) (On the storage of seawater samples for ammonia determination)

Response: Samples were frozen immediately after sampling and remained frozen for a maximum of 24 hours – according to Degobbis (1973), this approach would have resulted in a minimal effect on the ammonium concentrations. Additionally, samples were collected in duplicate, and the pooled standard deviation was ±0.02 µM, indicating very good agreement between replicate collections, which would be unlikely if samples had been altered post-collection. Finally, we note that we spoke with Malcolm Woodward, the Head of the Nutrient Facility at Plymouth Marine Lab and co-Chair of SCOR working group 147: *Towards comparability of global nutrient data*, and he advised us to take the approach with regards to ammonium sampling and analysis that we have outlined in the manuscript.

Uptake rates – Based on L205/216 the incubations for N uptake were <6 hrs in length.
However, by converting the short duration incubation length to day fraction in equation 1 the authors are extrapolating their results. Implicit in this approach is extrapolation to 24 hours from <6 hourly uptake expts. How confident are the authors that the measured rates of NPP and N uptake remain constant over the day/night cycle? Is it realistic to assume NPP or N uptake continues at the same rate around the clock? How could diel variations in N uptake impact the results? Could this extrapolation be one reason why the NPP rates are higher than reported previously (L397-399?)

Response: The 'per hour' rates were not extrapolated to 24 hours in the calculations – the 'per hour' rates were scaled up to 'per day' rates by multiplying them by the number of daylight hours (calculated using day of the year and latitude). We have added the following clarifying sentence to the methods –
> "Daily rates were computed by multiplying the hourly rates by the number of daylight hours, the latter calculated using the sampling latitude and day of the year (Forsythe et al., 1995)."

L682-683: The authors conclude that NH4 production must be high, which in turn indicates that the NH4 uptake rates are likely biased by isotope dilution within the incubation and that final rates are thus low. This is not corrected for (nor apparently can it be given the experimental work undertaken) but needs to be formally acknowledged somewhere as it is a significant caveat for the overall discussion.

Response: As noted by the reviewer, we are unable to correct for a dilution effect given our experimental approach – i.e., we did not collect subsamples at multiple timepoints during the period of incubation and we also did not measure coincident rates of NH$_4^+$ regeneration. However, our experiments were short (3 to 7.5 hours) and the additions of $^{15}$NH$_4^+$ were high (100 nM) relative to both the ambient NH$_4^+$ concentrations and the K$_m$ for NH$_4^+$ uptake (Mdutyana, 2021), making a significant dilution effect less likely (Lipschultz, 2008). Additionally, we conclude in the manuscript that elevated rates of NH$_4^+$ production likely occurred prior to our sampling (i.e., in late-summer and autumn), with NH$_4^+$ assimilation outpacing NH$_4^+$ production in winter (see section 5.2 and Figure 8 of the modified main text). Nonetheless, we have added the following caveat to the manuscript:

> "We note that isotope dilution (i.e., the dilution of $^{15}$NH$_4^+$ by co-occurring $^{14}$NH$_4^+$ regeneration) during the NH$_4^+$ uptake and oxidation experiments could potentially lead to an underestimation of the rates (Glibert et al., 1982; Mdutyana, 2021). For the NH$_4^+$ uptake experiments, their short duration (3 to 7.5 hours) would have rendered the effect of regeneration minor (Mdutyana et al., 2020). Moreover, the $^{15}$NH$_4^+$ additions were

high (100 nM) relative to both the ambient $NH_4^+$ concentrations north of the SAF and the $K_m$ values derived for $NH_4^+$ uptake and oxidation in the winter Southern Ocean (150-405 nM and 28-137 nM, respectively; Mdutyana, 2021), making a significant dilution effect unlikely (Lipschultz, 2008). Finally, at the stations south of the SAF, the ambient $NH_4^+$ concentrations were so high that even if the regeneration of $^{14}NH_4^+$ occurred at an elevated rate (e.g., 50 nM day$^{-1}$; as has been measured in the late-summer Southern Ocean when remineralization is expected to be high; Goeyens et al., 1991), the $^{15}N/^{14}N$ of the $NH_4^+$ pool would decrease by <1-2%. We thus consider the potential effect of isotope dilution to be minor.

A further consideration is possible stimulation of the $NH_4^+$ uptake and oxidation rates by $^{15}NH_4^+$ addition (Lipschultz, 2008). Given the $K_m$ values listed above and the high ambient $NH_4^+$ concentrations measured in the PFZ and AZ, a stimulation effect could only be significant at the stations north of the SAF where the $NH_4^+$ concentrations were 10-100 nM, and even then, to a lesser extent for $NH_4^+$ oxidation than $NH_4^+$ uptake given that ammonia oxidizers in the winter Southern Ocean become saturated at $NH_4^+$ concentrations of 100-200 nM (Mdutyana, 2021). The rates reported for the stations north of the SAF should therefore be considered "potential rates." However, since our focus is mainly on explaining the accumulation of $NH_4^+$ south of the SAF, having "potential" rather than "true" rates for the STZ and SAZ does not affect our conclusions."

Units: Throughout the paper the authors report uptake rates as nM d-1. One assumes shorthand for nmol N L-1 d-1, but please clarify volumetric basis somewhere in methods section.
Response: This has been clarified in the methods. The units are indeed shortened from nmol N L$^{-1}$ day$^{-1}$ to nM day$^{-1}$.

L286: Why not also present specific uptake rates for all nutrients individually?
Response: We used specific uptake rates only to compare the biomass-normalised rates of inorganic carbon fixation and total nitrogen uptake (i.e., Figure S7). In general, comparisons between stations are made using ρC or ρN (i.e., the transport rate) as this metric describes the ecosystem-level response, which is our primary interest here. Additionally, we are reluctant to add more data and text to an already lengthy manuscript. We have thus removed all mention of the specific uptake rates from the main text and only present them in the Supplement in support of possible heterotrophic $NH_4^+$ assimilation (a minor point that we raise briefly in section 5.1.2 of the Discussion).

Eq3: Reliance upon specific uptake rates to calculate the f-ratio (rather than actual uptake rates) can lead to errors and/or inconsistencies with other studies. F-ratio values are very occasionally stated in the text and not in Table 1. Please include an indication in table 1
Response: We thank the Reviewer for pointing out this inconsistency. Equation 3 was written incorrectly in the original version of the manuscript – we did use the actual uptake rates (in nM day$^{-1}$) to calculate the f-ratios even though Equation 3 implied otherwise. This error has now been corrected. Additionally, the f-ratio estimates have been included in Table 1.

L294: An error of 4-8% when urea is included/excluded seems very low. Usually this would be higher – see Wafar et al 1995 (Wafar, M.V.M., P. Lecorre and S. L'helguen (1995). f-ratios calculated with and without urea uptake in nitrogen uptake by phytoplankton. Deep-Sea Research Part I 42(9), 1669-1674.)
Response: We thank the Reviewer for this comment. There was an error in our calculation – the range should be 8-25%. This error has been corrected in the text.

L300: Correct/clarify oxidation rate units (i.e. nM d-1 = nmol N L-1 d-1)
Response: This has been clarified as above.

L306: As above, does extrapolation from short term incubations to daily rates have an implication for the results? I.e any diel variability to consider?

**Smith et al.** Biogeochemical controls on wintertime ammonium accumulation in the surface layer of the Southern Ocean

Response: The ammonia oxidation experiments were conducted for 24 hours, as stated in the Methods ("NH$_4^+$ oxidation bottles were incubated for 24 hours under the same temperature conditions as the N uptake and NPP experiments."), and as such, directly yield daily rates.

L311: Is a 20 ml subsample a sufficiently large enough volume to accurately enumerate from? Why not settle and analyse the whole 250 ml sample?
Response: We settled 20 mL for practical reasons given the equipment and time that we had available. However, for each sample, at least 100 cells were counted to ensure a statistically valid estimate. This information has been included in the Methods text.

L353: Any difference in the SST gradient between Leg S and Leg N?

Response: We found similar gradients in SST along legs S and N – please see Figure R1 below showing this result (purple = leg S; green = leg N). We have added the following phrase to the manuscript at L354 in response to this comment – "…with similar gradients measured for legs S and N…".

[Figure]

*Figure R1*: Sea surface temperature (SST) along the transect during legs S (purple) and N (green). The linear regression between 40°S and 52°S for each leg was used to predict the SST every 0.2° latitude – the predicted values and 95% confidence intervals are shown for each leg as dashed lines and shaded bands, respectively. The linear equation used, the adjusted r$^2$, and the associated p-values are shown for reference.

**Results:**

Generally clear and understandable but the section does have a tendency to segue into discussion. Present only the facts, leave the discussion to the Discussion section.
Response: Discussion text has been removed, except for the occasional comparison with previous studies, which we view as evincing the reliability of our results rather than constituting an interpretation/discussion thereof. In total, we have shortened the Results text by 85 words (before the addition of sections 4.6 and 4.7 that show the seasonal data and production rate calculations) and removed seven references.

**Discussion:**

The discussion is extremely long and should be shortened. Large parts of the discussion read like a

literature review, presented solely for the purpose of supporting the seasonal scenario outlined by the authors and not necessarily to contextualise the observations presented here. Fundamentally, this paper presents NH4 uptake and NH4 oxidation rates from a single cruise, whereas the discussion incorporates additional cruise data and relies heavily upon literature observations to establish a new, somewhat speculative, view of NH4 cycling in the southern ocean, many aspects of which are not supported by data but only by assumption of their significance. Some sections e.g. section 5.3 can probably be deleted in their entirety as they go way beyond a discussion of the data presented here and stray into unnecessary speculation. The discussion section could probably be a standalone opinion paper or the premise for a proposal in its own right. Much is unsubstantiated (though not necessarily wrong) and some references used derive from rather different subtropical environments where the relevance of NH4 and urea can differ markedly.

Response: Please see the general response (1 and 3) to the Reviewers above. In brief, we have 1) shortened the Discussion significantly (1611 words in total), removing 42, 94, and 39 words from the paragraphs starting at L470, L596, and L621, respectively, removing the paragraph starting at L654 (195 words), shortening the section on heterotrophic bacteria by 259 words, and shortening section 5.2 by 251 words (by integrating the topic into other sections of the manuscript); 2) removed the *Implications* section 5.3. entirely; 3) fully integrated the seasonal dataset and Text S3 into all sections of the manuscript; and 4) made the entire Discussion more concise and removed speculation.

L520: Based on Fig 4c alone it is not possible to say how long d15N conditions lasted

Response: The Reviewer's concern may be valid based only on Figure 4c, but should be assuaged by the $\delta^{15}$N-PON data available for different seasons in our region, many of which were generated by our group (e.g., Forrer, 2021; Smart et al., 2015; 2020). Where possible, we have endeavoured to place our dataset into broader context, and $\delta^{15}$N-PON is particularly useful in this regard as it yields an integrated view of the autotrophic N uptake regime. Measurements of $\delta^{15}$N-PON allows us to place our N uptake data, which represent only a snapshot in time, into broader temporal context. Nonetheless, we have removed the reference to Figure 4c from the original statement and have softened the language as follows –

> "The **available** $\delta^{15}$N-PON data suggest that this preferential reliance on recycled N **may have** persisted from the late summer."

L726-733: Without supporting observations this is speculative and unsupported.

Response: Please see the general response (2) to the Reviewers above.

L737-747: Relevance? Without any supporting measurements its just not possible to ascertain validity of the statements presented here

Response: Please see our general response (1-ii) above. Since DON degradation is a possible mechanism of $NH_4^+$ supply, we feel it must be addressed, although in the amended version of the manuscript, we have done so more succinctly. Moreover, while we did not measure DON degradation directly, we can make an argument against this process being significant based on our other observations, discussed in the context of previous work done by others.

L748-762: Relevance? No measurements of NH4 input are reported here so the connection is unclear.

Response: Please see our general response (1-ii) regarding including discussion of processes described in the literature. We have nonetheless shortened the text as much as possible but have chosen not to completely remove our discussion of other $NH_4^+$-producing processes that could potentially be significant in the winter Southern Ocean.

**Summary:**

L900: Measurements of heterotrophic NH4 production rates are required to support the speculative scenario outlined in the discussion, not your conclusions

Response: The Reviewer's point is well-taken; we have changed this sentence to –

> "Measurements of heterotrophic $NH_4^+$ production rates are required to confirm the hypothesized seasonal cycle of $NH_4^+$ in the Southern Ocean mixed layer, and higher spatial resolution sampling of plankton community composition and N removal rates

may help to explain local variability in NH$_4^+$ concentrations, particularly near the fronts."

L911: Use of references in a 'summary' section. Not something I agree with personally. The statement being made here should be moved to the discussion.
Response: We have changed section 6 to "*Summary and implications*" (see (3) of our general response to all Reviewers above). Since this sentence outlines a potentially major implication of our study, we have decided to retain it in section 6 rather than move it to the Discussion.

**Figures:**

Generally clear and readable

Fig 5: Please clarify what the black box in the legend represents (dark > 2.7 um) or remove.
Response: The idea was that the >2.7 µm plankton group was represented by the opaque section of the bar (i.e., darker colours on the plot), while the more transparent section of the bar (i.e., lighter colours) represented the 0.3-2.7 µm group, with the total length of the bar (i.e., dark + light) indicating the rate for the bulk (>0.3 µm) community. We have removed the black box from the figure legend and will instead articulate the meaning of the colour scale in the caption (please see the revised figure (Figure R2) below).

Fig 5b: Difficult to read, may need resizing. Please also clarify % uptake by the 0.3-2.7 um size fraction (as per Fig 5a)
Response: We will increase the text size and add the % uptake to Fig. 5b (now broken up into Fig. 5b-d; see the revised figure (Figure R2) below). However, given the comments of Reviewer 3, we will list the % nano+ contribution rather than that of the picoplankton.

**Smith et al.** Biogeochemical controls on wintertime ammonium accumulation in the surface layer of the Southern Ocean

[Figure]

*Figure R2 (Figure 5 in the main manuscript): Surface rates of a) net primary production (NPP) and rates of b) ammonium (ρNH₄⁺), c) nitrate (ρNO₃⁻), and d) urea (ρUrea) uptake by the pico (light colours) and nano+ (dark colours) size fractions, with the full length of the bars indicating the bulk rates, and e) NH₄⁺ oxidation. Error bars indicate ±1 standard deviation of duplicate experiments. The percentage of total NPP and N uptake attributable to the nano+ size fraction is written next to each bar in panels a-d. NPP and NH₄⁺ uptake were not measured for the nano+ size fraction at 58.5°S, and urea uptake was not measured at 50.7°S and 55.5°S. Rates were not measured at the latitudes where no data are shown. In panels b-e, the surface NH₄⁺ concentration at each station is shown by the yellow circles. Leg N stations (at which samples were collected from Niskin bottles fired at 10 m) are indicated by black boxes surrounding the latitude. By contrast, samples were collected at the Leg S stations (no square surrounding the latitude) from the ship's underway system (~7 m). Fronts are indicated with arrows (labeled in panel e), and abbreviations are as in Figure 1. Figure produced using the package ggplot2 (Wickham, 2016).*

**Smith et al.** Biogeochemical controls on wintertime ammonium accumulation in the surface layer of the Southern Ocean

Fig 6 & Fig 7: Just getting to be too small to clearly read fonts etc. Please resize
Response: We have increased the text size to be more readable (please see the revised figures below).

[Figure]

*Figure R3 (Figure 6 in main manuscript): Surface community composition for a) plankton ≥15 μm (enumerated by microscopy) and b) the total community <15 μm (enumerated by flow cytometry). For context, the surface NH₄⁺ concentration at each station is shown by the yellow circles. * indicates stations at which no measurements were made while the absence of a bar with no * indicates that no cells were detected. Note that the abundances shown on panel b (top x-axis) are >2 orders of magnitude greater than those shown in panel a. The "microplankton" shown in panel a are included on panel b (slim black bars) to illustrate the difference in abundance between the micro- and pico+nano populations. The frontal positions are indicated on panel b, with abbreviations as in Figure 1.*

[Figure]

*Figure R4 (Figure 7 in main manuscript): Relative abundances of a) total photosynthetic versus heterotrophic bacteria and b) detritus versus heterotrophic bacteria at the surface for Leg S. The surface NH₄⁺ concentration at each station is indicated by the yellow dots. The values in maroon text on the right side of panel a are the photosynthetic-to-heterotrophic cell ratios. The upper x-axis in panel b begins at 75% in order to highlight the (much smaller) heterotrophic bacterial contribution to the summed detrital + heterotrophic particles. Frontal abbreviations are as in Figure 1.*

Fig 8: If authors retain the long and extensive discussion then perhaps addition of approximate rates and pool sizes to Fig 8 will help.
Response: We appreciate this suggestion and seriously considered including such information in the figure. However, given the Reviewer comments regarding overspeculation, combined with the large uncertainty surrounding approximate rates and pool sizes associated with the processes that we did not

measure (i.e., that we would need to extract from the literature), and since we have now shortened the Discussion considerably, we have decided not to change Figure 8 (renumbered to Figure 9 following reorganization of the manuscript).

**Reviewer 2: Anonymous**

Referee comment on "Biogeochemical controls on wintertime ammonium accumulation in the surface layer of the Southern Ocean" by Shantelle Smith et al., Biogeosciences Discuss., https://doi.org/10.5194/bg-2021-149-RC2, 2021

**General comments**

The authors present a study of ammonium cycling activity during an oceanographic cruise between Cape Town (South Africa) and the Marginal Ice Zone of the Southern Ocean. This is achieved through ship-based experiments. The authors present data that describes DIN concentration, N-assimilation and ammonium oxidation rate data. The manuscript is generally well written, citing appropriate refences in support of the arguments presented. The study addresses an area where there is genuinely very little information. The data is high quality and the insights are therefore valuable. I found the text expansive throughout; topics introduced and discussed are mostly relevant, although the level of detail frequently detracts from the focus of the study. In my view the manuscript would benefit from a considerably sharper focus of the main insights achieved, with less space/reliance dedicated to speculation and inference.

Response: We appreciate the comments from Reviewer 2 regarding the novelty and quality of the data presented. Please see our general responses (1 and 2) above regarding the length and level of speculation.

**Specific comments**

**Abstract:**

L18-37: Abstract, and the manuscript more broadly. My view is that the abstract should focus on the new information/data presented, rather than drawing upon additional data sets to support the conclusions. The data certainly offers support for the implication presented in conclusion of this manuscript (that the Southern Ocean is a net CO2 source for half of the annual cycle). However, my view is that the authors over-reach the scope of their data to draw this conclusion.

Response: Please see our general response above (3) regarding our more thorough integration of the seasonal dataset into all sections of the amended manuscript – while our focus remains the winter, the seasonal $NH_4^+$ concentration data are now fully integrated into the paper, which means that they are presented earlier and in a manner that emphasizes their importance to our interpretation and conclusions. This change should address the Reviewer's concern regarding our mention of the seasonal dataset in the abstract. Additionally, the idea that the Southern Ocean is a net *biological* $CO_2$ source for half the year is not new (e.g., Gibson & Trull, 1999; Gray et al., 2018; Hauck et al., 2015; Mongwe et al., 2018; Shadwick et al., 2015) – what our study shows is that the mechanism(s) underlying this biological $CO_2$ flux include net heterotrophy (as indicated by mixed-layer $NH_4^+$ accumulation), in addition to weak photoautotrophic nutrient drawdown.

**Introduction:**

L38: Introduction - very detailed scene setting, perhaps overly so. Topics introduced are relevant, but expansive. The introduction would benefit from a tighter focus.

Response: Please see the general response (1) to the Reviewers above. In sum, we have shortened the Introduction by 268 words and improved its focus.

L39-49: In the first paragraph there are 11 references. 7 of these are not in the reference list. I didn't continue to check, but this needs to be done.

**Smith et al.** Biogeochemical controls on wintertime ammonium accumulation in the surface layer of the Southern Ocean

Response: We thank the Reviewer for spotting this oversight. All in-text citations are now included in the reference list.

L150-152: While the stoichiometry of nutrient assimilation into organic material is relatively clear, I'm not sure that the stoichiometry of CO2 release per NH4 regenerated is as clear. Does it then follow that the SO may be heterotrophic for half the year? Maybe this hypothesis needs some additional support?

Response: We are not arguing here for the magnitude of the $NH_4^+$ flux, but rather making the point that, since the heterotrophic production of $NH_4^+$ is accompanied by the release of $CO_2$ (regardless of the stoichiometry), if $NH_4^+$ remains elevated in the mixed layer in which it was produced, this evinces a net $CO_2$ source.

L652-657: No graph showing the relationship between particulate stocks (POC, PON) and regenerated N concentrations and uptake is shown.

Response: Unfortunately, we do not know what the Reviewer is referring to here (L652-657 are not in the Introduction, and the text at L652-657 deals with trace metals and nitrifiers), and so cannot respond.

**Methods:**

L164: I am curious as to why the sampling regime differed between the two legs of this cruise - Southward leg involved only surface underway samples, while the Northern leg included CTD casts. How might this inconsistency between legs affect the study (I suspect this difference related to ships logistics; I may have missed this information).

Response: Yes, the reason for the different sampling approaches was purely logistical. Leg N was part of a GEOTRACES-endorsed sampling of the WOCE IO6 line (i.e., involving CTD and GoFlo casts), while Leg S was simply the transit to the Marginal Ice Zone where sampling efforts were focused. There was no time on Leg S to conduct CTD casts. Similarly, during Leg N, the scheduling and small sampling team meant that underway sampling was not possible. We have amended the text as follows –

> "Leg S of VOY25 in winter 2017 crossed the Atlantic sector and due to logistical constraints, involved only surface underway collections, while leg N bordered the Atlantic and Indian sectors (30°E; WOCE IO6 line) and included eight conductivity-temperature-depth (CTD) hydrocast stations"

L171: rosette mounted oxygen sensors are notorious for drifting. Was this unit calibrated?

Response: Yes, the unit was calibrated. We have added the following text –

> "The salinity and oxygen sensors were calibrated against seawater samples that were analyzed for salinity using a Portasal 8410A salinometer and for dissolved oxygen by Winkler titration (Strickland & Parsons, 1972)."

L177-178: are the authors confident that this difference in sample collection methods did not influence the results presented? My concern would specifically be with using the ships underway system for ammonium measurements. The pipework in these systems are far from clean (by scientific standards) and offer an extensive surface for biofilm formation. Microbes growing in such films rapidly exchange resources with their surroundings (i.e. the surface sea water supply), potentially modifying nutrient and gas concentrations. Separately, physically pumping water is likely to disrupt biological processes associated with cells/aggregates in the pumped water (pressure pulses/turbulence). Was a direct comparison between sampling methods done on concentrations/processes?

Response: Within our research group, we have worried extensively about this potential issue and are confident that our sampling approach did not influence the results. First, the underway pipes on the *SA Agulhas II* are lined with epoxy and flushed with bleach and copious quantities of fresh water after and prior to each cruise to remove any microbial growth. Second, within each zone of the Southern Ocean sampled here, the biogeochemical ammonium concentrations measured for the CTD samples fall within the same range as those collected from the underway system. Third, the CTD samples are used only to supplement the underway samples – after all, there are only eight CTD stations (compared to 32 underway stations). That said, in spring 2019 during VOY040 onboard the R/V *SA Agulhas II*, we investigated the possibility that the ship's underway system may affect the $NH_4^+$ concentrations. We collected surface

samples from the underway system and Niskin bottles concurrently and measured a $NH_4^+$ concentration difference between them of $0.07 \pm 0.15$ µM (n=17), with no noticeable trend of one method consistently yielding higher concentrations. We have added some text to the Methods section in this regard.

L204: was the DIC concentration measured to confirm this enrichment, or is it assumed?
Response: DIC concentrations were measured *in situ* by the Council for Scientific Research (unpublished results) using a VINDTA 3C instrument (range of measured DIC concentrations = 2017 to 2130 µM). The concentration of $^{13}C$-DIC tracer in the 2L bottles was 100 µmol/L. We have clarified these points in the amended version of the manuscript.

L210: collected from ships sea-water supply – depth of 7m
Response: We have added the following to the text – "…using seawater from the ship's underway system…"

L215: presumably these concentrations were measured post-cruise to allow accurate enrichment determination? Worth stating this.
Response: Yes. At sea, the $^{15}N$ additions were estimated to be 10% of the ambient concentrations based on published data and our own prior measurements. The concentrations of $^{15}N$ in the various polycarbonate bottles were 0.1 µM and 0.2 µM for $^{15}NH_4^+$ and $^{15}N$-urea, respectively, and for $^{15}NO_3^-$, were 0.1 µM at 37.0°S, 1 µM at all other leg S stations, and 3 µM along leg N. The % enrichments were recalculated post-cruise using the measured ambient nutrient concentrations. We have added text to the Methods to clarify this –
> "$^{15}N$ enrichment was re-calculated post-cruise using the measured nutrient concentrations, and these enrichments were used in all rate calculations.".

L217: If I understand this correctly, a constant 15NH4 addition of 100nM was made to NH4+ oxidation vessels. In Fig 2, the ambient concentration was 0-0.7µM, representing an enrichment of 12-50%. If so, this could lead to a potential overestimation of nitrification rate.
Response: We agree that stimulation of the $NH_4^+$ oxidation rates is generally a potential concern, although in the case of this study, there are reasons to be less worried about it. As we discuss in the manuscript (L667 in the original version), the $K_m$ for $NH_4^+$ oxidation in the winter Southern Ocean has recently been shown to be very low (0.03 to 0.14 µM), with ammonia oxidizers becoming saturated at ambient $NH_4^+$ concentrations of ~0.1-0.2 µM (Mdutyana, 2021). This means that at the stations with ambient $NH_4^+$ concentrations >0.2 uM (i.e., everywhere south of the SAF), the ammonia oxidizers were already substrate-saturated at the ambient $NH_4^+$ concentrations so would not have been stimulated by the addition of $^{15}NH_4^+$. However, at stations in the STZ (37.0°S and 41.9°S) and SAZ (43.0°S and 44.8°S), the $NH_4^+$ concentrations were <0.1 µM, such that rate stimulation is a distinct possibility. Since our primary interest is in explaining the accumulation of $NH_4^+$ south of the SAF, the (in)accuracy of the STZ and SAZ rates is not a significant concern and will not change our interpretation. Nonetheless, we have added a caveat to the Methods text explaining that the $NH_4^+$ oxidation rates from north of the SAF should be considered 'potential rates' due to the possibility of stimulation by $^{15}N$-tracer addition (see our response to Reviewer 1's comment on L682-683 above).

L217: why duplicate analysis (presumably logistics – I appreciate this is a lot of work)? This could limit the confidence in the statistical analysis of results.
Response: Due to the water budget and size of the incubators, as well as the number of experiments run concurrently, we could only sample in duplicate.

L218: I appreciate that carrier N needs to be added under certain circumstances in order to satisfy detection limits of analysis. However, this leads to a loss of sensitivity. Are the authors confident that their measurements were sufficiently separated from the inherent 'noise' of mass spec isotope analysis? Is the ambient NO2 data presented anywhere – I couldn't see it.
Response: $^{14}NO_2^-$ was added as an isotope "trap". This is because "if the ambient product pool is quite small (as is the case for $NO_2^-$ in most of the global ocean), any $^{15}NO_2^-$ […] that is produced is [from $NH_4^+$ oxidation] essentially lost immediately to oxidation to nitrate […]. Addition of the carrier [(i.e., isotope trap)] allows the recently produced product to be diluted into a larger pool, which can then be

recovered at the end of the incubation" [Ward, B.B. (2011). Measurement and distribution of nitrification rates in the ocean. In Microbial Nitrification and Related Processes. M. G. Klotz, editor, *Methods in Enzymology*, 486: 307-323]. This is a standard approach for $NH_4^+$ oxidation measurements made via the analysis of $^{15}NO_2^-$ following its conversion to $^{15}N_2O$ (i.e., Mdutyana et al., 2020; Mdutyana, 2021; Newell et al., 2013; Peng et al., 2018; Ward, 2011), and we are confident that our data are reliable (particularly given that the measurements are made at natural abundance level – i.e., as $\delta^{15}N$ and not At%).

The ambient $NO_2^-$ concentration data are shown in Figure S2b (renumbered to Fig. S3c in the amended version) and briefly described in the Results section (i.e., "The $NO_2^-$ concentrations were consistently low across the transect ($0.16 \pm 0.02$ µM; Fig. S3c) …").

L231: I suspect that surface seawater supply pipework is a higher contamination risk than sample invasion due to temperature gradients…
Response: Please see our response above regarding the underway system. Moreover, we note that we discussed the risk of $NH_4^+$ invasion/efflux (with efflux being our larger concern given the strong temperature dependence of the Henry's Law constant of $NH_3/NH_4^+$) with Malcolm Woodward, the Head of the Nutrient Facility at Plymouth Marine Lab and co-Chair of SCOR working group 147: *Towards comparability of global nutrient data*, and he advised us to take the approach we have detailed in the manuscript.

L269: Are the authors confident that system drift was not an issue during the analysis of so many samples? From personal experience, and that of colleagues who undertake isotope measurements, batch runs of as little as 15 samples, but generally no greater than 25 are used, punctuated by standards and filter blanks. I am not familiar with system described but would be surprised if any MS system was sufficiently stable for a run of this length.
Response: We apologize, the text originally included at L269 was misleading. Eight blank filters were prepared (packaged) with each batch of 88 samples; however, each *batch run* on the mass spectrometer included fewer samples (38 samples, 12 standards, and four blanks) with standards included at regular intervals (after every 5-7 samples) during the batch runs. We have removed the mention of 88 samples and amended the text to clarify the details given above –

> "Unused pre-combusted filters (blanks) were included in each batch run." and "…, internal laboratory standards calibrated against IAEA reference materials that were measured after every 5-7 samples."

L281: again, was DIC measured directly? Perhaps I missed this…
Response: Yes, please refer to the response given above for the query regarding L204.

L306: evidence from elsewhere would support this assumption, but a short consideration of alternative NO2 sources would perhaps be useful. In a well-mixed environment with a sufficient NO3 supply, light transitions can lead to NO2 release from phytoplankton (as well documented). By undertaking bottle incubations, such transitions do not take place thankfully, so this mechanism shouldn't influence results (i.e. by diluting the enriched NO2 pool leading to an underestimation of NH4 oxidation rate).
Response: While we agree that alternate processes ongoing coincident with $NH_4^+$ oxidation could result in the production of $NO_2^-$, it is difficult to imagine how they could result in the production of $^{15}N$-labeled $NO_2^-$. For this to occur via the mechanism given as an example by the Reviewer, $^{15}NH_4^+$ would have to be oxidized (via $^{15}NO_2^-$) to $^{15}NO_3^-$, then taken up by phytoplankton, partially reduced, and subsequently released as $^{15}NO_2^-$. The net effect even in this case (which we assume to be highly unlikely given the relatively short duration of our experiments, the slow nitrification rates, the fact that $^{14}NO_2^-$ was added as an isotope trap, and the high concentration of the ambient $^{14}NO_3^-$ pool) is that the $^{15}NO_2^-$ would have derived from $^{15}NH_4^+$ oxidation.

L315-341: AFC is a relatively standard analytical tool. The level of detail provided is unnecessary – refer to published methods.
Response: We have shortened this section considerably, moving much of the original text to the Supplement (now Text S2). This supplemental section now includes information regarding how we

determined cell size and how we separated the DNA-containing cells from each other, as well as a new figure (Figure S2 in the modified supplement) showing the cytograms and histograms used to identify each population.

L343: 'potential heterotrophic activity was evaluated… '
Response: Please see our general response (2) to the Reviewers above.

L342-350: I doubt this approach can yield useful information about the NH4+ regeneration rate. Cell abundance (any type of cell) is no indication of cellular activity. While particulate material is decomposed leading to the regeneration of inorganic nutrients, the more labile material is likely to be associated with the dissolved organic pool, especially the material actively released from living phytoplankton cells during growth. I appreciate that the authors wish to get a handle on this aspect in order to build a view of regional NH4+ cycling, however I think this stretch detracts from the dataset.
Response: Please see our general response (2) to the Reviewers above.

**Results:**

L398 and onwards within results section: refs and associated text into discussion.
Response: Discussion text has been removed, except for the occasional comparison with previous studies, which we view as evincing the reliability of our results rather than constituting an interpretation/discussion thereof. In total, we have shortened the Results text by 85 words (before the addition of sections 4.6 and 4.7 that show the seasonal data and production rate calculations) and removed seven references.

L1500: Is it necessary to name the software package used to generate figures? I find the figures and their text on the small side, especially Fig 2, 4 and 5 (the latter has a great deal of information and appears rather cluttered), 7, 9.
Response: The software package is open source, as such it is appropriate to reference it. The text size on figures will be increased. Please see the revised Figures 5 and 7 (Fig. R2 and R4) above as part of our response to Reviewer 1. Below are the revised versions of Figures 2, 3, and 4.

[Figure]

*Figure R5 (Figure 2 in main manuscript): Concentrations of dissolved ammonium (NH₄⁺) a) at the surface for Legs S and N and b) with depth (0-300 m) for Leg N, and c) concentrations of nitrate (NO₃⁻) at the surface for Legs S and N. Pink circles in panel b show the mixed layer depth at the CTD stations. Abbreviations are as in Figure 1. Figure produced using the package ggplot2 (Wickham, 2016).*

**Smith et al.** Biogeochemical controls on wintertime ammonium accumulation in the surface layer of the Southern Ocean

[Figure]

*Figure R6 (Figure 3 in main manuscript): a) Bulk chlorophyll-a (chl-a) concentrations and b) the proportion of chlorophyll-a in the nano+ size fraction at the surface for Legs S and N. Abbreviations are as in Figure 1. Figure produced using the package ggplot2 (Wickham, 2016).*

[Figure]

*Figure R7 (Figure 4 in main manuscript): Bulk $\delta^{15}$N-PON at the surface for Leg S in winter 2017. Two stations nearest South Africa at which biomass concentrations were extremely high have been excluded. Abbreviations are as in Figure 1. Figure produced using the package ggplot2 (Wickham, 2016).*

Finally, we will ensure that Figure 9 (now Figure 8 in the amended version of the manuscript) is printed in landscape orientation once the paper is accepted, which will ensure that it is large enough to be easily readable (see next page).

**Smith et al.** Biogeochemical controls on wintertime ammonium accumulation in the surface layer of the Southern Ocean

[Figure]

*Figure R8 (Figure 8 in main manuscript): Surface concentrations of NH₄⁺ across the eastern Atlantic sector of the Southern Ocean measured between December 2018 and November 2019. Five unique transects (additional to the winter 2017 dataset presented in Fig. 2a) are shown: a) early summer 2018, b) late summer 2019, c) winter 2019, d) early spring 2019, and e) late spring 2019. f) The proposed seasonal cycle of NH₄⁺ concentrations in the mixed layer south of the Subantarctic Front. The colour gradient in panel f shows the transition between late summer and late winter. Panels a and b cover a latitudinal extent of 30-70°S, while panels c-e cover 30-60°S due to the presence of sea-ice. Abbreviations are as in Figure 1, with AZ referring to the combined OAZ and PAZ. Figure produced using the package ggplot2 (Wickham, 2016).*

L1532: co plot of cell abundance with [NH4] – what's the rationale for the co-plot? Is there a link suggested or is it to provide context?

Response: Since the focus on the study is understanding NH₄⁺ cycling, we included the concentration data to provide context, not to suggest a relationship. We have updated the caption of the figure as

**Smith et al.** Biogeochemical controls on wintertime ammonium accumulation in the surface layer of the Southern Ocean

follows –

> "For context, the surface $NH_4^+$ concentration at each station is shown by the yellow circles."

That said, we did observe some trends between cell abundance and $NH_4^+$ concentrations that are easier to describe with the $NH_4^+$ data included in the figure (L614) –

> "Additionally, *Synechococcus* abundance was strongly correlated with $NH_4^+$ concentration south of the SAF (r = 0.65)." and "By contrast, nanoeukaryotes, which have a higher per-cell nutrient requirement than the equally-abundant picoeukaryotes, may have dominated $NH_4^+$ uptake in the PFZ and AZ given that higher nanoeukaryote abundances corresponded with lower $NH_4^+$ concentrations at a number of stations (e.g., stations 50.0°S, 51.1°S, and 55.5°S; Fig. 6b)."

L462: '….food source available to heterotrophs…' this statement is somewhat vague. Heterotrophs would include everything from heterotrophic protists to zooplankton and beyond. What is specifically referred to here?

Response: We have changed this paragraph to simply contain a statement of our observations –

> "The contribution of heterotrophic bacteria to total small cells varied considerably (10-62%), reaching a maximum south of the PF at 53.0°S and 57.8°S (62% and 50%), and with higher abundances in the SAZ than in the PFZ and OAZ (Fig. 7). Additionally, heterotrophic bacterial abundances were ten-fold lower to two-fold higher than the total pico- and nanophytoplankton cell counts. Detrital particles were most abundant near the southern edge of the SAF, and were generally more abundant in the PFZ than in the SAZ and OAZ (Fig. S5)."

**Discussion:**

L470-496: While it is important to try and constrain the factors that are significant to NH4+ cycling, my view is that there is too much reach beyond the data. It is not robust to infer process rates from cell or detrital abundance data. The foundation of the paper is the observational data surrounding NH4 assimilation and oxidation, and the new insight this provides. My main criticism of this contribution is that it reaches well beyond this data, to inferred contributions and speculation, to build the view of NH4 cycling. While this view may ultimately be proven to be reasonable, I think a stronger case needs to be made through direct observations of the inferred processes and rates.

Response: Please see our general response (1 and 2) above. In general, we agree with the Reviewer that a stronger case could be made through direct observations. We have thus moderated our conclusions related to the processes that we did not measure. We still feel that it is necessary to mention these other processes, however, since they are highly relevant to our discussion of $NH_4^+$ cycling in the shallow Southern Ocean, but we have taken care to remove as much inference and speculation as possible.

L505: '…growth temperatures of temperate and…' attention.

Response: We apologise but we do not know what the Reviewer is asking for here and so cannot provide a response.

L509: '….and west Indian…' West.

Response: We have changed the directional references in this sentence to 'eastern' and 'western' to avoid using a possibly confusing proper noun.

L589: '…could dampen total…' Not sure what this means.

Response: We have changed the sentence as follows –

> "…the preferential uptake of urea and other DON species by some organisms (e.g., cyano- or heterotrophic bacteria) could cause a net decrease in the total $NH_4^+$ uptake rates."

L639: 'supporting role for iron…' This is speculation.

Response: We have removed this clause from the sentence.

**Smith et al.** Biogeochemical controls on wintertime ammonium accumulation in the surface layer of the Southern Ocean

L659-662: Both NH4 and NO2 are intermediates in a number of microbial processes. It would be difficult to infer anything about how one process influences this balance.
Response: Please see our general response (1-iii) above. This paragraph and Figure S6 (showing a surface map of the ratio) have been removed.

L687: The bacterial decomposition of DON leads to NH4+ regeneration. i.e. not just PON.
Response: Agreed, we have changed 'PON' to 'organic nitrogen'.

L692: 'fresh' PON – specifically, do you mean labile material that can be readily decomposed?
Response: Yes. We have changed 'fresh' to 'labile'.

L699: this is speculation – what support is there for this statement?
Response: Please see our general response (2) above. We have significantly amended this paragraph to make it less speculative.

L705-707: what support is there for a link between the ratio of detrital to heterotrophic particles and the NH4 concentration?
Response: This statement has been removed from the amended manuscript. Additionally, please see our general response (2) above.

L716: 'bacteria more efficient at lower temperatures..' efficient at what? This is loose language.
Response: We have removed the paragraph at L713 to make the section on heterotrophic bacteria more succinct and related to the available data.

L817: I do not follow this. Is this specifically referring to grazing activity? Bacterial activity is predominantly heterotrophic and will most certainly be taking place here.
Response: This is referring to all forms of heterotrophy. We have amended the sentence as follows –
"By late summer, the $NH_4^+$ concentrations increased (Fig. 8b) due to elevated heterotrophic activity (i.e., bacterial decomposition and zooplankton grazing) following the accumulation of algal biomass (Mengesha et al., 1998; Le Moigne et al., 2013), coupled with iron- and/or silicate-limitation of phytoplankton (Hiscock et al., 2003; Sosik & Olson, 2002) and enhanced grazing pressure (Becquevort et al., 2000)."

L831-856: Is this section necessary? This aspect was not directly investigated (it needs dedicated spike experiments). This is an example of discussion and speculation that add little to the manuscript.
Response: Please see the general response (1-iv) above. In short, we have removed the implications section (formerly section 5.3) entirely and have instead incorporated a vastly shortened version of some of the text into our concluding remarks (section 6: *Summary and Implications*).

L857-886: the manuscript now strays a considerable distance from the focus of the study. My view is that this section adds nothing to the discussion of the results.
Response: This text has been removed.

L915: Having read through the discussion I find it hard to pull out the headline from this study – there are steps forward, but they need to be stated more concisely, with less speculation and inference.
Response: Please see our general responses (1-3) above. We have endeavoured to shorten and streamline the manuscript, and the Discussion in particular, removing speculation and non-essential information, with the goal of ensuring that the focus of the study remains clear. We have also better integrated the seasonal dataset into all sections of the manuscript so that it does not read as an 'add-on' in the Discussion.

**Technical corrections:**

L95: 'consumption' and 'assimilation' are used interchangeably. I'd associated consumption with phagotrophy/mixotrophy/grazing. Assimilation is technically more appropriate here as the underlying

process referred to is inorganic nutrient utilisation by phototrophs (nutrient uptake, reduction where necessary and assimilation into organic molecules).

Response: As suggested, we have changed 'consumption' to 'assimilation' throughout the manuscript when referring to nutrient uptake. When referring to the consumption of ammonium more broadly (i.e., by a combination of phytoplankton uptake and nitrification), we have used 'removal'.

**Reviewer 3: Anonymous**

Referee comment on "Biogeochemical controls on wintertime ammonium accumulation in the surface layer of the Southern Ocean" by Shantelle Smith et al., Biogeosciences Discuss., https://doi.org/10.5194/bg-2021-149-RC3, 2021

**General comments**

Ammonium ($NH_4^+$) is an important macronutrient in marine ecosystems and the dynamics of its production, utilisation, and regeneration are reasonably well studied within the marine microbial food web. However, how these dynamics play out in the Southern Ocean is not well understood and this is especially so during the winter months when conditions in this region are challenging due to large storms, low temperatures, limited light availability, and the presence of sea ice. In their paper, Smith et. al provide a detailed snapshot of $NH_4^+$ concentration and dynamics (uptake and oxidation rates) in the surface water and winter mixed layer during a winter voyage in the Atlantic sector, bordering the Indian sector, of the Southern Ocean. To better understand these dynamics, the authors investigate links between macronutrient concentrations, microbial community composition and biomass, net primary production, particulate organic matter, and nitrogen isotopic fractionation. This is a substantial data set to both analyse and interpret and I commend the authors for their very thorough analysis of the data and its links to the available literature on this topic.

There are, however, some weaknesses in the manuscript that need to be addressed. Most notable of these is the presentation of new results and data analysis in the discussion. Particularly Section 5.2, which presents a completely new data set of three additional cruises adjacent to the region being studied. This year-round analysis of $NH_4^+$ in Southern Ocean surface waters is indeed complementary to the current study and allows for analysis of seasonal cycling of $NH_4^+$. However, this seems like a separate paper on its own and is indeed, presented as such in this section (methods, results & discussion). In addition to this, there are also data and analysis presented in the supplement (Text S2 & S3) that appear to be critical for some of the analysis presented in the discussion and have not been presented or referenced at all in the results section.

Visualising such an extensive data set is a difficult task, but a lot of information is being presented in the figures and tables, which makes some of them very difficult to read or interpret, especially at publication size. The authors discuss implications of a better understanding of the seasonal $NH_4^+$ dynamics in the Southern Ocean but don't really explain how this present study may alter or affirm the current knowledge base. Lastly, the authors seem to focus on CO2 uptake and drawdown in the mixed layer in the introduction and conclusion but there is no real mention of this in the discussion. If the authors consider this an important implication of the research being presented and it should also be discussed.

Response: We appreciate the comments made by the Reviewer regarding the novelty of our data and thoroughness of our analysis. Please see our general responses (1-3) above wherein we outline the actions we have taken in response to Reviewer concerns regarding speculation and lack of focus, manuscript length, and our treatment of the seasonal dataset. In brief, we do not want to remove the seasonal dataset from this paper because it adds a great deal to our understanding of $NH_4^+$ cycling in the upper Southern Ocean, as well as allowing us to place our winter 2017 results into stronger temporal context. However, we do agree that it is better having these data fully integrated into the paper.

Text S3 outlines how we calculated $NH_4^+$ residence time and estimated $NH_4^+$ production rates; given Reviewer 3's comment and our integration of the seasonal dataset into the all sections of the amended paper, we have moved a version of this text out of the Supplement and into the main Methods and Results sections. However, we feel that Text S2 does not need to be included in the main paper because it is not referenced broadly in the manuscript, nor is it central to our arguments. Additionally, as pointed

out by all three Reviewers, the manuscript is already lengthy and includes too much text that strays from its main focus.

**Specific comments**

**Introduction:**

L124-135: NH4+ oxidisers are an important group of microorganisms in this study and are discussed in length in the discussion (L643-680), but it is never really explained what this microorganism group is composed of. Please provide some context in the introduction so the reader understands this better.
Response: We have added some more information about the $NH_4^+$ oxidizing groups in the introduction as follows –

"Nitrification, the oxidation of $NH_4^+$ to nitrite ($NO_2^-$) and then $NO_3^-$ by chemoautotrophic bacteria and archaea…"

**Methods:**

L188: what concentration of acetone was used?
Response: We used 90% acetone; this detail has been added to the Methods section.

L208-9: were these filters combusted? Storing them in combusted foil suggests they were?
Response: Yes, all filters were pre-combusted. We have clarified this in the text as –

"…gently vacuum filtered through combusted 0.3 μm, and 2.7 μm glass fibre filters…".

L257-8: here, the authors distinguish each of the fractionated size classes into 0.3-2.7um "picophytoplankton", >2.7um "nanophytoplankton", and >0.3um "bulk" but do not consistently use this terminology throughout the rest of the manuscript. It would aid in understanding if the manuscript was updated so that this terminology was consistently applied through the rest of the analysis and discussion (examples below).
Response: We have made the suggested changes here (to "nano+" and "pico") and now use this nomenclature consistently throughout the manuscript.

L310: the conventional size range for microplankton is 20-200um so this microscopic analysis of cells >5um also includes most of the nanoplankton size range.
Response: We have changed the text here to "(>15 μm)" instead of "(>5-10 μm)". We know from the microscopy analysis that most of the cells were >15 μm, with very few <15 μm. Cells that fell into the <15 μm size range and that were counted under the microscope are not included in the microscopy results presented in the manuscript as we feel that the flow cytometry counts are a lot more accurate. In any case, the exact size of the cells is not integral to the discussion and as such, the changed definition will not affect our conclusions.

L315: there is no mention that the flow cytometry analysis was on cells sized <15um. It is only mentioned in the caption of Fig. 6.
Response: We have added this detail to the Methods.

L328: it appears from the discussion (Section 5.1.2) that these "small heterotrophic cells" are being counted as heterotrophic bacteria. Is this correct? It appears to be implied from the data but not directly specified in this section of the methods.
Response: In essence, yes. We have updated the Methods section to make this clear –

"Additionally, small heterotrophic prokaryotes (i.e., bacteria and possibly archaea; hereafter "bacteria") were identified as DNA-containing particles with the lowest detected autofluorescence (Marie et al., 1997; Gasol & del Giorgio, 2000)."

**Results:**

L383: the >2.7um size class has previously been defined as "nanophytoplankton" (see L257-8 above)

**Smith et al.** Biogeochemical controls on wintertime ammonium accumulation in the surface layer of the Southern Ocean

so there should not be a need to redefine it with a different, albeit very similar, name.

Response: We have 1) defined the size classes only once in the amended version of the paper (at L57 as "…0.3-2.7 µm size class (hereafter, "pico" size class) was calculated by subtracting the measured [chl-a] of the >2.7 µm size class (hereafter, "nano+" size class) from the >0.3 µm size class (hereafter, "bulk").") and 2) ensured that we use the same nomenclature throughout the text.

L380-6: there is no mention of the picophytoplankton (0.2-2.7um) size class results here.

Response: The nano+ contribution is reported as a percent of the total (nano+ and pico) phytoplankton community; therefore, the picophytoplankton contribution is just 100% minus the % of nano+. As such, we think it would be superfluous to report both.

L388-9: the percentage contributions of POC & PON of the nanophytoplankton size class, when SD is taken into account, range from 48.8-112.4% and 19.5-120.1%, respectively. From Table 1 it appears that in the PAZ the proportion of POC in the nanophytoplankton class was 143.02%, and the PON for this class in the PFZ has a SD of 121.41%. I would question an analysis where a size-fractionated class displays values that are far greater than the bulk. Can the authors explain why the POC & PON proportions reported are higher in the >2.7um filters than the >0.3um filters? The authors may want to consider whether the way the data is being presented is appropriate.

Response: The magnitude of the standard deviations to which the Reviewer refers is largely a result of the propagation of error (see Ku, 1966). In the PAZ (and across the entire transect), there was only one station (CTD station at 58.5°S) at which the average concentrations of POC and PON were greater on the >2.7 µm filter than on the 0.3 µm filter. Unfortunately, we only had one 2.7 µm filter replicate for this station and since its measurement is clearly questionable, we have removed it from our analysis. The text has been updated as follows –

> "The concentrations of bulk POC and PON were highest north of the STF and slightly higher in the OAZ than in the SAZ and PFZ (Fig. S4a and b)."

This change also means that we do not report a percent contribution of the nano+ group to total POC at 58.5°S. However, since we made duplicate PON measurements for all the [15]N experiments conducted at this station, we can still present a value for PON. We have also made small corrections (<0.1 µM) to the standard deviations for the PON and POC concentrations in the nano+ size fraction at some CTD stations (58.5°S, 53.5°S, 48.0°S, and 43.0°S) where the error had not been propagated appropriately. These standard deviations are now of the same order of magnitude as for similar samples, and the net result is substantially smaller standard deviations associated with the zone averages (see excerpt from the new version of Table 1 below).

|  | STZ | SAZ | PFZ | OAZ | PAZ |
|---|---|---|---|---|---|
| **POC (bulk) (µM)** | 4.4±6.7 | 3.4±0.4 | 3.2±0.3 | 3.4±0.5 | 3.5+0.2 |
| **POC (nano+) (µM)** | 2.6±0.5 | 2.6±0.4 | 1.9±1.2 | 1.9±0.4 | 4.6 |
| **PON (bulk) (µM)** | 0.6±0.2 | 0.5±0.1 | 0.4±0.1 | 0.5±0.1 | 0.5±0.1 |
| **PON (nano+) (µM)** | 0.3±0.1 | 0.3±0.1 | 0.2±0.3 | 0.2±0.1 | 0.4±0.0 |
| **POC (% of nano+)** | 79.7±24.6 | 79.6±19.0 | 50.9±33.2 | 77.2±21.8 | ND |
| **PON (% of nano+)** | 69.0±31.9 | 67.1±17.2 | 53.8±24.1 | 67.0±21.9 | 51.1±24.7 |
| **POC:chl-a (g g⁻¹)** | 103.0±22.1 | 102.5±14.4 | 122.5±11 | 234.1±29.2 | 219.3±1.0 |
| **POC:PON (M/M)** | 7.81±6.49 | 6.90±1.25 | 7.13±0.71 | 6.72±1.62 | 5.80±3.75 |

L393: the statistical analysis here is not mentioned in the methods and appears to be the only time difference among zones was analysed and reported. Are there significant trends with any of the other factors being assessed? This would be interesting to know.

Response: There are some variables (particularly the nutrient concentrations) that were significantly different between zones, especially between the SAZ and OAZ, and for which the latitudinal gradients are clear from the figures and Table 1; we thus did not include statistics. However, we are happy to include t-test results for these variables if the Editor thinks it necessary.

**Smith et al.** Biogeochemical controls on wintertime ammonium accumulation in the surface layer of the Southern Ocean

In the specific case of the POC-to-PON ratio, we have removed this text from the amended version of the manuscript as part of our effort to make the Discussion more concise and less speculative.

L401: the "small size class (0.2-2.7um)". Another example where a size class (picophytoplankton) has been redefined.
Response: Please see the response above. We have changed "small" here to "pico".

L400-402: this is the only time where the relative contribution of the picophytoplankton size class is presented. It's not really clear why the authors have chosen to present this information in this context.
Response: We have changed this to the contribution of the nano+ size class for consistency.

L423: see comment above, L310, about microplankton size range.
Response: Please see our response above.

L423: Section 4.5 – there is a very big difference in the counts presented between the microscopy (>5-10um) and flow cytometry results (<15um). If there is an overlap between the microscopy and flow cytometry of 5-10um in the nanophytoplankton range, then were there cells in the microscopy samples that were observed but not counted? Can the authors explain this discrepancy?
Response: Cells in the 5-15 μm size range (as reported in the flow cytometry results) *were* counted in the microscopy analysis but are not reported in the manuscript because we do not have high confidence in the accuracy of these counts due to the magnification used (200x). Instead, we rely on the flow cytometry data to accurately represent the smaller cells (<15 μm). We have clarified this in the Method section (at L314 as "The 200x magnification limited the cell sizes that could be reliably distinguished to those ≥15 μm.").

L445: "small cells" have been previously defined as 0.3-2.7um (see L401) so this creates confusion by lack of consistency again.
Response: We have changed all the references to size ranges to "pico", "nano+", and "bulk", as requested by the Reviewer. These are defined by size ranges were determined by the filters used for POC and PON, and the NPP and N uptake experiments. We have also defined the size ranges associated with the microscopy-counted and flow cytometry-counted cells as >15 μm and <15 μm, respectively, to avoid confusion. Since these changes should make things a lot clearer, we have decided to keep using "small cells" in our collective reference to the groups counted using flow cytometry.

**Discussion:**

L596-634: there are a lot of results and correlations presented in this section that should be in the results section.
Response: We respectfully disagree – the results to which the Reviewer refers are included in this paragraph in support of our discussion points and as such, we think it necessary to retain them.

L602-5: "0.3-2.7 um size fraction". This has already been defined by the authors as "picophytoplankton". See comment on L257-8 above.
Response: We have changed this text from "0.3-2.7 μm size fraction" to "pico size fraction".

L630-2: where does this relationship data come from? The correlation between pennate abundance and NH4+ at the PF seems tenuous considering the low pennate count numbers and high variability of NH4+ south of the SAF.
Response: Agreed. The text has been removed as the relationship was determined qualitatively and is in any case not required for our discussion.

L643-5: the data presented here comparing NH4+ oxidation and uptake rates is not at all clear in Fig. 5.
Response: We have reworked this figure such that the y-axes are the same in all N cycle panels (see Figure R2). This should make comparison of the N uptake and $NH_4^+$ oxidation rates much easier.

L660-1: this is new data again.
Response: Please see our response above; in short, these data and the text describing them have been removed from the manuscript.

L685-6: This section references the supplemental Text S2 and I can't see how these two relate. It is also not clear why the results presented in Text S2 are not presented in the results section of the manuscript as they seem to relate to Section 3.2.5 of the methods.
Response: Apologies, the reference should have been to Text S3. We have now integrated Text S3 into the main manuscript text. However, we feel that Text S2 does not need to be included in the main paper because it is not referenced broadly in the manuscript or fundamental to our arguments. Additionally, as pointed out by all three Reviewers, the manuscript is already lengthy and includes too much text that strays from its main focus.

L688-736: I would suggest a reassessment of this entire section on heterotrophic activity by bacteria and zooplankton. It contains a lot of results and there are number of relationships and assumptions made that don't seem strongly correlated to the available data (e.g., L702-3 – the POC:PON relationship with zones is reported at non-significant in the results (L391-3) so I'm not sure a southward increase can be inferred, L730-732 – this assumption is made off a single data point and other stations with similar NH4+
concentrations don't show the same thing).
The presentation of heterotrophic-to-photosynthetic cell ratios is misleading here because the terminology is the wrong way around. The ratio presented on the side of Fig. 7a is the ratio of photosynthetic-to-heterotrophic cells (e.g., 50.0°S is 9.6:1). Thus, it makes no sense to be discussing "higher ratios of heterotrophic-to-photosynthetic cells" when the data presented shows low ratios of photosynthetic-to-heterotrophic cells. This makes the analysis of this section very confusing.
Response: Please see our general response (2) above.
Additionally, we have moved the results text mentioned to the Results section where appropriate and have switched the reported ratio to photosynthetic-to-heterotrophic in the text and the caption of Fig. 7a.

Section 5.2 & 5.3: comments on these sections made in the "General Comments" above.
Response: Please see our general responses (1 and 2) above.

**Figures & Tables:**

I found it very difficult to see all the detail on the figures and interpret them at the size presented in the printed publication. All of the overlays and contour values are quite distracting and make the figures overly complex and difficult to interpret.
Response: As outlined in our responses to Reviewer 1 and 2 above, we have increased the text size on all the figures (see Fig. R2-8). We will also provide higher resolution figures to the journal, which should further improve the readability of the contour labels and other text. We feel that the contours make the changes in the colour gradients less ambiguous, and so have chosen to keep them on the figures. However, we are happy to remove them should the Editor feel it necessary.

Table 1: I'm not sure how the "% of total of >2.7um" values were calculated but they don't seem to add up to the other data presented. I don't think these add anything to the analysis and should be removed.
Response: These data are mentioned in Section 3.2.3 of the Results and are used to provide context to the nano+ nutrient uptake rates. The values of "% of total of >2.7 µm" are calculated as the average of the percentage contribution by the nano+ size fraction at each station in the respective zones. We will add a clarification to the Table 1 caption as follows –

> " The percentage of the total by the >2.7 µm (nano+) size fraction shown for chl-a, POC, and PON, is the average of the percentage contribution calculated for each station within a zone."

Figure 5: there is too much information presented on these figures. It is not clear why the percentage of NPP for the 0.3-2.7 size fraction is presented here. It's also worth noting that the y-axes are different

for each sub-figure, which was not immediately clear. A lot of the data in these figures that are discussed in the manuscript, such as differences in concentrations between zones, is better displayed and easier to interpret in Table 1.

Response: We have amended Figure 5 as follows (please see Figure R2 above): 1) the N uptake rates are presented in separate panels; 2) the y-axis scales are now the same on all panels; 3) % values have been added to panels a-d for consistency, as requested by Reviewer 1; 4) all the text has been made bigger; 5) the $NH_4^+$ concentration data have been moved to a secondary x-axis at the bottom of the panels to reduce confusion, and 6) the % values have been changed to % nano+ contribution for consistency with the rest of the manuscript (where our focus is generally on the nano+ rather than the picoplankton).

Figure 6: microzooplankton is present on the legend in Fig. 6b, which displays flow cytometry data <15um.

Response: The legend label in Figure 6b includes 'microplankton' (not 'microzooplankton'). These data are the total microscopy counts and are included here to show the difference in abundance between the cells <15 μm, counted using flow cytometry, and those >15 μm, counted using microscopy. We have added a clarifying sentence to the caption as follows –

> "'Microplankton' are included on panel b (slim black bars) to illustrate the difference in abundance between the populations shown on panel a (>15 μm) versus panel b (<15 μm)."

Figure 9: it is incredibly difficult to interpret this figure due to the size.

Response: We have amended Figure 9 (renumbered to Figure 8) as follows (please see Figure R8 above): 1) We will request it to be printed in landscape, 2) we have increased the size of all text on all panels, and 3) we have changed the y-axes layout to use the space more efficiently.

**Supplement:**

It can be helpful for the authors to provide additional commentary and background on analyses presented in the manuscript to aid the reader in their understanding. Text S1 is a good example of this. The content of Text S2 & 3 include results and discussion that seem to be integral in their analysis and should be presented in the results section.

Response: Please see our response above regarding Text S2 and S3. Text S3 outlines how we calculated $NH_4^+$ residence time and estimated $NH_4^+$ production rates; given Reviewer 3's comment and our integration of the seasonal dataset into the all sections of the amended paper, we have moved this text out of the Supplement and into the main Methods and Results sections. However, we feel that Text S2 does not need to be included in the main paper because it is not referenced broadly in the manuscript, nor is it fundamental to our arguments. Additionally, as pointed out by all three Reviewers, the manuscript is already lengthy and includes too much text that strays from its main focus.

**Technical Corrections:**

L372: "from <1 uM and <10 uM, respectively, in the STZ",
Response: We have added the suggested text.

L741: the starting point of the rebuttal ("However,") of the previous statements is not very clear. I would suggest using a different phrase.
Response: We have removed the word "however".

L744 – "likely to be",
Response: We have changed this to "…cellular $NH_4^+$ efflux by ammonia oxidisers is likely extremely low…"

L757 – It's not "Finally," if the next sentence starts with "Additionally,". Reword.
Response: We have changed the wording as follows –

[revised manuscript text omitted]

Newell, S.E., Fawcett, S.E. and Ward, B.B., (2013). Depth distribution of ammonia oxidation rates and ammonia-oxidizer community composition in the Sargasso Sea. *Limnology and Oceanography*, 58(4), pp.1491-1500.

Peng, X., Fawcett, S.E., Van Oostende, N., Wolf, M.J., Marconi, D., Sigman, D.M. and Ward, B.B., (2018). Nitrogen uptake and nitrification in the subarctic North Atlantic Ocean. *Limnology and Oceanography*, 63(4), pp.1462-1487.

Reay, D. S., Priddle, J., Nedwell, D. B., Whitehouse, M. J., Ellis-Evans, J. C., Deubert, C., and Connelly, D. P. (2001). Regulation by low temperature of phytoplankton growth and nutrient uptake in the Southern Ocean. *Marine Ecology Progress Series*, 219(1990), pp.51–64.

Shadwick, E.H., Trull, T.W., Tilbrook, B., Sutton, A.J., Schulz, E. and Sabine, C.L., (2015). Seasonality of biological and physical controls on surface ocean CO2 from hourly observations at the Southern Ocean Time Series site south of Australia. *Global Biogeochemical Cycles*, 29(2), pp.223-238.

Talmy, D., Martiny, A.C., Hill, C., Hickman, A.E., and Follows, M.J., (2016). Microzooplankton regulation of surface ocean POC: PON ratios. *Global Biogeochemical Cycles*, 30(2), pp.311-332.

Ward, B.B., (2011). Measurement and distribution of nitrification rates in the oceans. *Methods in enzymology*, 486, pp.307-323.

---

## Author Response (AR2)

**Smith et al.** Biogeochemical controls on wintertime ammonium accumulation in the surface layer of the Southern Ocean

We thank the Reviewers for their final comments and for their recommendation that our manuscript be accepted after we respond to their final notes. Below are our responses, in blue text, to each of the Reviewer comments.

**Reviewer 1: Anonymous**

**General comments**

The authors have done a good job in simplifying this study and their responses to earlier comments seem appropriate and acceptable. Though there remain several speculative assumptions within the discussion, on balance I think that these are now more carefully handled and presented in a manner that makes clear the limitations of what is known about the N cycle in the South Ocean. I note a few areas where the methodological details still need to be clarified (see below) but overall this should make a nice contribution to the literature.
Response: We appreciate the Reviewer's comments regarding our efforts to improve the manuscript.

**Specific comments**

**Abstract:**

L35-37: The presented data would seem to support this statement, however I would encourage the authors to downplay the extrapolation of their results to the entire Southern Ocean. It is evident from recent circumpolar pCO2 observations there is considerable spatial and temporal variability in the strength of the Southern Ocean as a sink or source and it may not be true to state that the entire Southern Ocean becomes a biological source of CO2 for half the year. See for example figure 2 in Sutton et al 2021.
Sutton, A. J., Williams, N. L., & Tilbrook, B. (2021). Constraining Southern Ocean CO2 flux uncertainty using uncrewed surface vehicle observations. Geophysical Research Letters, 48, e2020GL091748.

Response: We take the Reviewer's point. However, our intention with this statement is not to highlight a new finding about the Southern Ocean's carbon sink, but rather to offer an additional (complementary) explanation for why Southern Ocean biology drives a net outgassing of $CO_2$ in winter, with the latter idea one that is already established and widely supported by both model results and observations (e.g., Gibson & Trull, 1999; Gray et al., 2018; Hauck et al., 2015; Mongwe et al., 2018; Shadwick et al., 2015). In other words, while we agree that there is bound to be spatial and temporal variability in the biologically-driven flux of $CO_2$ from ocean to atmosphere, it is already known that *in net*, the biological pump in both the Subantarctic and Antarctic Zones is weak in winter (red lines in **Figure R1** below), with heterotrophic $CO_2$ production occurring at a far higher rate than autotrophic $CO_2$ fixation. In the Antarctic Zone, this wintertime biological $CO_2$ production is stronger than the $CO_2$ drawdown facilitated by the solubility pump (blue line) such that the region becomes a net source (i.e., not just a biological source) of $CO_2$ to the atmosphere. By contrast, in the Subantarctic Zone, the solubility pump is comparatively far stronger in winter and the region never becomes a net $CO_2$ source to the atmosphere *despite being a biological source in winter*. The mechanism that is typically invoked to explain why the wintertime Southern Ocean is a biological source of $CO_2$ (i.e., the lack of photosynthesis) manifests as weak autotrophic nitrate removal, as least relative to the rate of the upward supply of nitrate, which is accompanied by a stoichiometric quantity of $CO_2$. Our point is y that in addition to this commonly invoked mechanism, the sustained net production of ammonium in excess of the autotrophic process that remove it (i.e., evincing net heterotrophy in the Southern Ocean mixed layer) contributes to biological $CO_2$ outgassing in winter.

**Smith et al.** Biogeochemical controls on wintertime ammonium accumulation in the surface layer of the Southern Ocean

[Figure]

*Figure R1: Seasonal cycle of the $CO_2$ flux between the Southern Ocean surface and the atmosphere in all three major ocean basins; positive (negative) values of $FCO_2$ indicates that the Southern Ocean is a $CO_2$ sink (source). The red lines show the modelled biological component of the flux while the blues lines show the modelled physical component (means of numerous models; see Mongwe et al (2018) for details). The black lines show the mean of the observations from Landschützer et al. (2014). [Figure from Mongwe et al. 2018].*

The data presented for the early winter period (May – July 2019) by Sutton et al. (2021) that was referenced by the Reviewer show the same thing – that the wintertime Southern Ocean is a net source of $CO_2$ to the atmosphere while in the summer, it is a sink. We note that the Sutton et al. (2021) dataset is showing the net (i.e., biological + physical) $CO_2$ flux and does not, therefore, directly relate to our argument about the biological flux. Further, Sutton et al. (2021) do not explicitly show spatial variation between the Southern Ocean sectors in winter (or indeed, summer) since their sampling track covers 196 days (i.e., the dominant mode of variability is seasonality). Nonetheless, we have changed "…for half of the year…" to "…in autumn and winter…" in the abstract (line 36), since we cannot know the exact period over which the Southern Ocean is a biological $CO_2$ source, and to avoid the insinuation that during the other half of the year, the Southern Ocean is a $CO_2$ sink.

Finally, we have amended the discussion text to clarify our meaning as follows –

> "In net, the Southern Ocean mixed layer is a biological source of $CO_2$ to the atmosphere in autumn and winter (Mongwe et al., 2018). The persistence of elevated $NH_4^+$ concentrations across the polar Southern Ocean between late summer and winter implies that this biological $CO_2$ production occurs not only because $NO_3^-$ drawdown is weak relative to $NO_3^-$ supply at this time (e.g., Gibson & Trull, 1999; Gray et al., 2018; Hauck et al., 2015; Mongwe et al., 2018; Shadwick et al., 2015), but also because the ambient conditions allow for $NH_4^+$ accumulation."

**Methods:**

Some care and clarification needed in presentation of methodological details.

L192-200: It is not clear from the method description that the authors conducted size-fractionated N uptake experiments though this is indicated later on L269-270, L420-421 etc. Nor is it clear from the description whether the authors split the duplicate 1L bottles to obtain the size fractions in each bottle i.e. 500 ml per fraction or whether they used 1 replicate for the bulk and the second replicate for the nanoplankton fractions. A minor detail, but please clarify the methods used.

**Smith et al.** Biogeochemical controls on wintertime ammonium accumulation in the surface layer of the Southern Ocean

Response: We have added to the sentence at L199 –

"Incubations **and filtration** were carried out as for NPP**, although 500 mL was used per size fraction**."

L248: Heading for section 3.2.4 indicates that only bulk POC, PON and d15N measurements were collected. Text on L406-408 implies size-fractionated POC and PON data exist. If the latter is true please amend section 2.3.4 with the analytical details.

Response: We have amended the methods sentence to the following –

"Duplicate seawater samples (4 L) were also gently vacuum-filtered through combusted 47 mm-diameter, 0.3 μm GF-75 **and 2.7 μm Grade-D** filters for POC and PON concentrations and $\delta^{15}$N-PON."

L357: Typo - NH4 consumption rate is not defined in equations 6- 8. Please check and correct

Response: We have fixed the typo to the intended term as in equation 8 –

"Where, $NH_4^+{}_{\text{removal rate}} = \rho NH_4^+ + NH_4^+{}_{\text{ox}}$."

**Results:**

L467 & Fig 7: Figure 7 could be improved by merging the two subplots such that the "% of particles" is reflective of the actual proportions observed (i.e. contribution to the total particle count, photo + hetero + detrital). Currently, the approach used is a little confusing, with Fig7a indicating that heterotrophic cells represent ~30% of particles (photo + hetero) (or as described in the text the % of small cells L465), and Fig 7b suggesting heterotrophic cells represent ~<10% of particles (hetero + detrital). Given the presentation of results on L467, would it not be clearer to state the proportion of each particle type in a single figure where the % contribution is the contribution to the total particle count (photo + hetero + detrital)? The real value of Fig 7a and 7b only appears in the discussion (L698-701) and not really in the results section.

Response: We have heeded the suggestion of the Reviewer and combined Figure 7a and 7b (see Figure R2 below). We have moved the previous version of Figure 7 to the supplement as Figure S5.

[Figure]

*Figure R2 (Figure 7 in main manuscript): Relative contributions of photosynthetic, heterotrophic bacterial, and detrital particles to*

**Smith et al.** Biogeochemical controls on wintertime ammonium accumulation in the surface layer of the Southern Ocean

*the total flow cytometry counts at the surface during leg S. The coincident $NH_4^+$ concentrations are shown as yellow dots. Abbreviations are as in Figure 1.*

Section 4.7 & L523: To be fair to your intended readership can the authors add their estimates of NH4 residence times north of the SAF to the text for completeness

Response: We have changed the results paragraphs in section 4.7 to include the results from north of the SAF as follows –

"The $NH_4^+{}_{residence\ time}$ in winter 2017, computed using Eqn 5, ranged from 10 to 38 days (median of 21 days) south of the SAF and **from 0 to 6 days (median of 2 days)** north of the SAF. These values were estimated using wintertime measurements only and as such, may not be representative of the transition from summer to winter. To refine our estimates, we used average $\rho NH_4^+$ and $NH_4^+$ concentration measurements. South of the SAF in late summer, $\rho NH_4^+ = 50.6 \pm 24.0$ nM day$^{-1}$ and the $NH_4^+$ concentration $= 0.81 \pm 0.92$ µM (Deary, 2020), which together yield an $NH_4^+{}_{residence\ time}$ of 2 to 27 days (median of 5 days). **The $NH_4^+{}_{residence\ time}$ north of the SAF, calculated using $\rho NH_4^+ = 20.7 \pm 8.6$ nM day$^{-1}$ and $NH_4^+$ concentration $= 0.16 \pm 0.45$ µM (Deary, 2020) was 1 to 17 days (median of 14 days).**

The $NH_4^+{}_{production\ rate}$ south of the SAF, calculated using Eqn 8 and an $[NH_4^+]_{decline}$ of 330 nM (i.e., the difference between late summer and winter 2019; 810 nM – 480 nM), $t$ of 141 days, and $NH_4^+{}_{removal\ rate}$ of $50.6 \pm 24.0$ nM day$^{-1}$ (here, the average late-summer $\rho NH_4^+$ south of the SAF is used to approximate $NH_4^+{}_{removal\ rate}$), was $52.9 \pm 25.0$ nM day$^{-1}$. **Similarly, north of the SAF (using an $[NH_4^+]_{decline}$ of 20 nM, i.e., 160 nM – 140 nM, and $NH_4^+{}_{removal\ rate}$ of $20.7 \pm 8.6$ nM day$^{-1}$), the $NH_4^+{}_{production\ rate}$ was $50.7 \pm 9.3$ nM day$^{-1}$.** If we instead use the average $NH_4^+{}_{removal\ rate}$ and $NH_4^+$ concentration measured in winter 2017 south ($21.4 \pm 0.6$ nM day$^{-1}$ and $520 \pm 110$ nM) **and north ($18.4 \pm 0.8$ nM day$^{-1}$ and $80 \pm 10$ nM)** of the SAF, the $NH_4^+{}_{production\ rate}$ was $23.4 \pm 6.6$ nM day$^{-1}$ **and $18.5 \pm 6.6$ nM day$^{-1}$**, respectively. Using the range of $NH_4^+{}_{removal\ rate}$ estimates and the average ambient $NH_4^+$ concentration measured south of the SAF in winter 2017 (16.7 to 31.2 nM day$^{-1}$ and 520 nM) and late summer 2019 (22.6 to 98.6 nM day$^{-1}$ and 810 nM), we calculate that over the late-summer-to-winter transition, the $NH_4^+{}_{production\ rate}$ ranged from 18.8 to 100.9 nM day$^{-1}$ **(compared to 6.3 to 28.8 nM day$^{-1}$ north of the SAF)**."

**Discussion:**

L712-722: The role of zooplankton in NH4 production is certainly important in some studies (see Hernandez-Leon et al 2008; Priddle et al 1997 etc) but this section presents a rather limited assessment of its significance due to the acknowledged limitations of the dataset. It remains an interesting interpretation (already modified by the authors) but perhaps one that should still be strengthened with some literature support.
S. Hernández-León, C. Fraga, T. Ikeda, A global estimation of mesozooplankton ammonium excretion in the open ocean, Journal of Plankton Research, Volume 30, Issue 5, May 2008, Pages 577–585.
Priddle et al., (1997). Diurnal changes in near-surface ammonium concentration – interplay between zooplankton and phytoplankton. J. Plankton Res, 19(9), 1305-1330.
Response:
We have no doubt that at times, zooplankton make a potentially significant contribution to the mixed-layer NH$_4$ pool in the Southern Ocean. However, in response to the first round of review of the manuscript, we have tried to keep speculation in this regard to a minimum, particularly given our limited dataset (as noted by the Reviewer). That said, for the wintertime Southern Ocean, we do not think it is unreasonable to suggest a higher contribution to the $NH_4^+$ supply by heterotrophic bacteria than zooplankton given the low biomass concentrations (i.e., low food supply) and low zooplankton abundances that we observed during our sampling. It is likely that the zooplankton contribution to the $NH_4^+$ flux is far more significant in (late) summer and near the fronts (i.e., in response to elevated phytoplankton biomass following the growing

season and driven by frontal upwelling, respectively). Indeed, our late summer and spring 2019 datasets appear to show evidence of both these scenarios.

We have thus added the following sentence and relevant references –

> "That said, it is possible that the contribution of micro- (and/or macro-) zooplankton to the $NH_4^+$ pool surpasses that of heterotrophic bacteria under certain conditions (Koike et al., 1986; Priddle et al., 1998), such as in (late) summer and near regions of frontal upwelling in response to elevated rates of phytoplankton biomass accumulation."

We also allude to this possibility at L850 –

> "By late summer, the $NH_4^+$ concentrations increased (Fig. 8b) presumably due to elevated heterotrophic activity (i.e., bacterial decomposition and zooplankton grazing) following the accumulation of algal biomass (Mengesha et al., 1998; Le Moigne et al., 2013)."

We feel that we cannot make any more conclusive statements than the above given that previous studies have shown a highly variable contribution by zooplankton to the $NH_4$ pool in the Southern Ocean (i.e., <0.5 – 82%; Alcaraz et al., 1998; Atkinson & Whitehouse, 2001; Hernández-León et al., 2008; Whitehouse et al., 2011).

**Reviewer 2: Anonymous**

Referee comment on "Biogeochemical controls on wintertime ammonium accumulation in the surface layer of the Southern Ocean" by Shantelle Smith et al., Biogeosciences Discuss., https://doi.org/10.5194/bg-2021-149-RC2, 2021

**General comments**

The authors have done an excellent job of updating their manuscript following the previous reviewer's comments. I have two minor comments on this version to be considered for revision.

Response: We appreciate the comments from the Reviewer regarding our efforts on the revised manuscript.

**Specific comments**

**Methods:**

1. In the Methods Sec 3.1 there is no mention of how the NH4+ samples were taken on the 2018/19 voyages. Were they taken following the same method in L161-169 and were they underway or CTD samples?

Response: We have added the following sentence to the paragraph at L161-169 –

> "During the 2018-2019 cruises, $NH_4^+$ samples were collected every two hours from the ship's underway system."

**Discussion:**

2. In the Discussion Sec 5.2 the authors attribute that changes in the NH4+ concentration in late summer were "due to" (L758) a range of biological processes. However, they can only support this hypothesis by the literature, as these measurements were not made in this study. I suggest a slight change in the wording here to reflect that.

Response: We have changed the wording here to "…presumably due to…" to indicate our reliance on the literature as opposed to measurements made during the study.

**Smith et al.** Biogeochemical controls on wintertime ammonium accumulation in the surface layer of the Southern Ocean

**References**

Alcaraz, M., Saiz, E., Fernandez, J.A., Trepat, I., Figueiras, F., Calbet, A. and Bautista, B., 1998. Antarctic zooplankton metabolism: carbon requirements and ammonium excretion of salps and crustacean zooplankton in the vicinity of the Bransfield Strait during January 1994. *Journal of Marine Systems*, *17*(1-4), pp.347-359.

Atkinson, A. and Whitehouse, M., 2001. Ammonium regeneration by Antarctic mesozooplankton: an allometric approach. *Marine Biology*, *139*(2), pp.301-312.

Gibson, J.A. and Trull, T.W., (1999). Annual cycle of fCO2 under sea-ice and in open water in Prydz Bay, East Antarctica. *Marine Chemistry*, 66(3-4), pp.187-200.

Gray, A.R., Johnson, K.S., Bushinsky, S.M., Riser, S.C., Russell, J.L., Talley, L.D., Wanninkhof, R., Williams, N.L. and Sarmiento, J.L., (2018). Autonomous biogeochemical floats detect significant carbon dioxide outgassing in the high-latitude Southern Ocean. *Geophysical Research Letters*, 45(17), pp.9049-9057.

Hauck, J., Völker, C., Wolf-Gladrow, D.A., Laufkötter, C., Vogt, M., Aumont, O., Bopp, L., Buitenhuis, E.T., Doney, S.C., Dunne, J. and Gruber, N., (2015). On the Southern Ocean CO2 uptake and the role of the biological carbon pump in the 21st century. *Global Biogeochemical Cycles*, 29(9), pp.1451-1470.

Hernández-León, S., Fraga, C. and Ikeda, T., 2008. A global estimation of mesozooplankton ammonium excretion in the open ocean. *Journal of Plankton Research*, *30*(5), pp.577-585.

Mongwe, N., Vichi, M. and Monteiro, P., (2018). The seasonal cycle of pCO 2 and CO 2 fluxes in the Southern Ocean: diagnosing anomalies in CMIP5 Earth system models. *Biogeosciences*, 15(9), pp.2851-2872.

Shadwick, E.H., Trull, T.W., Tilbrook, B., Sutton, A.J., Schulz, E., and Sabine, C.L., (2015). Seasonality of biological and physical controls on surface ocean CO2 from hourly observations at the Southern Ocean Time Series site south of Australia. *Global Biogeochemical Cycles*, *29*(2), pp.223-238.

Whitehouse, M.J., Atkinson, A. and Rees, A.P., 2011. Close coupling between ammonium uptake by phytoplankton and excretion by Antarctic krill, Euphausia superba. *Deep Sea Research Part I: Oceanographic Research Papers*, *58*(7), pp.725-732.